# BAYES-NASH GENERATIVE PRIVACY PROTECTION AGAINST MEMBERSHIP INFERENCE ATTACKS

## ABSTRACT

Membership inference attacks (MIAs) expose significant privacy risks by determining whether an individual's data is in a dataset. While differential privacy (DP) mitigates such risks, it faces challenges in general when achieving an optimal balance between privacy and utility, often requiring intractable sensitivity calculations and limiting flexibility in complex compositions. We propose a game-theoretic framework that models privacy protection as a Bayesian game between a defender and an attacker, solved using a general-sum Generative Adversarial Network (general-sum GAN). The Bayes Generative Privacy (BGP) response, based on cross-entropy loss, defines the attacker's optimal strategy, leading to the Bayes-Nash Generative Privacy (BNGP) strategy, which achieves the optimal privacy-utility trade-off tailored to the defender's preferences. The BNGP strategy avoids sensitivity calculations, supports compositions of correlated mechanisms, and is robust to the attacker's heterogeneous preferences over true and false positives. A case study on binary dataset summary statistics demonstrates its superiority over likelihood ratio test (LRT)-based attacks, including the uniformly most powerful LRT. Empirical results confirm BNGP's effectiveness.

## 1 INTRODUCTION

Membership inference attacks (MIAs) exploit vulnerabilities in data analysis and machine learning, enabling adversaries to determine whether an individual's data is included in a dataset, such as medical records or training data. MIAs represent not only a significant privacy threat but also a dominant method for assessing privacy risks. To mitigate the privacy risks, noise perturbation strategies like differential privacy (DP) (Dwork, 2006) introduce randomness to reduce information leakage. DP provides strong theoretical guarantees by ensuring probabilistic near-indistinguishability of an individual's presence in a dataset based on the output of data sharing and processing mechanisms.

However, privacy protection inevitably leads to a tradeoff: adding noise increases uncertainty but reduces data utility, while insufficient noise leaves sensitive information vulnerable to inference attacks. Balancing this privacy-utility trade-off is essential for effective privacy protection across diverse applications. The DP framework quantifies privacy preservation using an $(\epsilon, \delta)$ scheme, which measures the extent to which individual privacy is protected. However, this scheme does not fully capture the utility of the released aggregate information, because for a given $(\epsilon, \delta)$ (representing a specific privacy level), different noise perturbation methods—such as varying noise distributions (e.g., Gaussian or Laplace) or magnitudes can yield differing levels of utility (Geng et al., 2020). Identifying optimal privacy parameters and noise mechanisms is therefore crucial to achieving an optimal trade-off for a given objective using any privacy-preserving framework, DP included.

Despite its theoretical appeal, guaranteeing a desired level of DP for a data-processing mechanism is often challenging. For instance, calculating sensitivity—the maximum possible change in the output when a single data point is replaced—is generally NP-hard (Xiao & Tao, 2008). Furthermore, determining the optimal composition of multiple independent DP mechanisms is $\#P$-complete (Murtagh & Vadhan, 2015). Moreover, the tight characterization of aggregate differential privacy risk under the composition of multiple mechanisms with arbitrary correlation remains open. These challenges complicate the design of DP mechanisms that optimally balance privacy and utility, particularly in scenarios involving multiple dataset accesses.

In this paper, we propose a novel game-theoretic framework to address the optimal privacy-utility trade-off by conceptualizing privacy risk as the outcome of strategic interactions between a *defender* and an *attacker*. We model this interaction as a general-sum Bayesian game, where the defender optimizes privacy while preserving utility, and the attacker seeks to perform MIAs. To solve for the Bayes-Nash Equilibrium (BNE), we introduce the general-sum Generative Adversarial Network (general-sum GAN), where the defender's privacy strategy acts as the generator and the attacker's MIA strategy serves as the discriminator. At the core of this approach is the Bayes Generative Privacy (BGP) response, which defines the attacker's best response to the defender's privacy strategy by minimizing a cross-entropy loss that quantifies the discrepancy between the prior distribution of the sensitive information and the attacker's probabilistic inference. The resulting privacy strategy, termed the Bayes-Nash Generative Privacy (BNGP) strategy, achieves the optimal privacy-utility trade-off tailored to the defender's preferences.

The BNGP strategy offers several key advantages. To address the attacker's heterogeneous preferences for true positives and false positives, we extend the membership advantage (MA) (Yeom et al., 2018) to a Bayes-weighted MA (BWMA). The BGP response captures the defender's worst-case privacy risk regardless of the attacker's preferences in terms of BWMA, ensuring no alternative strategy achieves strictly better privacy or utility for a given trade-off objective. Furthermore, the BGP response satisfies post-processing and composition properties, enabling BNGP strategies to optimize privacy for complex compositions involving arbitrary correlations—surpassing the typical independent mechanism assumptions of DP. In addition, we show that each BNGP privacy strategy is also an optimal approximate DP framework for a given trade-off objective. Furthermore, we establish a necessary and sufficient condition for equivalence between BNGP privacy and pure $\epsilon$-DP for a given choice of $\epsilon$. Unlike DP, BNGP avoids intractable sensitivity calculations for privacy guarantees and worst-case proofs for composition, and it can also handle compositions of correlated mechanisms. To demonstrate the effectiveness of our approach, we present a case study on sharing binary dataset summary statistics. Under mild assumptions, we show that a bounded-rational Bayesian attacker with a non-informative prior incurs higher worst-case loss for the defender than the uniformly most powerful likelihood ratio test (LRT) per the Neyman-Pearson lemma (Neyman & Pearson, 1933). Empirical results further confirm the efficacy of the BNGP strategy in achieving superior privacy-preserving sharing of summary statistics and classification models.

**Organization** Section 2 provides the necessary preliminaries. Section 3 introduces the Bayesian game framework for modeling privacy protection against MIAs. Sections 3.1 and 3.2 formally define the general-sum GAN, BGP response, and BNGP strategy, and explore their properties and relationship with differential privacy. Section 4 presents a case study on sharing summary statistics, comparing the proposed approach to state-of-the-art LRT methods. Section 5 discusses numerical experiments that validate the effectiveness of our framework, and Section 6 concludes the paper.

## 1.1 RELATED WORK

**Quantitative Notions of Privacy Leakage** Quantitative notions of privacy leakage have been extensively studied in various contexts which provides mathematically rigorous frameworks for measuring the amount of sensitive information that may be inferred by attackers. Differential privacy (Dwork et al., 2006; Dwork, 2006) and its variants (Bun & Steinke, 2016; Dwork & Rothblum, 2016; Mironov, 2017; Bun et al., 2018) formalize the privacy leakage using various parameterized statistical divergence metrics. For example, Rényi differential privacy (RDP) (Mironov, 2017) generalizes the standard pure DP and quantifies the privacy leakage through the use of Rényi divergence. Information-theoretic measures, such as mutual information (Chatzikokolakis et al., 2010; Cuff & Yu, 2016), f-divergence (Xiao & Devadas, 2023), and Fisher information (Farokhi & Sandberg, 2017; Hannun et al., 2021; Guo et al., 2022), provide alternative ways to quantify and characterize privacy loss. Empirical measurements are also widely studied (Shokri et al., 2017; Yeom et al., 2018; Nasr et al., 2021; Stock et al., 2022) that quantify the actual privacy guarantees or leakage under certain privacy protection methods.

**Privacy-Utility Trade-off** Balancing the trade-off between privacy and utility is a central challenge in designing privacy-preserving mechanisms. This balance is often modeled as an optimization problem (Lebanon et al., 2009; Sankar et al., 2013; Lopuhaä-Zwakenberg & Goseling, 2024; Ghosh et al., 2009; Gupte & Sundararajan, 2010; Geng et al., 2020; du Pin Calmon & Fawaz, 2012; Alghamdi et al., 2022; Goseling & Lopuhaä-Zwakenberg, 2022). For instance, Ghosh et al. (2009)

formulated a loss-minimizing problem constrained by differential privacy and demonstrated that the geometric mechanism is universally optimal in Bayesian settings. Similarly, optimization problems can be framed with utility constraints Lebanon et al. (2009); Alghamdi et al. (2022). Moreover, Gupte & Sundararajan (2010) modeled the trade-off as a zero-sum game, where the privacy mechanism maximizes privacy while information consumers minimize their worst-case loss using side information.

**GAN in Privacy** The use of generative adversarial networks (GANs) for privacy protection has gained increasing attention in recent years. Huang et al. (2018) introduced generative adversarial privacy (GAP), which frames privacy protection as a non-cooperative game between a generator (defender) and an adversarial discriminator (attacker). In GAP, the generator creates data that retains target utility while obfuscating sensitive information, while the discriminator attempts to identify the private data. The objective is to train a model that not only achieves high utility but is also resilient to the most powerful inference attacks (i.e., high privacy). Similar efforts have also been proposed in the form of compressive adversarial privacy (CAP), which compresses data before the adversarial training step to enhance privacy (Chen et al., 2018). Nasr et al. (2018) proposed an adversarial regularization method that mitigates this type of attack by adjusting the training process to reduce the information leakage from the model. Similarly, Jordon et al. (2018) presented PATE-GAN, combining the Private Aggregation of Teacher Ensembles (PATE) framework with GANs to generate synthetic data with differential privacy guarantees. Other works in this line includes privacy-preserving adversarial networks (Tripathy et al., 2019), reconstructive adversarial network (Liu et al., 2019b), and federated GAN (Rasouli et al., 2020). Adversarial training has also been applied to defend against MIAs specifically. For example, Li et al. (2021) explored methods where models are trained alongside adversaries attempting MIAs, which enables the models to learn representations that are less susceptible to such attacks. There are also works using GAN to perform attacks, where the generator represents the attacker's strategy (Baluja & Fischer, 2017; Hitaj et al., 2017; Zhao et al., 2018; Liu et al., 2019a; Hayes et al.).

## 2 PRELIMINARIES: MEMBERSHIP INFERENCE ATTACK

Membership inference attacks (MIA) aim to infer whether a particular data point is a part of the input dataset of a data analysis mechanism, which could output summary statistics (Sankararaman et al., 2009; Dwork et al., 2015), any model learned from the dataset (Abadi et al., 2016; Shokri et al., 2017), or other information or signals such as network traffic when processing the dataset (Chen et al., 2010). Let $U = [K]$ be a population of $K$ individuals, where each individual $k$ has a *data point* $d_k$ (e.g., a feature vector). Let the binary vector $b = (b_1, b_2, \ldots, b_K) \in W \equiv \{0,1\}^K$ denote the *membership vector*, where each $b_k \in \{0,1\}$. The membership vector $b$ indicates whether each data point is included in the dataset $B = \{b, d\}$, where $d = (d_k)_{k \in U}$; specifically, a data point $d_k$ is included in $B$ if and only if $b_k = 1$. Suppose the underlying distribution of the dataset induces a prior distribution of the membership, denoted by $\theta(\cdot) \in \Delta(W)$. Consider a (potentially randomized) mechanism $f(B) \in \mathcal{X}$, which takes the dataset $B$ as input and outputs $x \in \mathcal{X}$, where $\mathcal{X}$ is a set of outputs.

**Example: Summary Statistic Sharing** Consider a population $U \equiv [K]$ of $K$ individuals, where each individual's data is represented by a binary vector $d_k = (d_{kj})_{j \in Q}$, with $d_{kj} \in \{0,1\}$ specifies the binary value of the specific attribute at position $j$. The set $Q$ represents the set of all attributes under consideration, such as genomic positions or other features. The dataset $B = \{b, d\}$ includes a membership vector $b$ and data points $d = \{d_k\}_{k \in U}$, where an individual's data is included only if $b_k = 1$. The data-sharing mechanism $f(B) = x = (x_1, \ldots, x_{|Q|}) \in \mathcal{X} = [0,1]^{|Q|}$ computes the summary statistics $x$, which is the fraction of individuals with $d_{kj} = 1$ at each attribute $j$. For example, in genomic data, $d_k$ may represent single-nucleotide variants (SNVs), where each $d_{kj}$ indicates the presence of an alternate allele at SNV $j$ of individual $k$. The summary statistic in this case, known as the alternate allele frequency (AAF), is computed as $x_j = \frac{1}{\sum_k b_k} \sum_k b_k d_{kj}$, reflecting the fraction of individuals with the alternate allele at each SNV.

An MIA model is a (possibly randomized) mechanism $\mathcal{A}(d_k, x) \in \{0,1\}$, which predicts the individual's membership information given the target individual $k$'s data point and the output of the mechanism $f$. The standard *membership advantage* (MA) (Yeom et al., 2018) is a common perfor-

mance measure of the MIA model, defined for each $k \in U$ as:

$$\text{Adv}_k \left( \mathcal{A} \right) \equiv \Pr \left[ \mathcal{A}(d_k, x) = 1 | b_k = 1 \right] - \Pr \left[ \mathcal{A}(d_k, x) = 1 | b_k = 0 \right]. \tag{1}$$

In other words, $\text{Adv}_k \left( \mathcal{A} \right)$ captures the difference between the model $\mathcal{A}$'s *true positive rate* (TPR) and *false positive rate* (FPR). Other metrics used to assess MIA performance include accuracy (Shokri et al., 2017), area under the curve (AUC) (Chen et al., 2020), mutual membership information leakage (Farokhi & Kaafar, 2020), and privacy-leakage disparity (Zhong et al., 2022). For a comprehensive review, see (Niu et al., 2024).

## 3 PRIVACY PROTECTION AGAINST MIA AS A BAYESIAN GAME

We define the data curator of the private dataset $B$ as the *defender*, tasked with protecting privacy against MIA, and the entity performing MIA as the *attacker*.

**Defender** To protect membership privacy, the defender randomizes the mechanism $f$ via *noise perturbation*. Let $g_D : W \mapsto \Delta(\mathcal{D})$ denote the *privacy strategy*, where $g_D(\delta|b)$ specifies the probability distribution over noise $\delta \in \mathcal{D}$. The privacy strategy may also be independent of the membership vector, i.e., $g_D(\cdot) \in \Delta(\mathcal{D})$. The randomized version of $f$ is represented as the mechanism $\mathcal{M}(\cdot; g_D)$, and $\rho_D : W \mapsto \Delta(\mathcal{X})$ is the density function induced by $g_D$ and $f$. The defender, modeled as a Von Neumann-Morgenstern (vNM) decision-maker, aims to minimize the expected privacy loss.

**Noise Perturbation** Our noise perturbation aligns with standard randomization paradigms in DP, including input Dwork et al. (2006), objective Chaudhuri et al. (2011), and output Dwork et al. (2006) perturbations. When an output $x = \mathcal{M}(B; g_D)$ is realized with $g_D$ drawing a noise $\delta$, we denote it as $x = \text{r}(\delta)$. In output perturbation, $\delta$ is added to the output $\hat{x} = f(B)$, and the publicly released output is $x = \text{r}(\delta) = \text{R}(\hat{x} + \delta)$, where $\text{R}(\cdot)$ ensures the perturbed $x$ remains within the valid range $\mathcal{X}$. For example, as described in Section 2, when $\hat{x}$ represents frequencies, the formulation $x = \text{R}(\hat{x} + \delta) \equiv \text{Clip}_{[0,1]}(\hat{x} + \delta)$ ensures $x \in [0,1]^{|Q|}$.

**Attacker** The attacker performs MIA and aims to infer the true membership vector. We consider the attacker as a strategic Bayesian decision-maker, with their external knowledge represented by *subjective prior beliefs* about $b \in W$, denoted by $\sigma(\cdot) \in \Delta(W)$. We refer to this as a $\sigma$-*Bayesian attack*. The attacker employs a mixed strategy $h_A : \mathcal{X} \mapsto \Delta(W)$, which assigns a probability distribution over $W$ based on the output of $\mathcal{M}(\cdot; g_D)$. The MIA model is then written as $\mathcal{A}(\cdot; h_A, \sigma) \in \{0, 1\}$.

The attacker may face trade-offs between maximizing privacy extraction and operational costs of post-processing the inferred membership information (e.g., for personalized medicine or marketing). These costs can affect their preference for true positives and true negatives. We extend (1) to the *Bayes-weighted membership advantage* (BWMA) by introducing a coefficient $0 < \gamma \leq 1$ to weight TPR and FPR. That is, BWMA is defined as:

$$\text{Adv}^\gamma \left( h_A, g_D \right) \equiv (1 - \gamma)\text{TPR} \left( h_A, g_D \right) - \gamma\text{FPR} \left( h_A, g_D \right), \tag{2}$$

where $\text{TPR} \left( h_A, g_D \right) \equiv \sum_{k \in U, b_{-k}} \Pr \left[ \mathcal{A}(d_k, x; h_A, \sigma) = 1 | b_k = 1; g_D \right] \theta(b_k = 1, b_{-k})$ is the TPR, and $\text{FPR} \left( h_A, g_D \right) \equiv \sum_{k \in U, b_{-k}} \Pr \left[ \mathcal{A}(d_k, x; h_A, \sigma) = 1 | b_k = 0; g_D \right] \theta(b_k = 0, b_{-k})$ is the FPR. Decreasing $\lambda$ indicates a stronger preference for TPR while increasing $\lambda$ reflects a greater preference for FPR. When $\lambda = 0.5$, the attacker values TPR and FPR equally.

**Attacker's Expected Loss** Let $s = (s_k)_{k \in U} \in W$ denote the inference output of $h_A$. Define

$$\ell_A(s, b, \gamma) \equiv -v(s, b) + \gamma c_A(s),$$

where $v(s, b) \equiv \sum_{k \in U} s_k b_k$ captures the sum of true positives, and $c_A(s) \equiv \sum_{k \in U} s_k$ captures the operational costs to post-process positive inference outcomes (i.e., $s_k = 1$, for all $k \in U$). Maximizing $v(s, b)$ reflects maximizing true positives, while minimizing $c_A(s)$ reflects minimizing the operational costs. Given a privacy strategy $g_D$ (and the induced $\rho_D$), prior $\sigma$, and the attacker's strategy $h_A$, the expected loss is defined as:

$$\mathcal{L}_A^\gamma(g_D, h_A) \equiv \sum_{s,b} \int_x \ell_A(s, b, \gamma) h_A(s|x) \rho_D(x|b) dx \sigma(b). \tag{3}$$

**Proposition 1.** *Suppose* $\sigma = \theta$. *Then, for any* $g_D$, $h_A$, *and* $0 < \gamma \leq 1$, *we have* $\mathcal{L}_A^\gamma(g_D, h_A) = -\text{Adv}^\gamma \left( h_A, g_D \right)$.

Proposition 1 shows that when $\sigma = \theta$ and $0 < \gamma \leq 1$, the attacker's optimal strategy simultaneously minimizes $\mathcal{L}_A^\gamma(g_D, h_A)$ and maximizes $\text{Adv}^\gamma(h_A, g_D)$ for any given $g_D$. This equivalence simplifies the defender-attacker interaction by modeling it as a Bayesian game, where $s$ represents the attacker's pure strategy.

Given any $g_D$, define the *maximum MA* as $\text{Adv}_k(g_D) \equiv \max_{h_A} \{\text{TPR}(h_A, g_D) - \text{FPR}(h_A, g_D)\}$.

**Proposition 2.** *Let $g_D$ and $g'_D$ be two defense strategies, and suppose $\sigma = \theta$. Then, $\text{Adv}_k(g_D) \geq \text{Adv}_k(g'_D)$ for all $k \in U$ iff $\max_{h_A} \text{Adv}^{0.5}(h_A, g_D) \geq \max_{h_A} \text{Adv}^{0.5}(h_A, g'_D)$.*

Proposition 2 establishes that the ordering of the privacy strength of the defender's strategies, where privacy risk is measured by the standard per-individual membership advantages (MAs), can be fully characterized by the ordering of the Bayes-weighted membership advantage (BWMA). In other words, comparing the BWMA is sufficient to determine which privacy strategy offers stronger protection in terms of per-individual privacy risk.

**Defender's Expected Loss** The defender aims to optimally balance the privacy-utility trade-off. Given any $g_D$, $h_A$, let $\mathcal{L}_D(g_D, h_A)$ represent the *expected loss function*. We consider TPR or standard MA (hence $\text{Adv}^{0.5}$) as the *defender's perceived privacy risk* and impose Assumption 1 on $\mathcal{L}_D$.

**Assumption 1.** *For a given $g_D$, the defender's expected loss $\mathcal{L}_D(g_D, h_A)$ increases as either $\text{TPR}(h_A, g_D)$ or $\text{Adv}^{0.5}(h_A, g_D)$ increases.*

Assumption 1 establishes a relationship between the defender's expected loss and privacy risk (dependent on $h_A$) under a given $g_D$, indicating that as privacy risk increases, the defender incurs greater loss. The defender aims to minimize privacy risk while maximizing the utility of the mechanism $\mathcal{M}$. A common class of $\mathcal{L}_D(g_D, h_A)$ satisfying Assumption 1 is an additive combination of privacy risk and utility loss, with the utility loss independent of $h_A$. A useful way to model the utility loss is by the deviation of $x = \mathcal{M}(B; g_D)$ from the unperturbed output $\hat{x} = f(B)$. Specifically, let $\ell_U : \mathbb{R}_+ \mapsto \mathbb{R}_+$ be an increasing, differentiable function, and let $\|\cdot\|_p$ be a norm on $\mathcal{X}$, for $p \geq 1$. The *utility loss* is then defined by $\ell_U(\|x - \hat{x}\|_p)$. The defender's *privacy loss* can be either $v(s, b)$ (capturing TPR) or $-\ell_A(s, b, 0.5)$ (capturing MA). For simplicity, we use the membership vector $b$ to represent the dataset $B = \{b, d\}$. If the defender's privacy risk is measured by TPR, the loss function is expressed as

$$\ell_D(b, s) \equiv v(s, b) + \kappa \ell_U(\|\mathcal{M}(b; g_D) - f(b)\|_p). \tag{4}$$

Given any $g_D$ (and the induced $\rho_D$) and $h_A$, the defender's expected loss $\mathcal{L}_D$ is then given by

$$\mathcal{L}_D(g_D, h_A) \equiv \sum_{s,b} \int_x \ell_D(b, s) h_A(s|x) \rho_D(x|b) dx \theta(b). \tag{5}$$

The interaction between the defender and attacker is modeled as a game, with each optimizing their strategy. A $\sigma$-*Bayesian Nash Equilibrium* represents the point where neither can unilaterally improve their outcome.

**Definition 1** ($\sigma$-Bayes Nash Equilibrium). *Let $0 < \gamma \leq 1$. A profile $\langle g_D^*, h_A^* \rangle$ is a $\sigma$-Bayesian Nash Equilibrium ($\sigma$-BNE) if*

$$g_D^* \in \arg\min_{g_D} \mathcal{L}_D(g_D, h_A^*) \text{ and } h_A^* \in \arg\min_{h_A} \mathcal{L}_A^\gamma(g_D^*, h_A). \tag{6}$$

### 3.1 BAYES-NASH GENERATIVE PRIVACY MECHANISM

We train the BNE strategies using a GAN-like approach, termed general-sum GAN. The defender's strategy is represented by a neural network *generator* $G_{\lambda_D}(b, \nu)$, parameterized by $\lambda_D$, which takes the true membership vector $b$ and an auxiliary vector $v$ as inputs, outputting a noise vector $\delta$. Here, we assume that the auxiliary vector $\nu$ of dimension $q$ has entries uniformly distributed in $[0, 1]$, denoted by $\nu \sim \mathcal{U}$. The attacker's strategy is represented by a neural network *discriminator* $H_{\lambda_A}(x)$, parameterized by $\lambda_A$, which takes as input $x = \text{r}(G_{\lambda_D}(b, \nu))$ and outputs an inference $s \in W$, where $\text{r}(\cdot)$ represents the relationship between $\delta$ and $x$.

We use $G$ and $H$ to represent the general forms of the models $G_{\lambda_D}$ and $H_{\lambda_A}$, without reference to specific parameterization. Unless otherwise specified, $G$ and $H$ will be used in analysis, where the particular parameterization is not essential. For ease of exposition, we focus on output perturbation,

where $x = \mathtt{r}(\delta) = \mathtt{R}(\hat{x} + \delta)$, with $\hat{x} = f(b)$ as the unperturbed output. Our method applies to the general formulation of the privacy-utility trade-off objective under Assumption 1. Here, we use $\ell_U(\|\delta\|_{\mathtt{p}})$ as the utility loss for simplicity, as minimizing $\ell_U(\|\delta\|_{\mathtt{p}})$ also minimizes $\ell_U(\|\mathtt{R}(\hat{x} + \delta) - \hat{x}\|_{\mathtt{p}})$. Define the defender's and attacker's expected loss functions as:

$$\widetilde{\mathcal{L}}_D(G, H) \equiv \mathbb{E}^{\nu \sim \mathcal{U}}_{b \sim \theta} \left[ v\left(H\left(r\left(G(b, \nu)\right)\right), b\right) + \kappa \ell_U\left(\|G(b, \nu)\|\right) \right],$$

$$\widetilde{\mathcal{L}}^{\gamma}_A(G, H) \equiv \mathbb{E}^{\nu \sim \mathcal{U}}_{b \sim \sigma} \left[ -v\left(H\left(r\left(G(b, \nu)\right)\right), b\right) + \gamma c_A\left(H\left(r\left(G(b, \nu)\right)\right)\right) \right].$$

Then, the defender and the attacker play the following game:

$$G^* \in \arg\min_G \widetilde{\mathcal{L}}_D(G, H^*), H^* \in \arg\min_H \widetilde{\mathcal{L}}^{\gamma}_A(G^*, H). \tag{7}$$

This equilibrium reformulates the $\sigma$-BNE using neural networks. The $G$ and $H$ implicitly define probability distributions that match the mixed strategies $g_D$ and $h_A$, respectively. With abuse of notation, we denote $\mathtt{TPR}(\cdot)$ and $\mathtt{Adv}^{\gamma}(\cdot)$ by substituting $h_A$ and $g_D$ by $H$ and $G$. Hence, we have $\widetilde{\mathcal{L}}_D(G, H) = \mathcal{L}_D(g_D, h_A)$ and $\widetilde{\mathcal{L}}^{\gamma}_A(G, H) = \mathcal{L}^{\gamma}_A(g_D, h_A)$. Moreover, if $\mathcal{L}_D(g_D, h_A)$ satisfies Assumption 1, then so does $\widetilde{\mathcal{L}}_D(G, H)$. If $G$ and $H$ are idealized, nonparametric models with infinite capacity that accurately represent the true distributions of $g_D$ and $h_A$, then Proposition 1 implies $H^* \in \arg\max_H \mathtt{Adv}^{\gamma}(H, G^*)$ if and only if $h^*_A \in \arg\min_{h_A} \mathcal{L}^{\gamma}_A(g^*_D, h_A)$. Here, $G^*$ is the privacy strategy that achieves the optimal privacy-utility trade-off captured by $\widetilde{\mathcal{L}}_D$ under the worst-case privacy loss that can be induced.

**Proxies of Loss Functions** Since the function $v(s, b)$ requires binary outputs from $H$, it is inherently discrete. However, using sigmoid activation functions in the neural networks (particularly for $H$) results in continuous outputs, which makes $v(s, b)$ unsuitable for gradient-based optimization due to non-differentiability. We provide proxies for $v$ and $\ell_A$. Let $p = (p_k)_{k \in U}$, where each $p_k \in (0, 1)$, denote the output of $H$. We substitute $v(s, b)$ with $v(p, b) \equiv \sum_{k \in U} p_k b_k$, and use the *binary cross-entropy loss* for $\ell_A$, defined as $\widehat{\ell}_A(p, b) = -\sum_{k \in U}\left(b_k \log(p_k) + (1 - b_k) \log(1 - p_k)\right)$. Thus, the attacker's *expected cross-entropy loss* (CEL) is given by:

$$\mathcal{L}_{\mathrm{CEL}}(G, H) \equiv \mathbb{E}^{\nu \sim \mathcal{U}}_{b \sim \sigma} \left[ \widehat{\ell}_A\left(H\left(r\left(G(b, \nu)\right)\right), b\right) \right]. \tag{8}$$

**Definition 2** (Bayes Generative Privacy Response). *Given any $G$, the* Bayes generative privacy response (BGP response) *to $G$ is defined as $H^* \in \arg\min_H \mathcal{L}_{\mathrm{CEL}}(G, H)$.*

**Definition 3** (Bayes-Nash Generative Privacy Strategy). *The model $G^*$ is a* Bayes-Nash generative privacy (BNGP) strategy *for a given objective function $\widetilde{\mathcal{L}}_D(\cdot)$ and subjective prior $\sigma$ if it is constrained by the BGP response: $G^* \in \arg\min_G \widetilde{\mathcal{L}}_D(G, H^*), H^* \in \arg\min_H \mathcal{L}_{\mathrm{CEL}}(G^*, H)$.*

**Theorem 1.** *Let $G^*$ be a BNGP strategy for $\widetilde{\mathcal{L}}_D$ and $\sigma$, and let $H^*$ be a BGP response to $G^*$. Suppose that $\widetilde{\mathcal{L}}_D$ satisfies Assumption 1. Then, for any $G' \in \arg\min_G \widetilde{\mathcal{L}}_D(G, H')$ with $H' \in \arg\min_H \widetilde{\mathcal{L}}^{\gamma}_A(G', H)$ where $0 < \gamma \le 1$, and for any $\widehat{H}$, we have:*

(i) $\mathtt{TPR}(\widehat{H}, G^*) \le \mathtt{TPR}(H^*, G^*) \le \mathtt{TPR}(H', G')$.

(ii) $\mathtt{Adv}^{0.5}(\widehat{H}, G^*) \le \mathtt{Adv}^{0.5}(H^*, G^*) \le \mathtt{Adv}^{0.5}(H', G')$.

By definition, a BNGP strategy $G^*$ responds to the BGP response $H^*$, ensuring the optimal privacy-utility trade-off by considering the worst-case privacy loss when the attacker minimizes $\mathcal{L}_{\mathrm{CEL}}$. It is important to note that $\mathtt{TPR}$, $\mathtt{Adv}^{\gamma}$, $\widetilde{\mathcal{L}}_D$ are independent of $\mathcal{L}_{\mathrm{CEL}}$. Theorem 1 establishes that $G^*$ achieves the optimal privacy-utility trade-off given $\widetilde{\mathcal{L}}_D$ by leveraging the worst-case privacy risk under the chosen privacy strategy. Specifically, the first inequalities in (i) and (ii) show that, under $G^*$, an attacker using $H^*$ achieves the worst-case privacy risk for the defender, and no other attacker can induce a strictly higher privacy loss in terms of $\mathtt{TPR}$ or $\mathtt{Adv}^{0.5}$. The second inequalities in (i) and (ii) further demonstrate that $G^*$ minimizes the defender's perceived privacy risk, ensuring that no alternative privacy strategy $G'$ achieves a strictly lower privacy loss against the worst-case attacker.

## 3.2 PROPERTIES OF BGP RESPONSE

The BGP risk enjoys the properties of *post-processing* and *composition*. The post-processing property requires that processing a data-sharing mechanism's output cannot increase input data information. Let $\mathrm{Proc} : \mathcal{X} \mapsto \mathcal{Z}$ be a mechanism mapping $\mathcal{M}(b; G) \in \mathcal{X}$ to $\mathcal{Z}$, creating a new mechanism

Proc $\circ \mathcal{M}(b; G) \in \mathcal{Z}$. Proc $\circ G$ denotes the effective randomization device for Proc $\circ \mathcal{M}(\cdot; G)$. Proposition 3 shows that the BGP risk satisfies the post-processing property.

**Proposition 3** (Post-Processing). *Suppose that $G$ has BGP risk $H \in \arg\min_H \mathcal{L}_{\text{CEL}}(G, H)$ and $\widehat{\mathcal{L}}_A^\sigma(G, H)$. Suppose in addition that for any* Proc, Proc $\circ G$ *has BGP risk $H' \in \arg\min_H \mathcal{L}_{\text{CEL}}(\text{Proc} \circ G, H)$. Then, $\mathcal{L}_{\text{CEL}}(\text{Proc} \circ G, H') \geq \mathcal{L}_{\text{CEL}}(G, H)$.*

Consider a profile $\vec{G} = \{G_1, \ldots, G_n\}$ for $1 \leq n < \infty$, where each $G_j$ corresponds to the density function $g_D^j$. With a slight abuse of notation, let $\mathcal{M}_j(G_j) : \mathcal{B} \mapsto \mathcal{X}^j$ denote the mechanism $\mathcal{M}_j(g_D^j)$ (i.e., the randomized version of the mechanism $f^j$) for all $j \in [n]$, where $\mathcal{X}^j$ represents the output space of $\mathcal{M}_j$. Additionally, let $\rho_D^j : \mathcal{B} \mapsto \Delta(\mathcal{X}^j)$ denote the underlying density function of $\mathcal{M}_j(G_j)$.

Define the composition $\mathcal{M}(\vec{G}) : \mathcal{B} \mapsto \prod_{j=1}^n \mathcal{X}^j$ of mechanisms $\mathcal{M}_1(G_1), \ldots, \mathcal{M}_n(G_n)$ as

$$\mathcal{M}(b; \vec{G}) \equiv (\mathcal{M}_1(b; G_1), \ldots, \mathcal{M}_n(b; G_n)).$$

The *joint density function* of $\mathcal{M}(b; \vec{G})$, denoted by $\vec{\rho}_D : \mathcal{B} \mapsto \Delta\left(\prod_{j=1}^n \mathcal{X}^j\right)$, encodes any underlying correlations among the mechanisms. Mechanisms in $\mathcal{M}(\vec{G})$ are independent if $\vec{\rho}_D(x^1, \ldots, x^n | B) = \prod_{j=1}^n \rho_D^j(x^j | B)$; otherwise, they are correlated. For simplicity, let $\vec{r}(\vec{G}(b)) \equiv (r_1(G_1(b)), \ldots, r_n(G_n(b))) = \vec{x} = (x_1, \ldots, x_n)$.

Let $H(\vec{r}(\vec{G}(b)))$ denote the attacker's discriminator that utilizes all outputs (irrespective of their order), and let $H_j(r_j(G_j(b)))$ represent the discriminator that takes only $r_j(G_j(b))$ as input.

**Proposition 4** (Composition). *Suppose that $\mathcal{M}(\vec{G})$ is a composition of $n$ mechanisms with arbitrary correlation. Then, we have, for $H^* \in \arg\min_H \mathcal{L}_{\text{CEL}}(\vec{G}, H)$, $H_j^* \in \arg\min_{H_j} \mathcal{L}_{\text{CEL}}(G_j, H_j)$ for all $j \in [n]$,*

$$\mathcal{L}_{\text{CEL}}(\vec{G}, \vec{H}^*) = \sum_{j=1}^n \mathcal{L}_{\text{CEL}}(G_j, H_j^*) - \Lambda(\vec{G}, \theta).$$

*If mechanisms are* independent, *then $\Lambda(\vec{G}, \theta) = -\sum_b \theta(b) \int_{\vec{\mathcal{X}}} \vec{\rho}_D(\vec{x}|b) \cdot \log\left(\sum_{b'} \vec{\rho}_D(\vec{x}|b')\theta(b')\right) d\vec{x}$. If mechanisms are* correlated, *$\Lambda(\vec{G}, \theta) = -\sum_b \theta(b) \int_{\vec{\mathcal{X}}} \vec{\rho}_D(\vec{x}|b) \log\left(\frac{\sum_{b'} \vec{\rho}_D(\vec{x}|b')\theta(b')}{P(\vec{x})}\right) d\vec{x}$, where $P(\vec{x}) = \prod_{j=1}^n \sum_{b'} \int_{\vec{\mathcal{X}}_{-j}} \vec{\rho}_D(x_j, \vec{x}_{-j}|b')\theta(b')d\vec{x}_{-j}$.*

Proposition 4 demonstrates that when privacy risk is quantified in terms of the minimum $\mathcal{L}_{\text{CEL}}$ (induced by the BGP response) for a given $\vec{G}$, the privacy risk adheres to an additive composition property.

### 3.2.1 RELATIONSHIP TO DIFFERENTIAL PRIVACY

Differential privacy (DP) (Dwork et al., 2006) ensures data analysis outputs remain nearly indistinguishable regardless of an individual's inclusion, hindering membership inference attacks (MIA). For adjacent datasets $B \simeq B'$ differing by one entry, a mechanism $\mathcal{M}(G)$ satisfies $(\epsilon, \xi)$-DP ($\epsilon \geq 0$, $\xi \in [0, 1]$) if for all measurable $\widehat{\mathcal{X}} \subseteq \mathcal{X}$: $\Pr[\mathcal{M}(B; G) \in S] \leq e^\epsilon \Pr[\mathcal{M}(B'; G) \in S] + \xi$.

To align with the standard DP framework, we assume that each individual's membership information is independent of the others. With a slight abuse of notation, we represent $\theta(b)$ as $(\theta^1(b_1), \ldots, \theta^K(b_K))$, a vector of independent priors for each individual's membership.

**Proposition 5.** *Let $\vec{G}^* = \{G_1^*, \ldots, G_n^*\}$. Let $\mathcal{M}(\vec{G}^*)$ be a composition of $n \geq 1$ mechanisms with arbitrary correlation, where each $G_j^*$ is BNGP strategies for some $\widetilde{\mathcal{L}}_D^j$ satisfying Assumption 1. Then, $\mathcal{M}(\vec{G}^*)$ is $(\epsilon, \xi)$-DP for some $\epsilon \geq 0$ and $\xi \in [0, 1]$.*

Proposition 5 demonstrates that every mechanism employing a BNGP strategy profile is also differentially private. However, it does not specify the corresponding DP parameters. In the following, we outline how to design a BNGP strategy (or profile under composition) when the defender selects a specific value of $\epsilon$.

For any $\epsilon \geq 0$ and any $\vec{G} = (G_1, \ldots, G_n)$, we define the following set:

$$\text{DPH}\left[\vec{G}; \epsilon\right] \equiv \left\{ H \,\middle|\, \begin{array}{c} \theta(b)e^{-\epsilon} \leq H\left(\vec{\mathfrak{r}}\left(\vec{G}(b)\right)\right) \leq \theta(b)e^{\epsilon}, \forall b \in W \\ 1 - (1-\theta(b))\,e^{\epsilon} \leq H\left(\vec{\mathfrak{r}}\left(\vec{G}(b)\right)\right) \leq 1 - (1-\theta(b))\,e^{-\epsilon}, \forall b \in W \end{array} \right\}. \quad (9)$$

**Definition 4** ($\epsilon$-Bayes Generative Bounded Privacy Response). *The $\epsilon$-Bayes Generative Bounded Privacy response ($\epsilon$-BGBP response) for any* $\vec{G} = \{G_1, G_2, \ldots, G_n\}$ *is defined as* $H^* \in \arg\min_H \mathcal{L}_{\text{CEL}}(\vec{G}, H) \bigcap \text{DPH}[\vec{G}; \epsilon]$.

An $\epsilon$-BGBP response satisfies both (i) the conditions of a BGP response and (ii) the linear constraints in $\text{DPH}[\vec{G}; \epsilon]$. However, the attacker optimizing $\mathcal{L}_{\text{CEL}}$ does not consider $\text{DPH}[\vec{G}; \epsilon]$ as a constraint in their optimization. In other words, $\text{DPH}[\vec{G}; \epsilon]$ is not a restriction on the attacker's strategy. Instead, it is the defender's choice of $\vec{G}$ that must ensure the induced attacker's BGP response also satisfies the constraints in $\text{DPH}[\vec{G}; \epsilon]$. That is, $\text{DPH}[\vec{G}; \epsilon]$ constrains the defender's optimization problem.

**Proposition 6.** *For any* $\vec{G} = \{G_1, G_2, \ldots, G_n\}$ *and* $\epsilon \geq 0$, *the composition* $\mathcal{M}(\vec{G})$ *of* $n \geq 1$ *mechanisms is* $\epsilon$-*DP iff* all the BGP responses to $\vec{G}$ are $\epsilon$-BGBP responses.

Proposition 6 establishes the necessity and sufficiency of using the BGBP response to implement a pure ($\xi = 0$) differentially private mechanism. Consequently, for a composition (or a single mechanism) $\mathcal{M}(\vec{G}^*)$ to satisfy $\epsilon$-DP, the defender selects $\vec{G}^*$ based on a given $\epsilon$, ensuring:

$$\vec{G}^* \in \arg\min_{\vec{G}} \widetilde{\mathcal{L}}_D\left(\vec{G}, \text{s.t. } H^*\right), \quad H^* \in \arg\min_H \mathcal{L}_{\text{CEL}}(\vec{G}, H) \cap \text{DPH}[\vec{G}; \epsilon].$$

This choice of $\vec{G}^*$ guarantees that $\mathcal{M}(\vec{G})$ is an $\epsilon$-DP mechanism that optimally balances the privacy-utility trade-off for a given privacy risk characterized by $\epsilon$.

# 4 MIA IN SHARING SUMMARY STATISTICS

In this section, we apply Bayesian game-theoretic privacy protection to the sharing of summary statistics from binary datasets, as outlined in Section 2. Assuming the attributes in each $d_k$ are independent, SNVs can be prefiltered to retain only those in *linkage equilibrium* (Kimura, 1965). An MIA attacker uses the summary statistics $x$ output by $f(B)$ to infer whether specific individuals $k \in U$ belong to the private dataset $B$. We compare our Bayesian model with state-of-the-art (SOTA) Frequentist attacks, including *fixed(-threshold) LRT* (Sankararaman et al., 2009; Shringarpure & Bustamante, 2015; Venkatesaramani et al., 2021; 2023), *adaptive LRT* (Venkatesaramani et al., 2021; 2023), and the *optimal* LRT. These attacks rely on the log-*likelihood ratio statistic* $\text{lrs}(d_k, x)$, which compares observed summary statistics $x$ to *reference frequencies* $\bar{p}_j$ derived from a population dataset independent of $(b, d)$. Detailed definitions of these models and loss functions are provided in Appendix C.

The fixed LRT attacker determines whether individual $k$ is part of the dataset by rejecting $H_0^k$ (absence) in favor of $H_1^k$ (presence) if $\text{lrs}(d_k, x) \leq \tau$, where the fixed $\tau$ balances Type-I ($\alpha_\tau$) and Type-II ($\beta_\tau$) errors. The adaptive LRT dynamically adjusts $\tau^{(N)}$ using reference population data to refine the hypothesis test. The optimal LRT minimizes Type-II error $\beta_{\tau^*}$ for a given $\alpha_{\tau^*}$, achieving the most powerful test by Neyman-Pearson lemma (Neyman & Pearson, 1933). *Optimal $\alpha$-LRT attacks* refer to Likelihood Ratio Tests that are Neyman-Pearson optimal at a fixed significance level $\alpha$. The worst-case privacy loss (WCPL), representing the defender's strategy $g_D$ under each attack, is defined as the expected value of $v(s, b)$. For the optimal, adaptive, and fixed LRT attacks, the WCPL is denoted by $L_{\text{Opt-LRT}}^\alpha(g_D)$, $L_{\text{Adp}}^\alpha(g_D)$, and $L_{\text{Fixed}}^\alpha(g_D)$, respectively (see Appendix C for explicit definitions).

Let $L(g_D, h_A) \equiv \sum_{b,s} \int_x v(s,b) h_A(s|x) \rho_D(x|b) dx \theta(b)$ denote the expected true positive rate. Under $\sigma$-Bayesian attacks using the BGP response, the worst-case privacy loss (WCPL) is given by $L_{\text{Bayesian}}^\sigma(g_D) \equiv \max_{H^* \in \arg\min_H \mathcal{L}_{\text{CEL}}(G,H)} L(g_D, h_A)$, where $G$ and $H$ are neural networks (ideal, non-parameterized) that implicitly define $g_D$ and $h_A$.

In the absence of parameterized priors over $W$, we assume a uniform distribution as the non-informative prior, consistent with Laplace's principle (Fienberg, 2006). Our analysis focuses on subjective priors, considering their informativeness relative to the true prior $\theta$. For simplicity, let $\mathcal{BR}^\sigma[g_D] \equiv \{h_A^* \mid H^* \in \arg\min_H \mathcal{L}_{\text{CEL}}(G, H), H \text{ defines } h_A^*\}$. When $\sigma = \theta$, we denote this set as $\mathcal{BR}^\theta[g_D]$.

**Definition 5** (Aligned and Misaligned $\sigma$). *For a fixed* $g_D$, $\sigma$ *is* (weakly) informative *if* $L(g_D, h_A^\sigma) \leq L(g_D, h_A^\theta)$, *where* $h_A^\sigma \in \mathcal{BR}^\sigma[g_D]$ *and* $h_A^\theta \in \mathcal{BR}^\theta[g_D]$. *It is* non-informative *if uniformly distributed over* $W$, aligned *if either informative or non-informative, and* misaligned *otherwise.* $\sigma$ *is* strictly informative *if the inequality is strict.*

**Theorem 2.** *Fix any $g_D$ and $\alpha$. If $\sigma \in \Delta(W)$ is an* aligned prior, *then:*

$$L^\sigma_{\text{Bayesian}}(g_D) \geq L^\alpha_{\text{Opt-LRT}}(g_D) \geq L^\alpha_{\text{Adp}}(g_D) \geq L^\alpha_{\text{Fixed}}(g_D).$$

*If $\sigma$ is* strictly aligned, *then $L^\sigma_{\text{Bay}}(g_D) > L^\alpha_{\text{Opt-LRT}}(g_D)$.*

Theorem 2 establishes a ranking of WCPL from the defender's perspective across four types of attacks: $\sigma$-Bayesian, optimal $\alpha$-LRT, adaptive $\alpha$-LRT, and fixed $\alpha$-LRT. Among these, the $\sigma$-Bayesian attack produces the highest WCPL. However, this ordering—particularly between $L^\sigma_{\text{Bayesian}}(g_D)$ and $L^\alpha_{\text{Opt-LRT}}(g_D)$—may not hold if the attacker's prior is misaligned. Appendix D shows an example of how to comparison between $\sigma$-Bayesian attack and the $\alpha$-LRT attack when $\sigma$ is an arbitrary subjective prior.

## 5 EXPERIMENTS

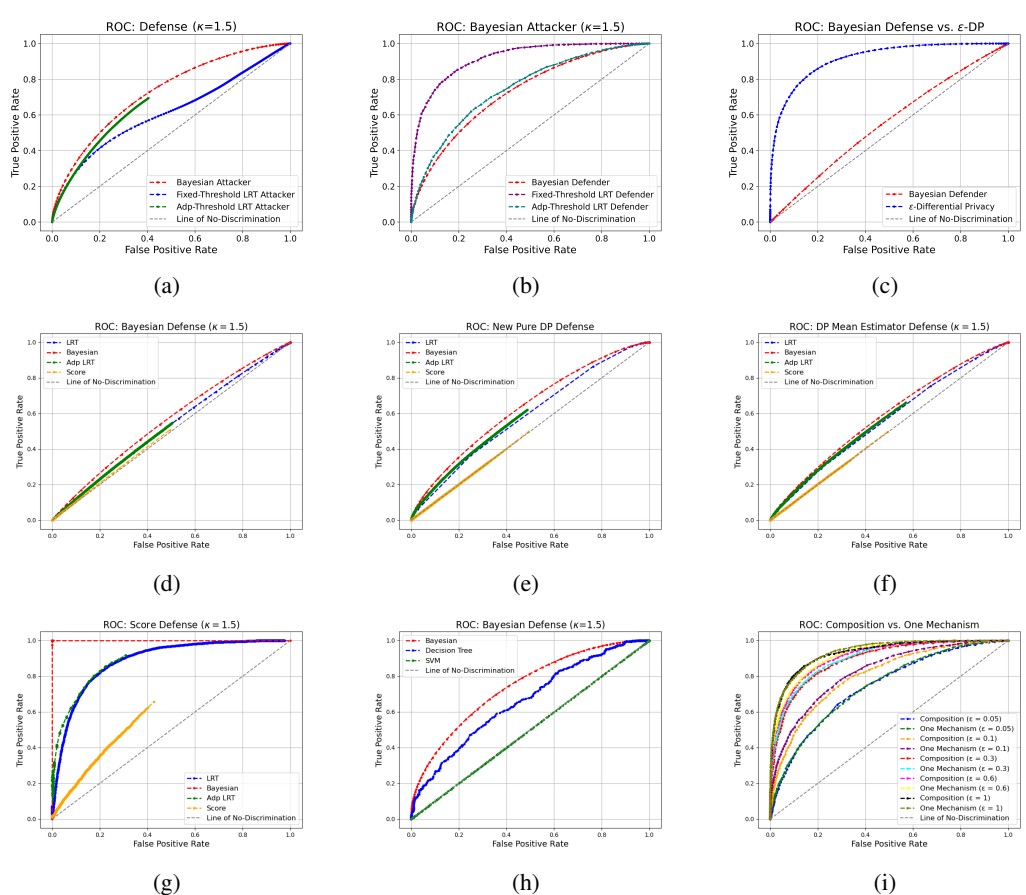

Figure 1: (a)-(c): Genomic dataset with 5000 SNVs (attributes) per individual. (d): Adult dataset. (e)-(f): Genomic dataset with 100 SNVs per individual. (g): Genomic dataset with 4000 SNVs per individual. (h): MNIST dataset. (i) Genomic dataset with 1000 SNVs per individual.

**Datasets and Baselines:** Our experiments use three datasets: the *Adult dataset* (UCI Machine Learning Repository), the *MNIST dataset*, and a *genomic dataset*. Detailed experimental setups and additional results are provided in Appendix P. We compare our Bayesian attacker (inducing the BGP response using $\mathcal{L}_{\text{CEL}}$) with the following baseline attack models: *fixed-threshold* and *adaptive-threshold* attackers (Sankararaman et al., 2009; Shringarpure & Bustamante, 2015; Venkatesaramani et al., 2021; 2023), the *score-based* attacker (Dwork et al., 2015), and *decision-tree* and *support vector machine (SVM)* attackers. The score-based attacker, proposed by Dwork et al. (2015), relaxes the LRT attack from Homer et al. (2008), requiring only that distorted summary statistics approximate the true marginals in $\ell_1$-norm. We also evaluate the BNGP strategy against baseline defenses, including *standard DP*, *new pure DP* (Steinke & Ullman, 2016), the *DP mean estimator* (Cai et al., 2021), and two state-of-the-art (SOTA) genomic defense models (Venkatesaramani et al., 2021; 2023). In the experiments for Figures 1a-1e, 1a, and 1g, the mechanism releases summary statistics of the genomic dataset.

For Figure 1f, the mechanism performs mean estimation under DP (Cai et al., 2021). In the experiments for Figure 1h, the mechanism serves as a classifier for the MNIST dataset. For all experiments, we assume a uniform prior $\theta$ and set the Bayesian attacker's $\sigma = \theta$. We measure the strength of privacy protection using the attacker's ROC curve and its Area Under the Curve (AUC), which quantifies the attacker's ability to distinguish members from non-members.

**Figure 1a** presents the defender's performance against Bayesian, fixed-threshold LRT, and adaptive LRT attackers, using the genomic dataset (see Appendix C for details). The results confirm that the Bayesian attacker outperforms both the fixed-threshold and adaptive LRT attackers. That is, the Bayesian attacker poses the greatest threat among the three attack models. **Figure 1b** illustrates the Bayesian attacker's performance across three scenarios, each with the mechanism protected by a different defense model, using the genomic dataset. The Bayesian defender employs the BNGP strategy, while the fixed-threshold LRT and adaptive LRT defenders adopt privacy strategies that best respond to their respective LRT attackers. The results demonstrate that the Bayesian defender using the BNGP strategy is the most robust defense against the Bayesian attacker among the three defense models.

**Figure 1c** compares the Bayesian attacker's performance under two defenses using the genomic dataset: the Bayesian defender employs the BNGP strategy, while the other uses conventional $\epsilon$-DP. Detailed setup information is in Appendix P.3. The Bayesian defender accounts for heterogeneous privacy-utility trade-offs by assigning weights $\vec{\kappa} = (\kappa_j)_{j \in Q}$ to SNV positions, with $\kappa_j = 0$ for 90% of 5000 SNVs and $\kappa_j = 50$ for the remaining 10%, meaning that the defender only cares about the utility loss for 10% of SNVs. In contrast, the $\epsilon$-DP strategy ignores these preferences but selects $\epsilon$ to match the Bayesian defender's expected utility loss. In the experiment, the utility loss for the BNGP strategy is about 0.0001 and the corresponding $\epsilon = 1.25 \times 10^5$ (see Appendix P.3 for explanation). The results show that, despite equal utility loss, the $\epsilon$-DP defense can incur significantly greater privacy loss under the Bayesian attack, with an AUC of 0.53 against the Bayesian defender and 0.91 against the $\epsilon$-DP defender.

**Figure 1d** presents the performance of four attackers when the mechanism is protected by the Bayesian defense adopting the BNGP strategy, using the Adult dataset. The results show that the Bayesian attacker outperforms the others, with the adaptive LRT slightly surpassing the fixed-threshold LRT, and the score-based attacker performing the weakest. **Figures 1e** and **1f** present the performance of four attackers (Bayesian, fixed-threshold LRT, adaptive LRT, and score-based) when the mechanism is protected by two defense models, using the genomic dataset. In **Figure 1e**, the defender employs the new pure DP defense (Steinke & Ullman, 2016), while in **Figure 1f**, the defender uses DP with peeling (Dwork et al., 2018) to protect the mean estimator (Cai et al., 2021). The results show that the Bayesian attacker consistently achieves the highest performance, followed by the adaptive LRT, which slightly outperforms the fixed-threshold LRT, with the score-based attacker performing the worst.

**Figure 1g** compares the performance of four attackers (Bayesian, fixed-threshold LRT, adaptive LRT, and score-based) when the mechanism is protected by a defender employing the strategy that best responds to the score-based attacker, using the genomic dataset. The results show that the Bayesian attacker significantly outperforms the others, while the fixed-threshold and adaptive LRT attackers perform similarly but lag behind, with the score-based attacker performing the worst.

**Figure 1h** presents the results for a classifier trained on the MNIST dataset. The ROC curve compares the performance of the Bayesian, decision-tree, and SVM attackers when the mechanism is protected by the Bayesian defender employing the BNGP strategy. The results indicate that the Bayesian attacker performs the best, followed by the decision-tree attacker, with the SVM attacker performing the worst.

The experiments in **Figure 1i** empirically evaluate the BNGP strategy with the BGP response satisfying the condition in Proposition 6. "Composition" refers to the combination of five mechanisms, while "One Mechanism" represents a single $\epsilon$-DP mechanism. The single $\epsilon$-DP mechanism serves as a reference to assess whether the composition using the BNGP strategy also satisfies $\epsilon$-DP by comparing the Bayesian attacker's performance. If the composition satisfies $\epsilon$-DP, the Bayesian attacker's performance should closely resemble that in the single mechanism case. The results demonstrate that the BNGP strategy using the BGP response satisfying the condition in Proposition 6 ensures that the composition is approximately $\epsilon$-DP.

## 6 CONCLUSION

This paper introduces a game-theoretic framework for optimal privacy-utility trade-offs, addressing the limitations of differential privacy in balancing privacy and utility. By modeling privacy protection as a Bayesian game between a defender and an attacker, we derive the Bayes-Nash Generative Privacy (BNGP) strategy, which achieves optimal trade-offs tailored to defender preferences. The BNGP strategy avoids intractable sensitivity calculations, supports complex compositions, and remains robust to heterogeneous attacker preferences. Empirical results validate the effectiveness of BNGP in privacy-preserving data sharing and classification, demonstrating its potential as a flexible and practical alternative to existing methods.

## REPRODUCIBILITY STATEMENT

We have taken extensive measures to ensure the reproducibility of our work. All theoretical proofs are included in the appendix for transparency. Appendix P provides a comprehensive description of the experiment setups and additional experiments to facilitate replication. Furthermore, the main source code has been submitted as supplementary material

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

# APPENDIX

## A NOTATIONS

### NOTATIONS FOR SECTION 2

| Symbol | Description |
|---|---|
| $U \in [K]$ | population of $K$ individuals |
| $b \in W$ | membership vector; $W$ is the membership vector space |
| $d = (d_k)_{k \in U}$ | $d_k$ : data point of individual $k$ |
| $B = \{b, d\}$ | dataset |
| $\theta$ | true prior distribution of membership vector |
| $f(\cdot)$ | (data processing) mechanism without privacy protection |
| $x \in \mathcal{X}$ | output of $f$; $\mathcal{X}$ is the output space |
| $\mathcal{A}(\cdot)$ | MIA model |
| $\mathrm{Adv}_k(\mathcal{A})$ | standard membership advantage of $\mathcal{A}(\cdot)$ |

### NOTATIONS FOR SECTION 3

| Symbol | Description |
|---|---|
| $g_D(\cdot)$ | defender's privacy strategy |
| $\mathcal{M}(\cdot; g_D)$ | randomized version of $f$ by $g_D$ |
| $\rho_D : W \mapsto \Delta(\mathcal{X})$ | density function induced by $g_D$ and $f$ |
| $\mathrm{r}(\cdot)$ | $x = \mathrm{r}(\delta)$ captures the relationship between an output sample $x$ and a noise sample $\delta$ |
| $\mathrm{R}(\cdot)$ | clipping processing used in output perturbation to ensure the output is within $\mathcal{X}$ |
| $\sigma$ | attacker's subjective prior |
| $h_A(\cdot)$ | Bayesian attacker's (inference) strategy |
| $\mathcal{A}(\cdot; h_A, \sigma)$ | MIA model given $h_A$ and $\sigma$ |
| $\mathrm{Adv}^\gamma(h_A, g_D)$ | Bayes-weighted membership advantage (BWMA) |
| $\mathrm{Adv}_k(g_D)$ | maximum (standard) MA: $\mathrm{Adv}_k(g_D) \equiv \max_{h_A} \{\mathrm{TPR}(h_A, g_D) - \mathrm{FPR}(h_A, g_D)\}$. |
| $0 < \gamma \leq 1$ | coefficient weights the attacker's preferences over TPR and FPR |
| $\ell_A(\cdot)$ | loss function of the attacker |
| $\mathcal{L}_A^\gamma(g_D, h_A)$ | expected loss function of the attacker given $g_D$ and $h_A$ |
| $\ell_U(\cdot)$ | utility loss function of the defender |
| $\ell_D(\cdot)$ | loss function of the defender |
| $\mathcal{L}_D(g_D, h_A)$ | expected loss function of the defender given $g_D$ and $h_A$ |
| $G$ | non-parameterized neural network generator that represents $g_D$ |
| $H$ | non-parameterized neural network discriminator that represents $h_A$ |
| $G_{\lambda_D}(b, \nu)$ | generator parameterized by $\lambda_D$, where $\nu$ is a uniform random variable |
| $H_{\lambda_A}(x)$ | discriminator parameterized by $\lambda_A$ |
| $\widetilde{\mathcal{L}}_D(G, H)$ | the defender's expected loss given $G$ and $H$ |
| $\widetilde{\mathcal{L}}_D^\gamma(G, H)$ | the attacker's expected loss given $G$ and $H$ |
| $\widehat{\ell}_A$ | binary cross-entropy loss function of the attacker |
| $\mathcal{L}_{\mathrm{CEL}}$ | expected cross-entropy loss function of the attacker |

Note that from Equation (4) onward, the notation of the dataset $B = (b, d)$ is simplified to its membership vector $b$ for clarity.

NOTATIONS FOR SECTION 3.2

| Symbol | Description |
|---|---|
| $\text{Proc} \circ \mathcal{M}(b; G)$ | post-processing of mechanism $\mathcal{M}$ |
| $\text{Proc} \circ G$ | underlying effective randomization device for $\text{Proc} \circ \mathcal{M}(b; G)$ |
| $\mathcal{M}_j(G_j)$ | $j$-th mechanism, where $G_j$ corresponds to $g_D^j$ |
| $\vec{G} = \{G_1, \cdots, G_n\}$ | a profile of $n \geq 1$ generators, with output $\vec{\delta} = (\delta_1, \ldots, \delta_n)$ |
| $\mathcal{M}(\cdot; \vec{G})$ | composition of $\mathcal{M}_1(G_1), \ldots, \mathcal{M}_n(G_n)$ |
| $\vec{\rho}_D$ | joint density function of $\mathcal{M}(b; \vec{G})$ |
| $\vec{r}(\cdot)$ | $\vec{x} = \vec{r}(\vec{\delta})$ captures relationship between $\vec{x}$ and $\vec{\delta}$ |
| $\text{DPH}[\vec{G}; \epsilon]$ | a set of linear conditions for the attacker's discriminator $H$, given by (9) |

NOTATIONS FOR SECTION 4

| Symbol | Description |
|---|---|
| $\texttt{lrs}(d_x, x)$ | log-likelihood ratio statistic defined in Appendix C |
| $\alpha, \alpha_\tau$ | significance level of a hypothesis testing, with threshold $\tau$ |
| $\beta_\tau$ | Type-II error rates given a threshold $\tau$ |
| $L_{\text{Opt-LRT}}^\alpha(g_D)$ | worst-case privacy loss (WCPL) under optimal $\alpha$-LRT attack defined in Appendix C |
| $L_{\text{Adp}}^\alpha(g_D)$ | WCPL under adaptive $\alpha$-LRT attack defined in Appendix C |
| $L_{\text{Fixed}}^\alpha(g_D)$ | WCPL under fixed $\alpha$-LRT attack defined in Appendix C |
| $L_{\text{Bayesian}}^\sigma(g_D)$ | WCPL under BGP response attacker |

# B   THEORETICAL INSIGHTS AND SUPPLEMENTARY INTUITIONS

## B.1   PROPOSITION 1

(**Proposition 1 Restated**). *Suppose $\sigma = \theta$. Then, for any $g_D$, $h_A$, and $0 < \gamma \leq 1$, we have $\mathcal{L}_A^\gamma(g_D, h_A) = -\texttt{Adv}^\gamma(h_A, g_D)$.*

The proof of Proposition 1 is given by Appendix F.

The proof of Proposition 1 begins by reformulating the attacker's loss function $\ell_A(s, b, \gamma)$ to explicitly capture the contributions of true positives ($b_k = 1$) and false positives ($b_k = 0$). The weights $(1 - \gamma)$ and $-\gamma$ highlight the balance between the attacker's trade-offs for these cases, directly linking the loss to the attacker's inference strategy $h_A(s|x)$. The expected loss $\mathcal{L}_A^\gamma(g_D, h_A)$ is then expressed as an integral over the attacker's strategy $h_A(s|x)$, the defender's output distribution $\rho_D(x|b)$, and the prior distribution $\sigma(b)$ of the membership vector. This formulation ties the attacker's loss to the probabilistic structure of the problem. A key simplification occurs when $\sigma = \theta$, aligning the prior distribution with the true membership distribution. Under this assumption, the expected loss is simplified into terms weighted by $(1 - \gamma)$ and $\gamma$, representing probabilities associated with membership inference. Taking the expectation over the defender's output distribution $\rho_D(x|b)$, the proof shows that:

$$\mathcal{L}_A^\gamma(g_D, h_A) = -\texttt{Adv}^\gamma(h_A, g_D).$$

This result establishes that minimizing the attacker's expected loss $\mathcal{L}_A$ is equivalent to maximizing their $\gamma$-weighted membership advantage. Thus, the proof connects the attacker's strategy to membership advantage within the Bayesian game-theoretic framework.

## B.2   PROPOSITION 2

(**Proposition 2 Restated**). *Let $g_D$ and $g_D'$ be two defense strategies, and suppose $\sigma = \theta$. Then, $\texttt{Adv}_k(g_D) \geq \texttt{Adv}_k(g_D')$ for all $k \in U$ iff $\max_{h_A} \texttt{Adv}^{0.5}(h_A, g_D) \geq \max_{h_A} \texttt{Adv}^{0.5}(h_A, g_D')$.*

The proof of Proposition 2 is given by Appendix G.

The proof of Proposition 2 establishes the equivalence between individual membership advantages and the optimal Bayes-weighted membership advantage for two defense strategies, $g_D$ and $g'_D$.

For the *if* direction, the proof assumes $\text{Adv}_k(g_D) \geq \text{Adv}_k(g'_D)$ for all $k \in U$. By applying the prior probabilities of $b_k = 0$ and $b_k = 1$ to both sides of the inequality and summing over all individuals, it follows that the total membership advantage of $g_D$ is greater than or equal to that of $g'_D$, completing the *if* direction.

For the *only if* direction, the proof introduces $\text{Adv}^*(g_D)$, the maximum Bayes-weighted membership advantage, and $\mathcal{L}_A^*(g_D) = \min_{h_A} \mathcal{L}_A(g_D, h_A)$ (i.e., the corresponding minimal attacker loss ). By Proposition 1, $\text{Adv}^*(g_D) \geq \text{Adv}^*(g'_D)$ is equivalent to $\mathcal{L}_A^*(g_D) \leq \mathcal{L}_A^*(g'_D)$. Using the Blackwell informativeness ordering, the proof shows that $g_D$ is at least as informative as $g'_D$, and this ordering is independent of the choice of priors. By considering a uniform prior, the optimal Bayes-weighted membership advantage simplifies to $\text{Adv}^\dagger(g_D) = \frac{1}{2} \sum_{k \in U} \text{Adv}_k(g_D)$. The informativeness ordering ensures no individual $k_0 \in U$ exists such that $\text{Adv}_{k_0}(g_D) > \text{Adv}_{k_0}(g'_D)$. This concludes the proof of the equivalence.

## B.3 THEOREM 1

(**Theorem 1 Restated**). *Let $G^*$ be a BNGP strategy for $\widetilde{\mathcal{L}}_D$ and $\sigma$, and let $H^*$ be a BGP response to $G^*$. Suppose that $\widetilde{\mathcal{L}}_D$ satisfies Assumption 1. Then, for any $G' \in \arg\min_G \widetilde{\mathcal{L}}_D(G, H')$ with $H' \in \arg\min_H \mathcal{L}_A^\gamma(G', H)$ where $0 < \gamma \leq 1$, and for any $\widehat{H}$, we have:*
(i) $\text{TPR}(\widehat{H}, G^*) \leq \text{TPR}(H^*, G^*) \leq \text{TPR}(H', G')$; (ii) $\text{Adv}^{0.5}(\widehat{H}, G^*) \leq \text{Adv}^{0.5}(H^*, G^*) \leq \text{Adv}^{0.5}(H', G')$; (iii) $\widetilde{\mathcal{L}}_D(\widehat{H}, G^*) \leq \widetilde{\mathcal{L}}_D(G^*, H^*) \leq \widetilde{\mathcal{L}}_D(G', H')$.

The proof of Theorem 1 is given by Appendix H.

The proof of Theorem 1 establishes that the Bayes-Nash Generative Privacy (BNGP) strategy $G^*$, when paired with the Bayes-Generative Privacy (BGP) response $H^*$, achieves optimal privacy-utility trade-offs under the given assumptions. The proof builds on the definitions of expected loss functions and their relationships with posterior beliefs, denoted by $\mu_\sigma$. The proof uses two key definitions. First, the function $Z(g_D, \sigma; V)$ is introduced (Equation 12 in Appendix H) to aggregate the expected value of any general function $V(s, b)$ over the posterior belief $\mu_\sigma$, which is induced by the defender's strategy $g_D$ and prior $\sigma$. Second, the function $L(g_D, h_A; V)$ is introduced (Equation 13 in Appendix H) to represent the attacker's expected loss under strategy $h_A$ for the same function $V(s, b)$. When $V(s, b) = \ell_A(s, b; \gamma)$ for a given $0 < \gamma \leq 1$, this loss corresponds to the attacker's expected utility $\mathcal{L}_A^\gamma(g_D, h_A)$.

By Proposition 7, the expected value $Z(g_D, \sigma)$ coincides with the attacker's loss $L^\sigma(g_D)$ when $h_A$ corresponds to the posterior belief $\mu_\sigma$. Furthermore, Proposition 8 ensures that every BGP response $H^*$ matches this posterior belief, guaranteeing that the expected loss $\mathcal{L}_A^\gamma(G, H^*)$ is minimized. This implies that the adversary's membership advantage $\text{Adv}^{0.5}(H^*, G^*)$ is smaller than or equal to that of any other strategy $H'$. The proof then extends this result to the defender's utility by considering $V(s, b) = v(s, b)$. Since the BGP response minimizes the expected loss for this utility function, the corresponding true positive rate (TPR) satisfies $\text{TPR}(G^*, H^*) \leq \text{TPR}(G', H')$, where $G'$ is any other privacy strategy and $H'$ is its corresponding best response.

These results collectively establish the inequalities in parts (i), (ii), and (iii) of the theorem, confirming that the BNGP strategy $G^*$, coupled with the BGP response $H^*$, achieves optimal privacy-utility trade-offs.

## B.4 PROPOSITION 3

(**Proposition 3 Restated**). *Suppose that $G$ has BGP risk $H \in \arg\min_H \mathcal{L}_{\text{CEL}}(G, H)$ and $\widehat{\mathcal{L}}_A^\sigma(G, H)$. Suppose in addition that for any Proc, Proc $\circ G$ has BGP risk $H' \in \arg\min_H \mathcal{L}_{\text{CEL}}(\text{Proc} \circ G, H)$. Then, $\mathcal{L}_{\text{CEL}}(\text{Proc} \circ G, H') \geq \mathcal{L}_{\text{CEL}}(G, H)$.*

The proof of Proposition 3 is given by Appendix I.

Proposition 3 establishes that applying a post-processing function Proc to a defender's privacy strategy $G$ cannot decrease the Bayes Generative Privacy (BGP) risk. This property aligns with the principle that post-processing cannot increase the informativeness of a mechanism.

The proof relies on Blackwell's informativeness ordering. By Theorem 2.10 of (Dong et al., 2021) (see also (Blackwell, 1951)), for any fixed significance level, the minimum false positive rates for inferring each individual's membership status are denoted by $T(G)$ and $T(\text{Proc} \circ G)$ when using $G$ and Proc $\circ G$, respectively. It is shown that $T(\text{Proc} \circ G) \geq T(G)$, meaning that $G$ is at least as informative as Proc $\circ G$. According to Theorem 1 of (de Oliveira, 2018), Blackwell's informativeness ordering implies that the attacker's minimum expected loss for the post-processed mechanism Proc $\circ G$ satisfies $\mathcal{L}_{\text{CEL}}(\text{Proc} \circ G, H') \geq \mathcal{L}_{\text{CEL}}(G, H)$. Therefore, the post-processing property ensures that applying Proc to $G$ does not reduce the attacker's expected loss, confirming the proposition.

### B.5 PROPOSITION 4

(**Proposition 4 Restated**). *Suppose that $\mathcal{M}(\vec{G})$ is a composition of $n$ mechanisms with arbitrary correlation. Then, we have, for $H^* \in \arg\min_H \mathcal{L}_{\text{CEL}}(\vec{G}, H)$, $H_j^* \in \arg\min_{H_j} \mathcal{L}_{\text{CEL}}(G_j, H_j)$ for all $j \in [n]$,*

$$\mathcal{L}_{\text{CEL}}(\vec{G}, \vec{H}^*) = \sum_{j=1}^n \mathcal{L}_{\text{CEL}}(G_j, H_j^*) - \Lambda(\vec{G}, \theta).$$

*If mechanisms are* independent, *then* $\Lambda(\vec{G}, \theta) = -\sum_b \theta(b) \int_{\vec{\mathcal{X}}} \vec{\rho}_D(\vec{x}|b) \cdot \log\left(\sum_{b'} \vec{\rho}_D(\vec{x}|b')\theta(b')\right) d\vec{x}$. *If mechanisms are* correlated, $\Lambda(\vec{G}, \theta) = -\sum_b \theta(b) \int_{\vec{\mathcal{X}}} \vec{\rho}_D(\vec{x}|b) \log\left(\frac{\sum_{b'} \vec{\rho}_D(\vec{x}|b')\theta(b')}{P(\vec{x})}\right) d\vec{x}$, *where* $P(\vec{x}) = \prod_{j=1}^n \sum_{b'} \int_{\vec{\mathcal{X}}_{-j}} \vec{\rho}_D(x_j, \vec{x}_{-j}|b')\theta(b') d\vec{x}_{-j}$.

The proof of Proposition 4 is given by Appendix J.

For independent mechanisms, the total BGP risk decomposes cleanly into the sum of individual mechanism risks. The interaction term $\Lambda(\vec{G}, \theta)$ reflects the joint contribution to the risk but simplifies due to the independence of outputs. This independence ensures that the attacker's best response to each mechanism depends solely on its marginal posterior distribution, making the overall composition straightforward to analyze. For correlated mechanisms, $\Lambda(\vec{G}, \theta)$ explicitly accounts for dependencies among outputs by incorporating joint densities and marginal probabilities. The joint posterior distribution $\mu_\theta(b|\vec{x})$ aligns the attacker's best response with the interdependent outputs of the mechanisms. This dependency modifies the interaction term and ensures that the total BGP risk reflects both individual risks and the additional information provided by the correlation.

### B.6 PROPOSITION 5

(**Proposition 5 Restated**). *Let $\vec{G}^* = \{G_1^*, \ldots, G_n^*\}$. Let $\mathcal{M}(\vec{G}^*)$ be a composition of $n \geq 1$ mechanisms with arbitrary correlation, where each $G_j^*$ is BNGP strategies for some $\widetilde{\mathcal{L}}_D^j$ satisfying Assumption 1. Then, $\mathcal{M}(\vec{G}^*)$ is $(\epsilon, \xi)$-DP for some $\epsilon \geq 0$ and $\xi \in [0, 1]$.*

The proof of Proposition 5 is given by Appendix K.

The proof demonstrates that the composition $\mathcal{M}(\vec{G}^*)$ satisfies $(\epsilon, \xi)$-differential privacy by applying the properties of likelihood-ratio tests and f-DP. The proof uses the properties of likelihood-ratio tests and trade-off functions to establish f-DP guarantees, which are subsequently converted to $(\epsilon, \xi)$-DP guarantees, ensuring that the composition mechanism satisfies differential privacy even under arbitrary correlations.

Using the Neyman-Pearson lemma Neyman & Pearson (1933), the likelihood-ratio test is identified as the Uniformly Most Powerful (UMP) test for distinguishing between the two hypotheses $\mathcal{H}_0^k$ and $\mathcal{H}_1^k$, which correspond to whether an individual's data is included in the dataset. This establishes a fundamental relationship between the test's significance level $\alpha^k$ and the corresponding rejection rule $\phi$. The symmetric trade-off function $f(\alpha^k)$ introduced in Dong et al. (2022) is then used to relate the false positive and false negative rates of this hypothesis test. The function $f(\alpha^k)$ has key properties, such as convexity, continuity, and monotonicity, which make it suitable for capturing the privacy guarantees of the mechanism. By employing results from the f-DP framework, the privacy guarantees of $\mathcal{M}(\vec{G}^*)$ as f-DP are translated into $(\epsilon, \xi)$-DP guarantees. Specifically, the composition satisfies $(\epsilon^k, \xi^k)$-DP for individual components, where $\xi^k$ is a function of $\epsilon^k$. Aggregating these guarantees across all components ensures that $\mathcal{M}(\vec{G}^*)$ satisfies $(\epsilon, \xi)$-DP for some $\epsilon \geq 0$ and $\xi \in [0, 1]$.

### B.7 PROPOSITION 6

(**Proposition 6 Restated**). *For any $\vec{G} = \{G_1, G_2, \ldots, G_n\}$ and $\epsilon \geq 0$, the composition $\mathcal{M}(\vec{G})$ is $\epsilon$-DP iff all the BGP responses $H^* \in \text{DPH}[\vec{G}; \epsilon] \neq \emptyset$.*

The proof of Proposition is given by Appendix L.

This proposition establishes a necessary and sufficient condition for a composition of mechanisms, $\mathcal{M}(\vec{G})$, to satisfy $\epsilon$-differential privacy (DP). The core insight is the equivalence between $\epsilon$-DP of the mechanism and the properties of its best-response discriminators, known as BGP responses. Specifically, $\mathcal{M}(\vec{G})$ is $\epsilon$-DP if and only if all BGP responses satisfy the conditions defined by $\text{DPH}[\vec{G}; \epsilon]$. This bridges the classical notion of $\epsilon$-DP with the Bayesian framework by characterizing privacy guarantees in terms of adversarial inference strategies. This result demonstrates the consistency between classical differential privacy and the Bayesian game-theoretic approach, showing that $\epsilon$-DP can be fully characterized through BGP responses. This provides a powerful perspective on privacy guarantees, uniting two complementary frameworks while maintaining rigorous mathematical consistency.

The proof leverages the posterior distribution $\mu_\theta(b|\vec{x})$ induced by $\vec{G}$ and the prior $\theta$. If $\arg\min_H \mathcal{L}_{\text{CEL}}(\vec{G}, H) \cap$ $\text{DPH}[\vec{G}; \epsilon] \neq \emptyset$, this posterior distribution satisfies the conditions for $\epsilon$-DP as specified by $\text{DPH}[\vec{G}; \epsilon]$. By the necessary and sufficient conditions established in (Dwork et al., 2006), this implies that $\vec{G}$ is $\epsilon$-DP. Conversely, if $\vec{G}$ is $\epsilon$-DP, the posterior distribution must also meet these conditions, ensuring that all BGP responses belong to $\text{DPH}[\vec{G}; \epsilon]$. The extension of Proposition 8 in Appendix H guarantees that the optimal BGP response aligns with the posterior induced by $\vec{G}$, which completes the equivalence.

## B.8 Theorem 2

(**Theorem 2 Restated**). *Fix any $g_D$ and $\alpha$. If $\sigma \in \Delta(W)$ is an* aligned prior, *then:*

$$L^\sigma_{\text{Bayesian}}(g_D) \geq L^\alpha_{\text{Opt-LRT}}(g_D) \geq L^\alpha_{\text{Adp}}(g_D) \geq L^\alpha_{\text{Fixed}}(g_D).$$

*If $\sigma$ is* strictly aligned, *then $L^\sigma_{\text{Bay}}(g_D) > L^\alpha_{\text{Opt-LRT}}(g_D)$.*

The proof of Theorem 2 is given by Appendix M.

Theorem 2 establishes a hierarchy of worst-case privacy losses incurred by the defender under different attacker models: Bayesian, optimal $\alpha$-LRT, adaptive $\alpha$-LRT, and fixed-threshold $\alpha$-LRT. It asserts that the Bayesian attacker leads to the highest loss when the prior $\sigma$ is aligned, and the Bayesian loss strictly exceeds the optimal $\alpha$-LRT loss if $\sigma$ is strictly aligned.

The proof hinges on several key insights. First, Lemma 4 demonstrates that when $\sigma$ is aligned, a Bayesian attacker using the posterior belief $\mu_\sigma$ as a best-response strategy minimizes its loss. Lemma 5 further shows that the Bayesian attacker cannot perform worse than the optimal $\alpha$-LRT under the same defense strategy $g_D$. This is achieved by leveraging the relationship between the Bayesian attacker's posterior and the likelihood ratio statistic of the $\alpha$-LRT, ensuring that the Bayesian strategy captures a broader range of risks. For non-informative priors, the proof verifies that the Bayesian attacker still outperforms $\alpha$-LRTs by demonstrating that any $\alpha$-LRT strategy can be virtually represented as a special case of the Bayesian framework with uniform prior. The hierarchy of losses for adaptive and fixed-threshold $\alpha$-LRTs follows directly from the Neyman-Pearson lemma and existing results in the literature (Venkatesaramani et al., 2021; 2023). These results highlight the robustness of the Bayesian approach in capturing privacy risks across various attacker models and priors.

## C Existence Frequentist Attack Models

**Likelihood Ratio Test Attacks** MIAs targeting genomic summary data releases are often framed as hypothesis testing problems (Sankararaman et al., 2009; Shringarpure & Bustamante, 2015; Raisaro et al., 2017; Venkatesaramani et al., 2021; 2023), where for each individual $k \in U$, the attacker tests $H_0^k : b_k = 1$ (i.e., the individual $k$ is in the dataset) versus $H_1^k : b_k = 0$ (i.e., the individual $k$ is not). Additionally, $\bar{p}_j$ denotes the frequency of the alternate allele at the $j$-th SNV in a reference population that is not included in the dataset $B$.

First, assume $\delta = 0$. The attacker is assumed to have external knowledge of the genomic data for individuals $[K]$, in the form of $\bar{p} = (\bar{p}_j)_{j \in Q}$ and $d = (d_{kj})_{k \in [K], j \in Q}$. The *log-likelihood ratio statistic* (LRS) for each individual $k$ is given by (Sankararaman et al., 2009):

$$\text{lrs}(d_k, x) = \sum_{j \in Q} \left( d_{kj} \log \frac{\bar{p}_j}{x_j} + (1 - d_{kj}) \log \frac{1 - \bar{p}_j}{1 - x_j} \right).$$

An *LRT attacker* performs MIA by testing $H_0^k$ against $H_1^k$ using $\text{lrs}(d_k, x)$ for each $k \in [K]$. The null hypothesis $H_0^k$ is rejected in favor of $H_1^k$ if $\text{lrs}(d_k, x) \leq \tau$ for a threshold $\tau$, and $H_0^k$ is accepted if $\text{lrs}(d_k, x) > \tau$.

Let $P_0^k(\cdot) \equiv \Pr(\cdot|H_0^k)$ and $P_1^k(\cdot) \equiv \Pr(\cdot|H_1^k)$ denote the probability distributions under $H_0$ and $H_1$, respectively.

**Definition 6** (($\alpha_\tau, \beta_\tau$)-LRT Attack). *An attacker performs ($\alpha_\tau, \beta_\tau$)-LRT Attack if $P_0^k(\text{lrs}(d_k, x) \leq \tau) = \alpha_\tau$ and $1 - P_1^k(\text{lrs}(d_k, x) \leq \tau) = \beta_\tau$, for all $k \in U$, where $\alpha_\tau$ is the* significance level *and $1 - \beta_\tau$ is the* power *of the test with the* threshold $\tau$.

Define the trade-off function (Dong et al., 2021), $T[P_0^k, P_1^k](\alpha) \equiv \inf_\tau \{\beta_\tau : \alpha_\tau \leq \alpha\}$. By Neyman-Pearson lemma (Neyman & Pearson, 1933), the LRT test is the uniformly most powerful (UMP) test for a given significance level. Specifically, for a given $\alpha_\tau$, there exists a threshold $\tau^*$ such that no other test with $\alpha_\tau \leq \alpha_{\tau^*}$ can achieve a strictly smaller $\beta_\tau < \beta_{\tau^*}$. Hence, $T[P_0^k, P_1^k](\alpha_{\tau^*}) = \beta_{\tau^*}$, for all $k \in U$. We refer to an $\alpha$-LRT as a UMP $(\alpha, \beta)$-LRT and will interchangeably add or omit the corresponding threshold notation as needed.

From the vNM defender's perspective, the expected privacy losses under an $\alpha$-LRT attack, without and with defense $g_D$, respectively, are given by

$$L^o(\tau^o, \alpha) \equiv \mathbb{E}\left[v(\tilde{s}, \tilde{b})\Big|\alpha\right] = \sum_k P_1^k\left[y_k(f(b, z), \tau^o) = 1\right]\theta(b_k = 1) = \sum_k (1 - \beta_{\tau^o})\theta(b_k = 1),$$

$$L(g_D, \tau^o, \alpha) \equiv \mathbb{E}\left[v(\tilde{s}, \tilde{b})\Big|g_D, \tau^o, \alpha\right] = \sum_k P_1^k\left[y_k(r, \tau^o) = 1|g_D\right]\theta(b_k = 1),$$

where $y_k(x, \tau^o) \equiv \mathbf{1}\{\mathtt{lrs}(d_k, x) \geq \tau^o\}$ is the indicator function for the likelihood ratio statistic, and $P_1^k[y_k(r, \tau^o) = 1|g_D] \equiv \int_r \mathbf{1}\{y_k(r, \tau^o) = 1\}\rho_D(r|b)dr$. Here, $\tau^o$ is the threshold associated with the $\alpha$-LRT.

**Fixed-Threshold LRT Attack (Sankararaman et al., 2009; Shringarpure & Bustamante, 2015; Venkatesaramani et al., 2021; 2023)** A *fixed(-threhsold)* LRT attacker performs MIA without accounting for any privacy defense strategies. Such an attacker selects a fixed threshold $\tau^o$ that balances Type-I and Type-II errors, resulting in a UMP $\alpha$-LRT test in the absence of defense. This approximation can be achieved by simulating Beacons on publicly available datasets or synthesized data using alternate allele frequencies (AAFs) Venkatesaramani et al. (2023).

Given a fixed threshold $\tau^o$, let

$$L_{\text{Fixed}}^\alpha(g_D) \equiv L(g_D, \tau^o, \alpha_{\tau^o})$$

The defender's optimal strategy against the naive $\alpha_{\tau^o}$-LRT attack is given by solving:

$$\min_{g_D} L_{\text{Fixed}}^\alpha(g_D) + \kappa\mathbb{E}\left[\ell_U(\|\delta\|_{\mathbf{p}})\|\big|g_D, \tau^o, \alpha_{\tau^o}\right], \qquad (\texttt{FixedLRT})$$

where $\mathbb{E}\left[\ell_U(\|\delta\|_{\mathbf{p}})\|\big|g_D, \tau^o, \alpha_{\tau^o}\right]$ is induced expected utility loss.

Let $\beta^k(\tau, g_D, \alpha) \equiv 1 - P_1^k[y_k(r, \tau) = 1|g_D]$ denote the actual Type-II error under the defense strategy $g_D$ for the naive $\alpha$-LRT attack. The defender can reduce privacy loss by choosing $g_D$ to increase $\beta^k(\tau, g_D, \alpha)$ for all $k \in U$. For the defense strategy $g_D^\dagger$ that solves (`FixedLRT`) to be effective in reducing privacy loss, it must be implemented in a *stealthy* manner.

**Adaptive-Threshold LRT Attack (Venkatesaramani et al., 2021; 2023)** In an *adaptive-threshold LRT* attack, the attacker is aware of the defense strategy and attempts to distinguish individuals in $U$ from those in the reference population $\bar{U}$ (individuals not in $U$). Let $\bar{U}^{(N)} \subset \bar{U}$ represent the set of $N$ individuals in $\bar{U}$ with the lowest LRS values. The *adaptive threshold* is defined as $\tau^{(N)}(x) = \frac{1}{N}\sum_{i \in \bar{U}^{(N)}} \mathtt{lrs}(d_i, x)$. The null hypothesis $H_0$ is rejected if $\mathtt{lrs}(d_k, x) \leq \tau^{(N)}(x)$.

The defender's problem is then:

$$\min_{g_D} L_{\text{Adp}}^{\alpha_{\tau^{(N)}(x)}}(g_D) + \mathbb{E}\left[\ell_U(\|\delta\|_{\mathbf{p}})\|\Big|g_D, \tau^{(N)}(x), \alpha_{\tau^{(N)}(x)}\right], \qquad (\texttt{AdaptLRT})$$

where $\alpha_{\tau^{(N)}(x)}$ is the Type-I error associated with the adaptive threshold $\tau^{(N)}(x)$, and

$$L_{\text{Adp}}^{\alpha_{\tau^{(N)}(x)}}(g_D) \equiv L(g_D, \tau^{(N)}(x), \alpha_{\tau^{(N)}(x)}).$$

The WCPL $L_{\text{Adp}}^{\alpha_{\tau^{(N)}(x)}}(g_D)$ in Section 4 has $\mathbb{E}[\alpha_{\tau^{(N)}(\tilde{r})}] = \alpha$.

**Optimal LRT Attack** Let $P_0^k(g_D) = P_0^k[\cdot|g_D]$ and $P_1^k(g_D) = P_1^k[\cdot|g_D]$ denote the probability distributions under $H_0^k$ and $H_1^k$, respectively, in the presence of defense $g_D$. The *worst-case privacy loss* (WCPL) for the defender occurs when the attacker's hypothesis test achieves $\beta^k(\tau^*, g_D, \alpha) = T[P_0^k(g_D), P_1^k(g_D)](\alpha)$ for some threshold $\tau^*$, corresponding to a UMP test under $g_D$. We refer to these as *optimal $\alpha$-LRT attacks*.

The defender's optimal strategy against such attacks solves the following problem:

$$\min_{g_D} L_{\text{Opt-LRT}}^\alpha(g_D) + \kappa\mathbb{E}\left[\ell_U(\|\delta\|_{\mathbf{p}})\|\big|g_D, \tau^*, \alpha\right], \qquad (\texttt{OptLRT})$$
$$\text{s.t.} \quad \beta^k(\tau^*, g_D, \alpha) = T[P_0^k(g_D), P_1^k(g_D)](\alpha),$$

where

$$L_{\text{Opt-LRT}}^\alpha(g_D) \equiv L(g_D, \tau^*, \alpha).$$

By the Neyman-Pearson lemma, the $\alpha$-LRT with likelihood ratio statistics $\mathtt{lrs}(d_k, r; g_D) \equiv \sum_{j \in Q} \frac{\rho_D(r|b_k=0, b_{-k})}{\rho_D(r|b_k=1, b_{-k})}$ for all $k \in U$ is optimal, attaining $\beta^k(\tau^*, g_D, \alpha) = T[P_0^k(g_D), P_1^k(g_D)](\alpha)$.

Furthermore, the defense $g_D$ obtained by solving (`OptLRT`) is robust against adaptive-threshold LRT attacks.

## D  GAUSSIAN DEFENSE STRATEGIES

In this section, we consider $g_D$ is a Gaussian mechanism and study the comparison between the $\sigma$-Bayesian attack and the $\alpha$-LRT attack when $\sigma$ is an arbitrary subjective prior.

Let $L^\sigma(g_D) \equiv \max_{h_A} L(g_D, h_A)$, where

$$L(g_D, h_A) \equiv \sum_{b,s} \int_x v(s,b) h_A(s|x) \rho_D(x|b) dx \theta(b),$$

Define $g_D(\delta|b) = \prod_{j \in Q} g_D^j(\delta_j|b)$, where $g_D^j(\cdot|b)$ is the density function of a Gaussian distribution $\mathcal{N}(\mathtt{M}_b^j, \mathtt{V}^j)$ with mean $\mathtt{M}_b^j$ and variance $\mathtt{V}^j$ for each $b \in U$ and $j \in Q$. Let $y = x + \delta = (x_j + \delta_j)_{j \in Q} \in \mathcal{Y}$, where $y_j = x_j + \delta_j \in \mathcal{Y}_j$. The resulting conditional probability distribution is denoted by $\rho_D(\cdot|b) \in \Delta(\mathcal{Y})$. Let $b_{[0]}^k$ and $b_{[1]}^k$ represent two *adjacent* membership vectors that differ only in individual $k$'s value, where $b_k = 0$ in $b_{[0]}^k$ and $b_k = 1$ in $b_{[1]}^k$. The maximum conditional probability of $s_k = 0$ given $b_k = 1$ is defined as

$$\mu_{0|1}^\sigma[|Q|] \equiv \max_{k \in U} \sum_{s_{-k}} \int_y \mu_\sigma(s_k = 0|y) \rho_D(y|b_{[1]}^k) \, dy,$$

where the posterior belief $\mu_\sigma$ is induced by $g_D$ and $\sigma$. For a Type-I error rate $\widehat{\alpha}$, let $\widehat{\beta}$ represent the minimum Type-II error rate achievable.

**Lemma 1.** *Define* $\mathcal{F}(\alpha, \beta) \equiv \frac{(z_\alpha + z_\beta)^2 \overline{\mathtt{V}}}{4\overline{\mathtt{M}}^2}$, *where* $z_a$ *is the* $100(1 - a)$-*th percentile of the standard normal distribution,* $\overline{\mathtt{M}} = \frac{1}{2} \sum_{j \in Q} \widehat{\mathtt{M}}_j^2$, *and* $\overline{\mathtt{V}} = \sum_{j \in Q} \widehat{\mathtt{M}}_j^2$. *Then the following holds:*

(i) $\mathcal{F}(\widehat{\alpha}, \widehat{\beta}) = |Q|$. (ii) *For a fixed* $\widehat{\alpha}$, *as* $|Q|$ *increases (resp. decreases),* $\widehat{\beta}$ *decreases (resp. increases).*

**Theorem 3.** *Let* $g_D$ *be a Gaussian mechanism with each* $g_D^j(\cdot|b) \in \Delta(\mathcal{Y}_j)$ *following* $\mathcal{N}(\mathtt{M}_b^j, \mathtt{V}^j)$ *for any* $b \in W$. *Suppose* $\mathtt{V}^j = \left( \frac{|Q|}{K^\dagger \widehat{\mathtt{M}}_j} \right)^2$ *for all* $j \in Q$, *where* $1 \leq K^\dagger \leq K$ *is the minimum number of individuals involved in B. Additionally, assume* $\max_{b,b'} |\mathtt{M}_b^j - \mathtt{M}_{b'}^j| \leq \frac{|Q|}{K^\dagger}$, *where the maximum is taken over all adjacent membership vectors. Then, for any Q with* $|Q| \geq 1$ *and any subjective prior* $\sigma$, *if* $\mathcal{F}(\alpha, \mu_{0|1}^\sigma[|Q|]) \geq |Q|$, *it holds that* $L_{\text{Opt-LRT}}^\alpha(g_D) \leq L^\sigma(g_D)$; *if* $\mathcal{F}(\alpha, \mu_{0|1}^\sigma[|Q|]) \leq |Q|$, *it holds that* $L_{\text{Opt-LRT}}^\alpha(g_D) \geq L^\sigma(g_D)$.

Theorem 3 provides conditions under which the $\sigma$-Bayesian attack outperforms or underperforms the $\alpha$-LRT attack in Gaussian mechanisms, even when $\sigma$ is an arbitrary subjective prior independent of the true prior $q$. For any $\alpha$, let $1 - \beta_{|Q|}^\alpha$ denote the power of the $\alpha$-LRT attack, and define $m^\alpha = \mathcal{F}(\alpha, \mu_{0|1}^\sigma[|Q|])$ as the number of SNVs used in the summary statistics such that $1 - \beta_{|Q|}^\alpha = 1 - \mu_{0|1}^\sigma[|Q|]$, i.e., the power of the $\alpha$-LRT matches the worst-case true positive rate (TPR) of the $\sigma$-Bayesian attack. By Lemma 1, as $m$ increases, $\beta^\alpha$ decreases. When $m^\alpha \geq |Q|$, the actual power $1 - \beta_{|Q|}^\alpha \leq 1 - \mu_{0|1}^\sigma[|Q|]$. Thus, by Proposition 7 in Appendix M, the lowest TPR achievable by the $\sigma$-Bayesian attacker exceeds the best power of the $\alpha$-LRT. Consequently, $L_{\text{Opt-LRT}}^\alpha(g_D) \leq L^\sigma(g_D)$. Similarly, when $\mathcal{F}(\alpha, \mu_{0|1}^\sigma[|Q|]) \leq |Q|$, the actual $\beta_{|Q|}^\alpha \leq \mu_{0|1}^\sigma[|Q|]$, thus we have $L_{\text{Opt-LRT}}^\alpha(g_D) \geq L^\sigma(g_D)$.

Based on the sensitivity of $f$ (see the proof at Appendix O for details), Theorem 3 considers the worst-case bound of the powers of the LRT attack when the attacker knows the membership of every individual in the dataset except for a single individual. This bound is evaluated over all possible input membership vectors. Notably, the comparison in Theorem 3 is independent of the true prior distributions $q = (q_k)_{k \in U}$ of the membership vectors and does not rely on specific true membership vectors forming the Beacon dataset.

When $\mathcal{F}(\alpha, \mu_{0|1}^\sigma[m]) < m$, the lowest true positive rate (TPR) of the $\sigma$-Bayesian attacker is strictly smaller than the best power of the $\alpha$-LRT attacker. However, this does not guarantee that every TPR of the $\sigma$-Bayesian attacker is smaller than every power of the $\alpha$-LRT attacker across different Beacon datasets. Therefore, $\mathcal{F}(\alpha, \mu_{0|1}^\sigma[m]) < m$ generally cannot imply that $L_{\text{Opt-LRT}}^\alpha(g_D) > L^\sigma(g_D)$. Moreover, the condition $\mathcal{F}(\alpha, \mu_{0|1}^\sigma[m]) \geq m$ is not necessary. That is, $L_{\text{Opt-LRT}}^\alpha(g_D) \leq L^\sigma(g_D)$ does not imply $\mathcal{F}(\alpha, \mu_{0|1}^\sigma[m]) \geq m$ for any arbitrary subjective prior $\sigma$. We can also conclude that the sufficient condition in Theorem 3 is not applied only to aligned subjective priors. The following corollary directly follows Theorem 3.

**Corollary 1.** *Given a Gaussian mechanism* $g_D$ *with Q, if the number of SNVs of the Beacon dataset satisfies* $|Q| \leq \mathcal{F}(\alpha, \mu_{0|1}^\sigma[|Q|])$, *then the mechanism* $g_D$ *that is optimal to the* $\sigma$-*Bayesian attacks with any arbitrary* $\sigma$ *is guaranteed to be robust to any optimal* $\alpha$-*LRT attacks.*

In this section, we relax Theorem 3 and study the comparison between the Bayesian attacks with arbitrary subjective priors and the optimal LRT attacks without considering the worst-case bound of the powers of the LRT attacks. Suppose in addition that the number of individuals involved in the Beacon dataset is fixed to

be $0 < n < K$. For ease of exposition, we consider the noises added to all SNVs to be iid. Consider a Gaussian mechanism $g_D(\delta|b) = \prod_{j \in Q} g_D^j(\delta_j|b)$, where each $g_D^j(\cdot|b)$ is the density function of $\mathcal{N}(\mathtt{M}_b, \mathtt{V})$. Let two adjacent membership vectors $b_0^{[k]}$ and $b_1^{[k]}$ differing in individual $k$'s $b_k$, where $b_0^{[k]}$ has $b_k = 0$ and $b_1^{[k]}$ has $b_k = 1$. Define two hypotheses: $H_0^{[k]}$: the true membership is $b_0^{[k]}$ vs. $H_1^{[k]}$: the true membership is $b_1^{[k]}$. For any $k \in U$, it is straightforward to see that each $\tilde{y}_j = \tilde{x}_j + \tilde{\delta}_j$ is a Gaussian random variable. That is, $\tilde{y}_j \sim \mathcal{N}(\mathtt{M}_0 + x_j^0, \mathtt{V})$ under $H_0^{[k]}$ and $\tilde{y}_j \sim \mathcal{N}(\mathtt{M}_1 + x_j^1, \mathtt{V})$ under $H_1^{[k]}$, where $\mathtt{M}_i = \mathtt{M}_{b_i^{[k]}}$ and $x_j^i = f(b_i^{[k]}, d_i)$ is the unperturbed summary statistics given $b_i^{[k]}$ with $d_i$, for $i \in \{0, 1\}$. Then, given any $(b_0^{[k]}, b_1^{[k]})$, the power of the optimal $\alpha$-LRT performed upon the observation $y_j$ for all $j \in Q$ can be obtained as

$$T(\mathcal{N}(\mathtt{M}_0 + x_j^0, \mathtt{V}), \mathcal{N}(\mathtt{M}_1 + x_j^1, \mathtt{V}))(\alpha) = \Phi\left(\Phi^{-1}(1 - \alpha) - \frac{|\mathtt{M}_1 - \mathtt{M}_0 + x_j^1 - x_j^0|}{\sqrt{\mathtt{V}}}\right),$$

where $\Phi$ is the cumulative distribution function (CDF) of the standard normal distribution.

Under the assumption of linkage equilibrium (i.e., each SNV is independent of the others), the power of the optimal $\alpha$-LRT performed upon $y = (y_j)_j$ can be obtained by the tensor product of $|Q|$ trade-off functions Dong et al. (2021). In particular, the power can be represented by

$$T\left(\times_{j \in Q}\mathcal{N}(\mathtt{M}_0 + x_j^0, \mathtt{V}), \times_{j \in Q}\mathcal{N}(\mathtt{M}_1 + x_j^1, \mathtt{V})\right)(\alpha) = T\left(\mathtt{N}_0^{[k]}, \mathtt{N}_1^{[k]}\right)(\alpha),$$

where $\mathtt{N}_0^{[k]} = \mathcal{N}(\mathtt{M}_0 + x_1^0, \ldots, \mathtt{M}_0 + x_{|Q|}^0, \Sigma(\mathtt{V}))$ and $\mathtt{N}_1^{[k]} = \mathcal{N}(\mathtt{M}_1 + x_1^1, \ldots, \mathtt{M}_0 + x_{|Q|}^1, \Sigma(\mathtt{V}))$, in which $\Sigma(\mathtt{V})$ is a $|Q| \times |Q|$ diagonal matrix where each principal diagonal element is $\mathtt{V}$. The Mahalanobis distance for the joint distributions is

$$d_{\Sigma(V)}\left((\mathtt{M}_0 + x_1^0, \ldots, \mathtt{M}_0 + x_{|Q|}^0), (\mathtt{M}_1 + x_1^1, \ldots, \mathtt{M}_1 + x_{|Q|}^1)\right) = \sqrt{\sum_{j \in Q} \frac{\left(\mathtt{M}_1 - \mathtt{M}_0 + x_j^1 - x_j^0\right)^2}{\mathtt{V}}}.$$

Therefore, we have

$$T\left(\mathtt{N}_0^{[k]}, \mathtt{N}_1^{[k]}\right)(\alpha) = \Phi\left(\Phi^{-1}(1 - \alpha) - \sqrt{\sum_{j \in Q} \frac{\left(\mathtt{M}_1 - \mathtt{M}_0 + x_j^1 - x_j^0\right)^2}{\mathtt{V}}}\right)$$

$$= T\left(\mathcal{N}(0, 1), \mathcal{N}(\mathtt{M}_{eq}[b_0^{[k]}, b_1^{[k]}], 1)\right)(\alpha),$$

where $\mathtt{M}_{eq}[b_0^{[k]}, b_1^{[k]}] = \sqrt{\sum_{j \in Q} \frac{\left(\mathtt{M}_1 - \mathtt{M}_0 + x_j^1 - x_j^0\right)^2}{\mathtt{V}}}$, in which we show $[b_0^{[k]}, b_1^{[k]}]$ to indicate that the trade-off function is based on $b_0^{[k]}$ and $b_1^{[k]}$.

Let $b_0^{[k]} = (b_k = 0, \hat{b}_{-k})$ and $b_1^{[k]} = (b_k = 1, \hat{b}_{-k})$. Define

$$\beta(\alpha, q) \equiv \sum_{b_k, \hat{b}_{-k}} T\left(\mathcal{N}(0, 1), \mathcal{N}(\mathtt{M}_{eq}[b_0^{[k]}, b_1^{[k]}], 1)\right)(\alpha)q_k(b_k)q_{-k}(\hat{b}_{-k}),$$

and

$$\mu_{0|1}(\sigma, q) \equiv \sum_{b_k, \hat{b}_{-k}} \sum_{s_{-k}} \int_y \mu_\sigma(s_k = 0|y)\rho_D(y|b_k = 1, \hat{b}_{-k})dyq_k(b_k)q_{-k}(\hat{b}_{-k}).$$

In addition, define

$$\Delta(\alpha, \sigma, q) \equiv \mu_{0|1}(\sigma, q) - \beta(\alpha, q).$$

The following corollary is straightforward.

**Corollary 2.** *Let $g_D(\delta|b) = \prod_{j \in Q} g_D^j(\delta_j|b)$ be a Gaussian mechanism, where each $g_D^j(\cdot|b)$ is the density function of $\mathcal{N}(\mathtt{M}_b, \mathtt{V})$. Then, $L_{\text{Opt-LRT}}^\alpha(g_D) \leq L^\sigma(g_D)$ if and only if $\Delta(\alpha, \sigma, q) \geq 0$.*

Corollary 2 represents shows a condition for $L_{\text{Opt-LRT}}^\alpha(g_D) \leq L^\sigma(g_D)$ when the Bayesian attacker's subjective prior $\sigma$ is arbitrary. Here, $1 - \beta(\alpha, q)$ is the expected power of the $\alpha$-LRT attacker perceived by the vNM defender, while $1 - \mu_{0|1}$ is the expected posterior beliefs of $\{s_k = 1\}_{k \in U}$. Thus, $\Delta(\alpha, \sigma, q) \geq 0$ implies that the expected accuracy of inferring $\{s_k = 1\}$ using the posterior beliefs is higher than the expected power of the $\sigma$-LRT. By Proposition 7, we have that the Bayesian strategy that mirrors the posterior belief leads to the WCPL. Therefore, given any $\rho_D$ and the true prior $q$, $\Delta(\alpha, \sigma, q) \geq 0$ is equivalent to $L_{\text{Opt-LRT}}^\alpha(g_D) \leq L^\sigma(g_D)$. This condition is independent of the sensitivity of $f$ but depends on $g_D$ and the true prior $q$.

### D.1 LRT vNM Defender

We use $g_N$, $g_{Adp}$, and $g_{Opt}$ to denote the typical solutions to (FixedLRT), (AdaptLRT) and (OptLRT), respectively. Suppose that all $g_N$, $g_{Adp}$, and $g_{Opt}$ are Gaussian mechanisms. We refer to the defender using $g_N$, $g_{Adp}$, and $g_{Opt}$, respectively, as the naive, adaptive, and optimal *LRT vNM defender*. Then, the WCPL is captured by the power of the UMP test given a significant level $\alpha$. Due to the Neyman-Pearson lemma, the WCPL is the power or the TPR of the optimal $\alpha$-LRT, $1 - T[P_0^k(g_D), P_1^k(g_D)](\alpha)$.

**Corollary 3.** *Fix any $g_D$ and $\alpha$. Let $\mathtt{TPR}(g_D, \sigma)$ denote the maximum TPR can be obtained by a $\sigma$-Bayesian attacker under $g_D$. Suppose that $g_D$ is chosen such that the WCPL is $1 - T[P_0^k(g_D), P_1^k(g_D)](\alpha)$. Then, the following hold.*

> *(i) If $\sigma$ is an informative or non-informative prior, then $\mathtt{TPR}(g_D, \sigma) \geq 1 - T[P_0^k(g_D), P_1^k(g_D)](\alpha)$.*

> *(ii) Suppose that $g_D$ is Gaussian as described in Theorem 3. If $\mathcal{F}\left(\alpha, \mu_{0|1}^\sigma[m]\right) \geq m$, then $\mathtt{TPR}(g_D, \sigma) \geq 1 - T[P_0^k(g_D), P_1^k(g_D)](\alpha)$. If $\mathcal{F}\left(\alpha, \mu_{0|1}^\sigma[m]\right) < m$, then $\mathtt{TPR}(g_D, \sigma) < 1 - T[P_0^k(g_D), P_1^k(g_D)](\alpha)$.*

Part *(i)* of Corollary 3 follows Theorem 2. In particular, from Theorem 2 we have $L^\sigma(g_D) \geq L(g_D, \tau^*, \alpha)$ for aligned subjective priors. Hence, $\mathtt{TPR}(g_D, \sigma) \geq 1 - T[P_0^k(g_D), P_1^k(g_D)](\alpha)$. Part *(ii)* of Corollary 3 follows Theorem 3. If $\mathcal{F}\left(\alpha, \mu_{0|1}^\sigma[m]\right) \geq m$, Theorem 3 implies that $L(g_D, \tau^*, \alpha) \leq L^\sigma(g_D)$, which gives $\mathtt{TPR}(g_D, \sigma) \geq 1 - T[P_0^k(g_D), P_1^k(g_D)](\alpha)$. If $\mathcal{F}\left(\alpha, \mu_{0|1}^\sigma[m]\right) < m$, then $L_{\text{Opt-LRT}}^\alpha(g_D) > L^\sigma(g_D)$, which implies $\mathtt{TPR}(g_D, \sigma) < 1 - T[P_0^k(g_D), P_1^k(g_D)](\alpha)$.

## E DIFFERENTIAL PRIVACY

**Standard Differential Privacy**   Differential privacy Dwork et al. (2006); Dwork (2006) is a widely used data privacy preservation technique based on probabilistic distinguishability. Formally, we say a randomized mechanism $F$ is $(\epsilon, \varrho)$-differentially private if for any two adjacent dataset $\mathtt{D}$ and $\mathtt{D}'$ differing in only one entry if holds that

$$\mathrm{P}\left(F((\mathtt{D}')) \in \mathcal{F}\right) \leq e^\epsilon \mathrm{P}\left(F(\mathtt{D}') \in \mathcal{F}\right) + \varrho$$

for any possible subset $\mathcal{F}$ of the image of the mechanism $F$. The parameter $\epsilon$ is usually referred to as the *privacy budget*, which is small but non-negligible. $(\epsilon, 0)$-DP or $\epsilon$-DP is known as *pure differential privacy*, while with a non-zero $\varrho > 0$, $(\epsilon, \varrho)$-DP is viewed as *approximate differential privacy*.

**Sensitivity**   Define the *sensitivity* of $f$ by

$$\mathtt{sens}(f) \equiv \max_{b, b'} |f(b, d) - f(b', d')|,$$

where the maximum is over all adjacent datasets $(b, d)$ and $(b', d')$ where $b$ and $b'$ differs only in a single individual with $d$ and $d'$ as the corresponding SNVs, respectively. For a given SNV in a dataset with $B \subseteq U$, $d_{kj}$ is either 0 or 1. Thus, the maximum possible difference between the averages over the columns that differ in one entry is $\frac{1}{|B|}$. Let $1 \leq K^\dagger \leq K$ be the minimum number of individuals involved in the Beacon dataset. Hence, $\mathtt{sens}(f) = \frac{m}{K^\dagger}$. Suppose we choose $g_D$ as a Laplace mechanism. That is, $g_D(\cdot|b)$ is $\mathtt{Laplace}(0, \frac{\mathtt{sens}(f)}{\epsilon})$, for all $b \in W$. Then, it satisfies (pure) $\epsilon$-differential privacy if $\mathtt{R}$ is the identity function since the Laplace mechanism performs output perturbation Dwork (2006). Due to the post-processing property of the standard differential privacy, it is clear that the Laplace mechanism $g_D$ is also $\epsilon$-differentially private for any non-identity $\mathtt{R}$.

**Gaussian Differential Privacy**   Next, we consider the scenario when $g_D$ is a Gaussian mechanism described in Theorem 3. In particular, given any $b \in W$, $g_D^j(\cdot|b) \in \Delta(\mathcal{Y}_j)$ is the density function of $\mathcal{N}(\mathtt{M}_b^j, \mathtt{V}^j)$ for all $j \in Q$, where $\mathtt{V}^j = \left(\frac{m}{K^\dagger \widehat{\mathtt{M}}_j}\right)^2$ and $\max_{b, b'} |\mathtt{M}_b^j - \mathtt{M}_{b'}^j| \leq \frac{m}{K^\dagger}$, for all $j \in Q$. By Lemma 6, we have

$$T\left[P_b(g_D^j), P_{b'}(g_D^j)\right](\alpha) \geq T\left[\mathcal{N}(0, 1), \mathcal{N}(\widehat{\mathtt{M}}_j, 1)\right],$$

for all adjacent $b$ and $b'$. Therefore, each $g_D^j$ satisfies $\widehat{\mathtt{M}}_j$-*Gaussian differential privacy* ($\widehat{\mathtt{M}}_j$-GDP) Dong et al. (2021), for all $j \in Q$. By Corollary 2.1 of Dong et al. (2021), this $\widehat{\mathtt{M}}_j$-GDP mechanism $g_D^j$ is also $(\epsilon_j, \varrho_j(\epsilon_j))$-DP for all $\epsilon_j \geq 0$ with

$$\varrho_j(\epsilon_j) = \Phi\left(-\frac{\epsilon_j}{\widehat{\mathtt{M}}_j} + \frac{\widehat{\mathtt{M}}_j}{2}\right) - e^{\epsilon_j} \Phi\left(-\frac{\epsilon_j}{\widehat{\mathtt{M}}_j} - \frac{\widehat{\mathtt{M}}_j}{2}\right),$$

where $\Phi$ is the cumulative distribution function (CDF) of the standard normal distribution. Under the assumption of linkage equilibrium and the construct of $g_D(y|b) = \prod_{j \in Q} g_D^j(y_j|b)$, the Gaussian defense strategy $g_D$ is $\overline{\mathsf{M}}$-GDP with $\overline{\mathsf{M}} = \sqrt{\sum_{j \in Q} \widehat{\mathsf{M}}_j^2}$ (Dong et al., 2021) due to the composition property.

# F    PROOF OF PROPOSITION 1

For any $0 < \gamma \le 1$, we can rewrite

$$\ell_A(s, b, \gamma) = -\sum_{k \in U} (s_k b_k - \gamma s_k) = -\sum_{k \in U} (b_k - \gamma) s_k$$

$$= -\sum_{k \in U} \left( (1-\gamma)\mathbf{1}_{\{s_k=1\}}\mathbf{1}_{\{b_k=1\}} - \gamma\mathbf{1}_{\{s_k=1\}}\mathbf{1}_{\{b_k=0\}} \right).$$

Then,

$$\mathcal{L}_A^\gamma(g_D, h_A) = \sum_{s,b} \int_x \ell_A(s, b, \gamma) h_A(s|x) \rho_D dr \sigma(b)$$

$$= -\sum_{s,b} \int_x \sum_{k \in U} \left( (1-\gamma)\mathbf{1}_{\{s_k=1\}}\mathbf{1}_{\{b_k=1\}} - \gamma\mathbf{1}_{\{s_k=1\}}\mathbf{1}_{\{b_k=0\}} \right) h_A(s|x) \rho_D(x|b) dx \sigma(b).$$

Since $\sigma = \theta$ and

$$\sum_{s_{-k}, b_{-k}} \left( (1-\gamma)\mathbf{1}_{\{s_k=1\}}\mathbf{1}_{\{b_k=1\}} - \gamma\mathbf{1}_{\{s_k=1\}}\mathbf{1}_{\{b_k=0\}} \right) h_A(s_k, s_{-k}|x)\theta(b_k, b_{-k})$$

$$= (1-\gamma)\sum_{b_{-k}} \Pr[s_k = 1|b_k = 1, x]\,\theta(b_k, b_{-k}) - \gamma\sum_{b_{-k}} \Pr[s_k = 1|b_k = 1, x]\,\theta(b_k, b_{-k}),$$

taking expectation over $x$ using $\rho_D$ yields $\mathcal{L}_A^\gamma(g_D, h_A) = -\mathtt{Adv}^\gamma(h_A)$. □

# G    PROOF OF PROPOSITION 2

We start by showing the *if* part. Suppose that $\mathtt{Adv}_k(g_D) \ge \mathtt{Adv}_k(g_D')$ for all $k \in U$. Then, the inequality also holds for all $k \in U$ if we apply both sides by the prior probabilities of $b_k = 0$ and $b_k = 1$. Thus, summing over all individuals yields $\mathtt{Adv}_k(g_D) \ge \mathtt{Adv}_k(g_D')$.

Next, we prove the *only if* part. Let $\mathtt{Adv}^*(g_D) \equiv \max_{h_A} \mathtt{Adv}^{0.5}(h_A, g_D)$, and let $\mathcal{L}_A^*(g_D) \equiv \min_{h_A} \mathcal{L}'^{\cdot \triangledown}{}_A(g_D, h_A)$. By Proposition 1, $\mathtt{Adv}^*(g_D) \ge \mathtt{Adv}^*(g_D')$ is equivalent to $\mathcal{L}_A^*(g_D) \le \mathcal{L}_A^*(g_D')$. By Theorem 1 of (de Oliveira, 2018), $g_D$ is more informative than $g_D'$ according to the Blackwell's informativeness ordering. Note that the informativeness ordering of $g_D$ and $g_D'$ is independent of the choice of priors. Thus, when $\mathcal{L}_A^*(g_D) \le \mathcal{L}_A^*(g_D')$ also holds when the prior $\theta$ is uniform. Let $\mathtt{Adv}^\dagger(g_D) \equiv \max_{h_A} \mathtt{Adv}^{0.5}(h_A, g_D)$ when $\theta$ is uniform. Thus, given any $g_D$, the optimal Bayes-weighted membership advantage simplifies to $\mathtt{Adv}^\dagger(g_D) = \frac{1}{2}\sum_{k \in U} \mathtt{Adv}_k(g_D)$. By definition of each $\mathtt{Adv}_{k_0}(g_D)$, the informativeness ordering of $g_D$ and $g_D'$ ensures that there exists no individual $k_0 \in U$ such that $\mathtt{Adv}_{k_0}(g_D) > \mathtt{Adv}_{k_0}(g_D')$. □

# H    PROOF OF THEOREM 1

In the proof, we use $g_D$ and $G$ interchangeably. For any function $V(s, b) \in \mathbb{R}$, define

$$Z(g_D, \sigma; V) \equiv \sum_{b,s} \int_x V(s, b)\mu_\sigma(s|x)\rho_D(x|b)dxq(b), \tag{10}$$

where $\mu_\sigma$ is the posterior belief induced by $g_D$ and $\sigma$, which is independent of the $\sigma$-Bayesian attacker's strategy $h_A$ and the test conclusions of $\alpha$-LRT attacker. In addition, define

$$L(g_D, h_A; V) \equiv \sum_{b,s} \int_x V(s, b)h_A(s|x)\rho_D(x|b)dx\theta(b). \tag{11}$$

Hence, when $V(\cdot) = \ell_A(\cdot; \gamma)$, $L(g_D, h_A; V) = \mathcal{L}_A^\gamma(g_D, h_A)$. Let $L^\sigma(g_D; V) \equiv \max_{h_A \in \mathcal{BR}^\sigma[g_D]} L(g_D, h_A; V)$. For simplicity, we write $Z(g_D, \sigma) = Z(g_D, \sigma; V)$ and $L^\sigma(g_D) = L^\sigma(g_D; V)$, unless otherwise stated. In addition, define the set

$$\mathcal{BR}^\sigma[g_D] \equiv \left\{ h_A^* \middle| h_A^* \in \arg\min_{h_A} \mathcal{L}_A^\gamma(g_D, h_A) \right\}.$$

**Proposition 7.** *For any $g_D$ and $\sigma$, $Z(g_D, \sigma) = L^\sigma(g_D)$.*

Proposition 7 shows that a $h_A$ that coincides with the posterior belief induced by $g_D$ and $\sigma$ leads to the minimum expected loss given any $V$.

**Proposition 8.** *Given any $G$ and $\sigma$, every $H^* \in \arg\min_H \mathcal{L}_{\mathrm{CEL}}(G, H)$ coincides with the posterior distribution $\mu_\sigma$ induced by $\sigma$ and $G$.*

Proposition 8 implies that every best response $H^* \in \arg\min_H \mathcal{L}_{\mathrm{CEL}}(G, H)$ leads to the probability distribution coincides with the posterior belief given $G$ and $\sigma$.

Then, by Proposition 7, every $H^* \in \mathrm{BN}[G; \sigma]$ leads to $\mathcal{L}_A^\gamma(G, H^*) \leq \mathcal{L}_A^\gamma(G, H')$ (i.e., when $V(\cdot) = \ell_A(\cdot; \gamma)$). Thus, we have $\mathrm{Adv}^{0.5}(H^*, \sigma, G^*) \leq \mathrm{Adv}^{0.5}(H', G')$. In addition, when $V(\cdot) = v(\cdot; \gamma)$, $H^* \in \mathrm{BN}[G; \sigma]$ leads to the minimum expected TPR. Therefor, it holds that $\mathrm{TPR}(G^*, H^*) \leq \mathrm{TPR}(G', H')$.

## H.1 POOF OF PROPOSITION 7

For simplicity, we omit $\gamma$ and denote $\ell_A(s, b) = \ell_A(s, b, \gamma)$

Given $\mu_\sigma$ (determined by $g_D$ and $\sigma$) and any $h_A$, let $\widehat{U}_A(h_A, b, x) \equiv \sum_s \ell_A(s, b) h_A(s|x) \mu_\sigma(b|x)$, which depends on the membership vector $b$ sampled by $\mu_\sigma$ but is independent of the samples $s$ drawn by $h_A$. Define

$$S^*[b, r; g_D] \equiv \left\{ h_A(\cdot|x) \Big| h_A(\cdot|x) \in \arg\min_{h'_A} \widehat{U}_A(h'_A, b, x) \right\},$$

for all $b \in W$ with $\mu_\sigma(b|x) > 0$, any $x \in \mathcal{X}$, where $S^*[b, x; g_D]$ depends on $g_D$ through $\mu_\sigma$. We first show that there is a $h_A^*(\cdot|x) \in S^*[b, x; g_D]$ that assigns probability 1 to $b$ (with $\mu_\sigma(b|x) > 0$). Suppose in contrast that $0 \leq h_A^*(b|x) < 1$. Then, it holds that $\sum_{s:s \neq b} \ell_A(s, b) h_A(s|x) \mu_\sigma(b|x) > 0$, which gives

$$\widehat{U}_A(h_A^*, b, x) = \sum_s \ell_A(s, b) h_A(s|x) \mu_\sigma(b|x)$$

$$= \sum_{s:s \neq b} \ell_A(s, b) h_A(s|x) \mu_\sigma(b|x) + \ell_A(b, b) h_A(b|x) \mu_\sigma(b|x)$$

$$> \ell_A(b, b) h_A(b|x) \mu_\sigma(b|x).$$

Thus, $\widehat{U}_A(h_A^*, b, x)|_{h_A^*(b|x) \neq 1} > \widehat{U}_A(h_A', b, x)|_{h_A'(b|x) = 1}$, which contradicts to $h_A^*(\cdot|x) \in S^*[b, x; g_D]$. Therefore, we have $\ell_A(b, b) \mu_\sigma(b|x) \leq \sum_s \ell_A(s, b) h_A(s|x) \mu_\sigma(b|x)$, for all $h_A(\cdot|x)$, $b \in W$, $x \in \mathcal{X}$, where the equality holds when $h_A(\cdot|x) \in S^*[b, x; g_D]$.

Let $h_A^\mu : \Gamma \mapsto \Delta(W)$ mirror the posterior belief $\mu_\sigma$; i.e., $h_A^\mu(s|x) = \mu_\sigma(b|x) \mathbf{1}\{s = b\}$, for all $s, b \in W$, $x \in \mathcal{X}$. It is clear that $h_A^\mu(\cdot|x) \in S^*[b, x; g_D]$ for all $b \in W$. Next, we show that if $h_A^\mu(s|x)$ is used by the $\sigma$-Bayesian attacker, it induces the WCPL for the vNM defender, which is captured by Proposition 9.

Let $L(g_D, h_A) \equiv \sum_{b,s} \int_r v(s, b) h_A(s|x) \rho_D(x|b) dx \theta(b)$ denote the expected true positive rate.

**Proposition 9.** *Given any $g_D$ and $\sigma$, $L(g_D, h_A^\mu) \leq L(g_D, h_A^*)$, for all $h_A^* \in \mathcal{BR}^\sigma[g_D]$.*

*Proof.* Define $\pi \equiv h_A \circ \rho_D : W \mapsto \Delta(W)$ by $\pi(s|b) = \sum_r h_A(s|r) \rho_D(r|b)$, for all $s, b \in W$. Define the set

$$\Pi[g_D] \equiv \{\pi = h_A \circ \rho_D | h_A : \Gamma \mapsto \Delta(W)\}.$$

That is, $\Pi[g_D]$ is the set of all feasible probabilistic mappings from a true membership vector $b$ to an inference $s$, perceived by the defender. We first establish the following lemma regarding the informativeness of $g_D$ in the sense of Blackwell's ordering of informatinveness (Blackwell, 1951; de Oliveira, 2018).

**Lemma 2.** *Fix any $\sigma \in \Delta(W)$. Given any two $g_D, g'_D$, $\Pi[g_D] \subseteq \Pi[g'_D]$, if and only if, for any function $\zeta : W \times W \mapsto \mathbb{R}$,*

$$\sum_{b,s} \zeta(s, b) \pi'(s|b) \sigma(b) \leq \sum_{b,s} \zeta(s, b) \pi(s|b) \sigma(b),$$

*where $\pi \in \Pi[g_D]$ and $\pi' \in \Pi[g'_D]$.*

*Proof.* We start by showing the *"only if"* part. Let $\Pi^*[g_D] \equiv \{\pi | \pi \in \arg\min_{\pi \in \Pi[g_D]} \sum_{b,s} \zeta(s, b) \pi'(s|b) \sigma(b)\}$. Since $\Pi[g_D] \subseteq \Pi[g'_D]$ and $\Pi^*[g_D] \subseteq \Pi[g_D]$, it must hold that $\Pi^*[g_D] \subseteq \Pi[g'_D]$. Hence,

$$\sum_{b,s} \zeta(s, b) \pi'(s|b) \sigma(b) \leq \sum_{b,s} \zeta(s, b) \pi(s|b) \sigma(b),$$

for all $\pi \in \Pi[g_D]$ and $\pi' \in \Pi[g'_D]$.

Next, we show the *"if"* part. Suppose in contrast that $\Pi[g_D] \not\subseteq \Pi[g'_D]$. Then, there exists a $\pi \in \Pi[g_D]$ such that $\pi \notin \Pi[g'_D]$. Since the set $\Pi[\bar{g}_D]$ for every $\bar{g}_D : W \mapsto \Delta(D)$ is closed under convex combinations of its elements, it is convex. In addition, it is a continuous image of a compact set in the space of probability distributions. Hence, the set $\Pi[\bar{g}_D]$ is also compact. The set $\Pi[\bar{g}_D]$ can be seen as a subset of $\mathbb{R}^{W \times W}$. Therefore, we can also perceive $\pi \in \mathbb{R}^{W \times W} \backslash \Pi[g'_D]$.

Let $\pi^\sigma(s, b) \equiv \pi(s|b)\sigma(b)$ for all $s, b \in W$. With abuse of notation, let $\Pi[g''_D, \sigma] \equiv \{\pi^\sigma | \pi \in \Pi[g''_D]\}$. Then, the set $\Pi[g''_D, \sigma]$ is a subset of $\mathbb{R}^{W \times W}$. Thus, $\pi^\sigma \in \mathbb{R}^{W \times W} \backslash \Pi[g'_D, \sigma]$. Let $\hat{\zeta} \in \mathbb{R}^{W \times W}$ represents the matrix form of the function $\zeta$. Since $|W| = 2^K$ with $K > 1$, there exists a separating hyperplane orthogonal to $\hat{\zeta}$, which separates the set $\Pi[g'_D]$ from the point $\pi$, such that

$$\sum\nolimits_{b,s} \zeta(s, b)\pi'(s|b) > \sum\nolimits_{b,s} \zeta(s, b)\pi(s|b),$$

for all $\pi' \in \Pi[g'_D]$. Then, the attacker with a non-informative (i.e., uniform prior) $\sigma$ obtains an ex-ante expected payoff using $h_A$ such that $\pi = h_A \circ g_D$ that is strictly better than any $h'_A$ such that $h'_A \circ \rho_D \in \Pi[g'_D]$. Thus, we obtain a contradiction to $\sum_{b,s} \zeta(s, b)\pi'(s|b)\sigma(b) \leq \sum_{b,s} \zeta(s, b)\pi(s|b)\sigma(b)$ for all $\sigma \in \Delta(W)$. $\square$

Next, we want to show that $\Pi[g_D] \subseteq \Pi[g'_D]$ is equivalent to $g'_D = \eta \circ g_D$ for some garbling $\eta : \Gamma \mapsto \Delta(\Gamma)$, which is another format of Blackwell's ordering of information structures (Blackwell, 1951; de Oliveira, 2018).

**Lemma 3.** *For any two $g_D, g'_D$, $\Pi[g'_D] \subseteq \Pi[g_D]$ if and only if $g'_D = \eta \circ g_D$ for some garbling $\eta : \mathcal{X} \mapsto \Delta(\mathcal{X})$.*

*Proof.* If $g'_D = \eta \circ g_D$, then there is a garbling $\hat{\eta} : \mathcal{X} \mapsto \Delta(\mathcal{X})$ such that $\rho'_D = \hat{\eta} \circ \rho_D$. Hence, $\pi' = \hat{\eta} \circ \pi$ for every $\pi' \in \Pi[g'_D]$ and $\pi \in \Pi[g_D]$. Then, from (1) and (2) of Theorem 1 in (de Oliveira, 2018), we obtain $g'_D = \hat{\eta} \circ g_D$ is equivalent to $\Pi[g'_D] \subseteq \Pi[g_D]$. $\square$

For simplicity, let $\mu_\sigma^x = \mu_\sigma(\cdot|x)$. Since $h_A \in \mathcal{BR}^\sigma[g_D]$, there exists a randomized correspondence $Y$ such that $h_A(\cdot|x) = Y(\cdot|\mu_\sigma^x)$ for all $x \in \mathcal{X}$. Then, from Blackwell's theorem (Blackwell, 1951; de Oliveira, 2018), there exists a garbling $y : W \mapsto \Delta(W)$ such that $h_A = y \circ \mu_\sigma$. Let $\hat{\rho}_D \equiv \mu_\sigma \circ \rho_D$ and let $\hat{\rho}'_D \equiv y \circ \hat{\rho}_D$. In addition, let $\hat{g}_D$ and $\hat{g}'_D$, respectively, be corresponding to $\hat{\rho}_D$ and $\hat{\rho}'_D$. Then, from Lemma 3, we have $\Pi[\hat{g}'_D] \subseteq \Pi[\hat{g}_D]$. In addition, Lemma 2 implies that

$$\sum\nolimits_{b,s} \zeta(s, b)\hat{\pi}(s|b)\sigma(b) \leq \sum\nolimits_{b,s} \zeta(s, b)\hat{\pi}'(s|b)\sigma(b)$$

for any $\sigma \in \Delta(W)$, any function $\zeta : W \times W \mapsto \mathbb{R}$, where $\hat{\pi} \in \Pi[\hat{g}_D]$ and $\hat{\pi}' \in \Pi[\hat{g}'_D]$. If we take $\zeta(\cdot) = V(\cdot)$ and $\sigma(\cdot) = \theta(\cdot)$, then we have $L^\sigma(\hat{g}_D) \geq L^\sigma(\hat{g}'_D)$. Therefore, $L(g_D, h^\mu_A) \leq L(g_D, h^*_A)$ for all $h^*_A \in \mathcal{BR}^\sigma[g_D]$, which concludes the proof of Proposition 9. $\square$

Next, we show that there is a $h^*_A \in \mathcal{BR}^\sigma[g_D]$ such that $h^*_A(s|x) = h^\mu_A(s|x)$ for all $s \in W$, $x \in \mathcal{X}$. Define $\widehat{U}^\natural(s, x) \equiv \sum_b \ell_A(s, b)\mu_\sigma(b|x)$, which depends on samples of $s \in W$ and $x \in \mathcal{X}$. Let

$$W^\natural[x] \equiv \left\{ s \in W \Big| s \in \arg\min\nolimits_{s'} \widehat{U}^\natural(s', x) \right\}.$$

Let $\hat{s} \in W$ such that $\hat{h}_A(\hat{s}|x) = 1$ for $\hat{h}_A \in S^*[b, x; g_D]$. We want to show $\hat{s} \in W^\natural[x]$. Suppose in contrast that $\hat{s} \notin W^\natural[x]$. Then, $\widehat{U}^\natural(s, x) < \widehat{U}^\natural(\hat{s}, x)$ for all $s \in W^\natural[x]$. That is, $\sum_b \ell_A(s, b)\mu_\sigma(b|x) < \sum_b \ell_A(\hat{s}, b)\mu_\sigma(b|x)$. Since $\hat{h}_A \in S^*[b, r; g_D]$, we have $\ell_A(\hat{s} = b, b) \leq \sum_s \mu_A(s, b)h'_A(s|x)$ for all $h'_A(\cdot|x)$, including $h'_A(s|x) = 1$ for any $s \in W$. Since every $\mu_\sigma(\cdot) \geq 0$, we have $\ell_A(\hat{s} = b, b)\mu_\sigma(b|x) \leq \mu_A(s, b)\mu_\sigma(b|x)$, for all $s, b \in W$. Then, $\widehat{U}^\natural(\hat{s}, x) \leq \widehat{U}^\natural(s, x)$, contradicting to $\hat{s} \notin W^\natural[x]$. Therefore, $\hat{s} \in W^\natural[x]$.

Next, we show that for every $s^* \in W^\natural[x]$, there is a $b \in W$ with $\mu_\sigma(b|x) > 0$ such that $\hat{h}_A(s^*|x) = 1$ for $\hat{h}_A \in S^*[b, r; g_D]$. Suppose in contrast that there exists a $s^* \in W^\natural[x]$ such that $\hat{h}_A(s^*|x) = 0$, for a $\hat{h}_A \in S^*[b, x; g_D]$. Then, there exists $\hat{s}$ with $\hat{h}_A(\hat{s}|x) = 1$ such that, for all $h'_A : \Gamma \mapsto \Delta(W)$,

$$\sum\nolimits_s \ell_A(s, b)\hat{h}_A(s|x)\mu_\sigma(b|x)$$
$$= \ell_A(\hat{s}, b)\mu_\sigma(b|x) \leq \ell_A(s^*, b)h'_A(s^*|x)\mu_\sigma(b|x) + \sum\nolimits_{s:s\neq s^*} \ell_A(s, b)h'_A(s|x)\mu_\sigma(b|x),$$

where the equality of the inequality holds when $h'_A = \hat{h}_A$. For all $h'_A \neq \hat{h}_A$, $h'_A(s^*|x) \in [0, 1]$, which implies $\ell_A(\hat{s}, b)\mu_\sigma(b|x) < \ell_A(\hat{s}, b)\mu_\sigma(b|x)$ for all $b \in W$ and $x \in \mathcal{X}$ with $\mu_\sigma(b|x) > 0$. Thus, $\sum_b \ell_A(\hat{s}, b)\mu_\sigma(b|x) < \sum_b \ell_A(s^*, b)\mu_\sigma(b|x)$, which contradicts to $s^* \in W^\natural[x]$. Therefore, $W^\natural[x] = \cup_b\{s \in$

$W|h_A(s|x) = 1, h_A \in S^*[b,x;g_D]\}$. It is not hard to see that every feasible mixed strategy $h_A(\cdot|x)$ that assigns strictly positive probability only to elements of $W^\natural[x]$ is a best response to $g_D$. Since $h_A^\mu \in S^*[b,x;g_D]$, we can conclude that $h_A^* \in \mathcal{BR}^\sigma[g_D]$ with $h_A^*(s|x) = h_A^\mu(s|x)$, for all $s \in W$, $x \in \mathcal{X}$. In addition, we can rewrite $Z(g_D,\sigma)$ in terms of $h_A^\mu$ as $Z(g_D,\sigma) = \sum_{b,s} \int_x v(s,b) h_A^\mu(s|x) \rho_D(x|b) dx q(b)$. Thus, by Proposition 9, we conclude that $L^\sigma(g_D) = Z(g_D,\sigma)$. $\qquad\square$

## H.2 PROOF OF PROPOSITION 8

For ease of exposition, we directly use the underlying density functions $g_D$ (and $\rho_D$) and $h_A$ of $G$ and $H$, respectively, so that $h_A(s_k = 1) = q_k$ and $h_A(s_k = 0) = 1 - q_k$. Thus, we can rewrite

$$\mathcal{L}_{\text{CEL}}(G,H) = \mathcal{L}_{\text{CEL}}(g_D, h_A) = -\sum_b \int_x \sigma(b) \log(h_A(b|x)) \rho_D(x|b) dx.$$

Let $\mu_\sigma$ denote the posterior distribution induced by $\sigma$ and $g_D$ according to Bayes' rule. Then,

$$\mathcal{L}_{\text{CEL}}(g_D, h_A) - \mathcal{L}_{\text{CEL}}(g_D, \mu_\sigma)$$
$$= -\sum_b \int_x \sigma(b) \log(h_A(b|x)) \rho_D(x|b) dx + \sum_b \int_x \sigma(b) \log(\mu_\sigma(b|x)) \rho_D(x|b) dx$$
$$= \sum_b \int_x \sigma(b) \rho_D(x|b) \left( \log(\mu_\sigma(b|x)) - \log(h_A(b|x)) \right) dx$$
$$= \sum_b \int_x \sigma(b) \rho_D(x|b) \log(\frac{\mu_\sigma(b|x)}{h_A(b|x)}) dx.$$

By definition of $\mu_\theta$ using Bayes' rule, we have

$$\sigma(b) \rho_D(x|b) = \mu_\sigma(b|x) \mathrm{P}^\sigma(x),$$

where $\mathrm{P}^\sigma(x) \equiv \sum_{b'} \sigma(b) \rho_D(x|b)$. Then, we have

$$\mathcal{L}_{\text{CEL}}(g_D, h_A) - \mathcal{L}_{\text{CEL}}(g_D, \mu_\sigma) = \sum_b \int_x \mu_\sigma(b|x) \mathrm{P}^\sigma(x) \log \left( \frac{\mu_\sigma(b|x)}{h_A(b|x)} \right) dx \geq 0$$

which is non-negative because it is the Kullback–Leibler (KL) divergence. In addition, $\mathcal{L}_{\text{CEL}}(g_D, h_A) - \mathcal{L}_{\text{CEL}}(g_D, \mu_\sigma) = 0$ if and only if $h_A(b|x) = \mu_\sigma(b|x)$ for all $b \in W$ and $x \in \mathcal{X}$.

$\qquad\square$

# I PROOF OF PROPOSITION 3

By by Theorem 2.10. of (Dong et al., 2021) (also see (Blackwell, 1951)), we have that for a fixed significance level, the minimum false positive rates (of inferring each individual $k$'s membership status), denoted by $T(G)$ and $T(\text{Proc} \circ G)$, can be achieved by $G$ and $\text{Proc} \circ G$ satisfy

$$T(\text{Proc} \circ G) \geq T(G).$$

Thus, $G$ is more informative than $\text{Proc} \circ G$ according to Blackwell's ordering of informativeness (Blackwell, 1951). By Theorem 1 of (de Oliveira, 2018), we can conclude that $\mathcal{L}_{\text{CEL}}(\text{Proc} \circ G, H') \geq \mathcal{L}_{\text{CEL}}(G, H)$. $\qquad\square$

# J PROOF OF PROPOSITION 4

For ease of exposition, we focus on the case when there are two mechanisms that are composed. That is, $\vec{G} = (G_1, G_2)$. The proof can be easily extended to general $n \geq 2$.

We start by proving the scenario when the mechanisms are independent. Let $\rho_1(\cdot|b) \in \Delta(\mathcal{X}_1)$ and $\rho_2(\cdot|b) \in \Delta(\mathcal{X}_2)$ be induced density functions by $G_1$ and $G_2$, respectively. In addition, we directly use the underlying density function $h_A$ of $H$, so that $h_A(s_k = 1) = q_k$ and $h_A(s_k = 0) = 1 - q_k$. We can easily generalize Proposition 8 in Appendix H to the case when there are multiple data-sharing mechanisms randomized by $\vec{G}$. That is, every $H \in \arg\min_{H'} \mathcal{L}_{\text{CEL}}(G, H')$ coincides with the posterior distribution $\mu_\theta(b_k, b_{-k}) =$

$\frac{\rho_1(x_1|b)\rho_2(x_2|b)\theta(b)}{\sum_{b'}\rho_1(x_1|b')\rho_2(x_2|b')\theta(b')}$ induced by $\vec{G}$ and the prior $\theta$. Thus, we have

$$\mathcal{L}_{\text{CEL}}(\vec{G}, \vec{H}^*) = -\sum_{b \in W} \int_{x_1, x_2} \theta(b) \log(h_A(b|x_1, x_2))\theta(b)\rho_1(x_1|b)\rho_2(x_2|b)dx_1 dx_2$$

$$= -\sum_{b \in W} \int_{x_1, x_2} \theta(b) \log(\mu_\theta(b|x_1, x_2))\theta(b)\rho_1(x_1|b)\rho_2(x_2|b)$$

$$= -\sum_b b\theta(b) \log(\theta(b)) \cdot \int_{x_1, x_2} \rho_1(x_1|b)\rho_2(x_2|b)dx_1 dx_2$$

$$- \sum_b \int_{x_1, x_2} \theta(b) \cdot \rho_1(x_1|b)\rho_2(x_2|b) \cdot \log(\rho_1(x_1|b)\rho_2(x_2|b))dx_1 dx_2$$

$$+ \sum_b \int_{x_1, x_2} \theta(b) \cdot \rho_1(x_1|b)\rho_2(x_2|b) \cdot \log\left(\sum_{b'} \rho_1(x_1|b')\rho_2(x_2|b')\theta(b')\right)dx_1 dx_2$$

$$= -\sum_b \theta(b) \log(\theta(b)) - \sum_b \int_{x_1, x_2} \theta(b) \cdot \rho_1(x_1|b)\rho_2(x_2|b) \cdot \log(\rho_1(x_1|b)\rho_2(x_2|b))dx_1 dx_2$$

$$+ \int_{x_1, x_2} \left(\sum_{b'} \rho_1(x_1|b')\rho_2(x_2|b')\theta(b')\right) \cdot \log\left(\sum_{b'} \rho_1(x_1|b')\rho_2(x_2|b')\theta(b')\right).$$

We can following the same steps for each $G_j$ for $j \in \{1, 2\}$ with $H_j \in \arg\min_{H_j'} \mathcal{L}_{\text{CEL}}(G_j, H_j')$:

$$\mathcal{L}_{\text{CEL}}(G_j, H_j) = -\sum_b \theta(b) \log(\theta(b)) - \sum_b \int_{x_j} \theta(b) \cdot \rho_j(x_j|b) \cdot \log\left(\rho_j(x_j|b)\right) dx_j$$

$$+ \int_{x_j} \left(\sum_{b'} \rho_j(x_j|b')\theta(b)\right) \log\left(\sum_{b'} \rho_j(x_j|b')\theta(b)\right) dx_j.$$

Summing individual losses yields

$$\sum_{j=1}^{2} \mathcal{L}_{\text{CEL}}(G_j, H_j) = -2\sum_b \theta(b) \log(\theta(b))$$

$$- \sum_{j=1}^{2} \sum_b \int_{x_j} \theta(b) \cdot \rho_j(x_j|b) \cdot \log\left(\rho_j(x_j|b)\right) dx_j$$

$$+ \sum_{j=1}^{2} \int_{x_j} \left(\sum_{b'} \rho_j(x_j|b')\theta(b')\right) \log\left(\sum_{b'} \rho_j(x_j|b')\theta(b)\right) dx_j.$$

Let $\mathcal{H} \equiv \sum_b \theta(b) \log(\theta(b))$, $\mathcal{F}_j \equiv \sum_b \int_{x_j} \theta(b) \cdot \rho_j(x_j|b) \cdot \log\left(\rho_j(x_j|b)\right) dx_j$, and $\mathcal{K} \equiv \sum_b \theta(b) \int_{x_1, x_2} \rho_1(x_1|b)\rho_2(x_2|b) \cdot \log\left(\sum_{b'} \rho_1(x_1|b')\rho_2(x_2|b')\theta(b')\right) dx_1 dx_2$.

Since

$$\sum_b \int_{x_1, x_2} \theta(b) \cdot \rho_1(x_1|b)\rho_2(x_2|b) \cdot \log(\rho_1(x_1|b)\rho_2(x_2|b))dx_1 dx_2$$

$$= \sum_b \int_{x_1, x_2} \theta(b) \cdot \rho_1(x_1|b)\rho_2(x_2|b) \cdot (\log(\rho_1(x_1|b)) + \log(\rho_2(x_2|b))) \, dx_1 dx_2$$

$$= \sum_{j=1}^{2} \sum_b \int_{x_j} \theta(b) \cdot \rho_j(x_j|b) \cdot \log(\rho_j(x_j|b)) \, dx_j = \sum_{j=1}^{2} \mathcal{F}_j,$$

we obtain

$$\mathcal{L}_{\text{CEL}}(\vec{G}, \vec{H}^*) = \left(\sum_{j=1}^{2} (\mathcal{L}_{\text{CEL}}(G_j, H_j) - \mathcal{F}_j) + 2\mathcal{H}\right) - \mathcal{H} + \mathcal{K}$$

$$= \sum_{j=1}^{2} \mathcal{L}_{\text{CEL}}(G_j, H_j) + \mathcal{H}(b) - \sum_{j=1}^{2} \mathcal{F}_j(\tilde{x}_j) + \mathcal{K}.$$

By combining $\Lambda(\vec{G}, \theta) = -\left(\mathcal{H}(b) - \sum_{j=1}^{2} \mathcal{F}_j(\tilde{x}_j) + \mathcal{K}\right)$, we have

$$\Lambda(\vec{G}, \theta) = -\sum_b \theta(b) \int_{x_1, x_2} \rho_1(x_1|b)\rho_2(x_2|b) \cdot \log \left(\sum_{b'} \rho_1(x_1|b)\rho_2(x_2|b)\theta(b')\right) dx_1 x_2.$$

For general $n \geq 2$, we have $\Lambda(\vec{G}, \theta) = -\sum_b \theta(b) \int_{\vec{x}} \vec{\rho}_D(\vec{x}|b) \cdot \log \left(\sum_{b'} \vec{\rho}_D(\vec{x}|b')\theta(b')\right) d\vec{x}$.

Next, we proceed with the proof when the mechanisms are correlated. Again, for simplicity, we first focus on $n = 2$, i.e., $\vec{G} = (G_1, G_2)$, and generalize to $n \geq 2$ afterward.

By Proposition 8, the posterior distribution $\mu_\theta(b|x_1, x_2)$ is given by:

$$\mu_\theta(b|x_1, x_2) = \frac{\vec{\rho}_D(x_1, x_2|b)\theta(b)}{\sum_{b'} \vec{\rho}_D(x_1, x_2|b')\theta(b')}.$$

The BGP risk $\mathcal{L}_{\text{CEL}}(\vec{G}, \vec{H}^*)$ for the composed mechanism becomes:

$$\mathcal{L}_{\text{CEL}}(\vec{G}, \vec{H}^*) = -\sum_b \int_{x_1, x_2} \theta(b) \log(\mu_\theta(b|x_1, x_2))\vec{\rho}_D(x_1, x_2|b) dx_1 dx_2.$$

Substituting $\mu_\theta(b|x_1, x_2)$ into the loss function:

$$\mathcal{L}_{\text{CEL}}(\vec{G}, \vec{H}^*) = -\sum_b \int_{x_1, x_2} \theta(b)\vec{\rho}_D(x_1, x_2|b) \log \left(\frac{\vec{\rho}_D(x_1, x_2|b)\theta(b)}{\sum_{b'} \vec{\rho}_D(x_1, x_2|b')\theta(b')}\right) dx_1 dx_2,$$

which can be broken into three terms:

$$\mathcal{L}_{\text{CEL}}(\vec{G}, \vec{H}^*) = -\sum_b \theta(b) \log(\theta(b)) \int_{x_1, x_2} \vec{\rho}_D(x_1, x_2|b) dx_1 dx_2$$

$$-\sum_b \int_{x_1, x_2} \theta(b)\vec{\rho}_D(x_1, x_2|b) \log(\vec{\rho}_D(x_1, x_2|b)) dx_1 dx_2$$

$$+\int_{x_1, x_2} \sum_b \theta(b)\vec{\rho}_D(x_1, x_2|b) \log \left(\sum_{b'} \vec{\rho}_D(x_1, x_2|b')\theta(b')\right) dx_1 dx_2.$$

Define the following terms:

$$\mathcal{H} \equiv -\sum_b \theta(b) \log(\theta(b)), \quad \text{(entropy of the prior)}$$

$$\mathcal{F} \equiv -\sum_b \int_{x_1, x_2} \theta(b)\vec{\rho}_D(x_1, x_2|b) \log(\vec{\rho}_D(x_1, x_2|b)) dx_1 dx_2, \quad \text{(conditional entropy)}$$

$$\mathcal{K} \equiv \int_{x_1, x_2} \sum_b \theta(b)\vec{\rho}_D(x_1, x_2|b) \log \left(\sum_{b'} \vec{\rho}_D(x_1, x_2|b')\theta(b')\right) dx_1 dx_2, \quad \text{(interaction term)}.$$

Thus, the loss can be expressed as:

$$\mathcal{L}_{\text{CEL}}(\vec{G}, \vec{H}^*) = \mathcal{H} + \mathcal{F} - \mathcal{K}.$$

By combining $\Lambda(\vec{G}, \theta) = -(\mathcal{H} - \mathcal{F} + \mathcal{K})$, we have

$$\Lambda(\vec{G}, \theta) = -\sum_b \theta(b) \int_{x_1, x_2} \vec{\rho}D(x_1, x_2|b) \log \left(\frac{\sum b' \vec{\rho}D(x_1, x_2|b')\theta(b')}{P(x_1, x_2)}\right) dx_1 dx_2,$$

where

$$P(x_1, x_2) = \prod j = 1^2 \sum_{b'} \int_{\mathcal{X}_{-j}} \vec{\rho}D(x_j, x_{-j}|b')\theta(b')dx_{-j}.$$

For $n \geq 2$, this expression naturally generalizes:

$$\Lambda(\vec{G}, \theta) = -\sum_b \theta(b) \int_{\vec{x}} \vec{\rho}D(\vec{x}|b) \log \left(\frac{\sum b' \vec{\rho}_D(\vec{x}|b')\theta(b')}{P(\vec{x})}\right) d\vec{x},$$

where $P(\vec{x}) = \prod_{j=1}^{n} \sum_{b'} \int_{\vec{\mathcal{X}}_{-j}} \vec{\rho}_D(x_j, \vec{x}_{-j}|b')\theta(b')d\vec{x}_{-j}$. $\qquad \square$

## K    PROOF OF PROPOSITION 5

Let $\vec{x} = (x_1, \ldots, x_n)$ denote the outputs of the composition $\mathcal{M}(\vec{G}^*)$, and let $\vec{\rho}_D$ is the joint distribution given $\vec{G}$, $\{f_1, \ldots, f_n\}$, and any intrinsic correlations among mechanisms. For simplicity, we consider each individual $k$ has probability $\theta_k = \theta_k(b_k = 1)$ to have $b_k = 1$ and probability $1 - \theta_k$ to have $b_k = 0$. Consider the following binary hypothesis test:

$$\mathcal{H}_0^k: b_k = 0 \text{ with } b_{-k} \text{ vs. } \mathcal{H}_1^k: b_k = 1 \text{ with } b_{-k},$$

where $b_{-k}$ is the same for both $\mathcal{H}_0^k$ and $\mathcal{H}_1^k$. Since $\vec{\rho}_D(\vec{x}|b)$ is well-defined, this binary hypothesis test is a well-defined simple binary hypothesis test. Then, the Neyman-Pearson lemma implies that the likelihood-ratio test is the Uniformly Most Powerful (UMP) test. Then, for any given significance level $\alpha^k$, there exists a rejection rule $\phi$ such that

$$\alpha^k = \mathbb{E}\left[\phi \Big| \mathcal{H}_0^k, \vec{G}^*\right] \text{ and } \mathtt{f}(\alpha) = 1 - \mathbb{E}\left[\phi \Big| \mathcal{H}_1^k, \vec{G}^*\right],$$

where $\mathtt{f}(\alpha^k) = \int \left\{ t \in [0,1] : \mathtt{f}(t) \le \alpha^k \right\}$ is the *symmetric trade-off function* introduced by Dong et al. (2022) , which is convex, continuous, non-increasing, and satisfies $\mathtt{f}(\alpha^k) \le 1 - \alpha^k$ and $\mathtt{f}(\alpha^k) = \inf \left\{ t \in [0,1] : \mathtt{f}(t) \le \alpha^k \right\}$ for $\alpha^k \in [0,1]$. Hence, the composition $\mathcal{M}(\vec{G}^*)$ is $\mathtt{f}$-differentially private ($\mathtt{f}$-DP) (Definition 2.3 of (Dong et al., 2022)). Then, by Proposition 2.4 of Dong et al. (2022), $\mathcal{M}(\vec{G}^*)$ is also $\mathtt{f}^*$-DP, where $\mathtt{f}^*(z) = \sup_{0 \le \hat{\alpha}^k \le 1} z\hat{\alpha}^k - f(\hat{\alpha}^k)$. Then, by Proposition 2.12 of Dong et al. (2022), for any $\epsilon^k \ge 0$, the composition $\mathcal{M}(\vec{G}^*)$ is $(\epsilon^k, \xi^k)$-DP, where $\xi^k = \delta(\epsilon^k)$, for any given $b_k$. Therefore, the composition $\mathcal{M}$ is $(\epsilon, \xi)$-DP for some $\epsilon \ge 0$ and $\xi \in [0,1]$.    $\square$

## L    PROOF OF PROPOSITION 6

Proposition 8 in Appendix H can be easily extended to the case when there are multiple data-sharing mechanisms randomized by $\vec{G}$. Thus, if $H^* \in \arg\min_H \mathcal{L}_{\mathrm{CEL}}(\vec{G}, H) \bigcap \mathtt{DPH}[\vec{G}; \epsilon]$, the posterior distribution induced by $\vec{G}$ and $\theta$ satisfies the conditions specified by $\mathtt{DPH}[\vec{G}; \epsilon]$. By the necessary and sufficient condition given by Claim 3 of (Dwork et al., 2006), $\vec{G}$ is $\epsilon$-DP. Again, by Claim 3 of (Dwork et al., 2006), if $\vec{G}$ is $\epsilon$-DP, the posterior distribution must satisfy the conditions specified by $\mathtt{DPH}[\vec{G}; \epsilon]$. Then, based on the extension of Proposition 8, it must holds that $H^* \in \arg\min_H \mathcal{L}_{\mathrm{CEL}}(\vec{G}, H) \bigcap \mathtt{DPH}[\vec{G}; \epsilon]$.    $\square$

## M    PROOF OF THEOREM 2

For any function $V(s,b) \in \mathbb{R}$, define

$$Z(g_D, \sigma; V) \equiv \sum_{b,s} \int_x V(s,b)\mu_\sigma(s|x)\rho_D(x|b)dxq(b), \tag{12}$$

where $\mu_\sigma$ is the posterior belief induced by $g_D$ and $\sigma$, which is independent of the $\sigma$-Bayesian attacker's strategy $h_A$ and the test conclusions of $\alpha$-LRT attacker. In addition, define

$$L(g_D, h_A; V) \equiv \sum_{b,s} \int_x V(s,b)h_A(s|x)\rho_D(x|b)dx\theta(b). \tag{13}$$

Hence, when $V(\cdot) = \ell_A(\cdot; \gamma)$, $L(g_D, h_A; V) = \mathcal{L}_A^\gamma(g_D, h_A)$. Let $L^\sigma(g_D; V) \equiv \max_{h_A \in \mathcal{BR}^\sigma[g_D]} L(g_D, h_A; V)$. For simplicity, we write $Z(g_D, \sigma) = Z(g_D, \sigma; V)$ and $L^\sigma(g_D) = L^\sigma(g_D; V)$, unless otherwise stated. Define the set

$$\mathcal{BR}^\sigma[g_D] \equiv \left\{ h_A^* \Big| h_A^* \in \arg\min_{h_A} \mathcal{L}_A^\gamma(g_D, h_A) \right\}.$$

We start by proving Lemma 4.

**Lemma 4.** *Fix any $g_D$. Suppose that $\sigma$ is aligned. Let $h_A^\mu : \mathcal{X} \mapsto \Delta(W)$ be defined by $h_A^\mu(s|x) = \mu_\sigma(b|x)\mathbf{1}(s = b)$ for all $s, b \in W$, $x \in \mathcal{X}$. Then, $h_A^\mu \in \mathcal{BR}^\sigma[g_D]$.*

*Proof.* Let $V_A^\ddagger(s,x) \equiv \sum_b \ell_A(s,b)\mu_\sigma(b|x)$ Define $W^\ddagger[x] \equiv \{s \in W | s \in \arg\min_{s'} V_A^\ddagger(s,x)\}$. Hence, each $h_A : \mathcal{X} \mapsto \Delta(W)$ that only assigns strictly positive probabilities to $s \in W^\ddagger[x]$ satisfies $h_A \in \mathcal{BR}^\sigma[g_D]$. In addition, let $W^\sharp[x] \equiv \{s \in W | \mu_\sigma(s|x) > 0\}$. By definition of $c_A$ and $v_{s,b}$, $\gamma c_A(s) - v_A(s,b)$ (weakly) decreases as $\sum_{k \in U} \mathbf{1}\{s_k = b_k\}$ increases. Thus, $V_A^\ddagger(s^\sharp, x) \le V_A^\ddagger(s,x)$ for all $s^\sharp \in W^\sharp[x]$ and $s \in W$. Hence, $W^\sharp[x] \subseteq W^\ddagger[x]$ for all $x$. Hence, $h_A^\mu \in \mathcal{BR}_\Gamma^\sigma[g_D]$ holds.    $\square$

With abuse of notation, we let $q(b)$ and $q(b_k) = \sum_{b_{-k}} q(b_k, b_{-k})$ denote the prior and the marginalized prior, respectively. Next, we show that optimal $\alpha$-LRT cannot strictly outperform $\sigma$-Bayesian under the same $g_D$.

**Lemma 5.** *Fix $g_D$ and $\alpha$. Suppose $\sigma = q$. Then, $Z(g_D, q) \geq L_{\text{Opt-LRT}}^{\alpha}(g_D)$.*

*Proof.* Suppose in contrast that $Z(g_D, q) < L_{\text{Opt-LRT}}^{\alpha}(g_D)$. Then,

$$\sum_k P_1^k \left[ y_k(x, \tau^*) = 1 | g_D \right] q(b_k = 1) > \sum_{b,s} \int_x v(s, b) \mu_\sigma(s|x) \rho_D(x|b) dx q(b)$$
$$= \sum_k P_\sigma^k \left[ s_k = 1 | g_D, b_k = 1 \right] q(b_k = 1),$$

where $P_\sigma^k \left[ s_k = 1 | g_D, b_k = 1 \right] = \int_x \mu_\sigma(s_k = 1|x) \rho_D(x|b) dx$. By letting $V(\cdot) = v(\cdot)$, from Proposition 7, we have $L^\sigma(g_D) = Z(g_D, q; v) < L_{\text{Opt-LRT}}^{\alpha}(g_D)$. Let $h_A^\dagger(s_k = 1|x) = \mathbf{1}\{y_k(x, \tau^*) = 1\}$ for all $x \in \mathcal{X}$. Since $\sigma = q$, $h_A^\dagger$ is the best response of the Bayesian attacker. Hence, $L_{\text{Opt-LRT}}^{\alpha}(g_D) = L(g_D, h_A^\dagger) \leq Z(g_D, q; v)$, which contradicts to $Z(g_D, q; v) < L_{\text{Opt-LRT}}^{\alpha}(g_D)$. Therefore, $Z(g_D, q; v) \geq L_{\text{Opt-LRT}}^{\alpha}(g_D)$. $\square$

If $\sigma$ is informative, we have $L(g_D, h_A^\sigma) \leq L(g_D, h_A^\theta)$. Hence, it also holds that $Z(g_D, \sigma; v) \geq Z(g_D, \theta; v)$. Lemma 5 imples $Z(g_D, \sigma; v) \geq L_{\text{Opt-LRT}}^{\alpha}(g_D)$.

Next, we show that when $\sigma$ is non-informative. Let $h_A^\sigma(s|x) = \mu_\sigma(b|x)\mathbf{1}\{s = b\}$, for all $s, b \in W$, $x \in \mathcal{X}$. By Lemma 4, it holds that $h_A^\sigma \in \mathcal{BR}^\sigma[g_D]$. Suppose in contrast that $L_{\text{Opt-LRT}}^{\alpha}(g_D) > Z(g_D, \sigma; v)$. Then, $h_A^\sigma \in \mathcal{BR}_\Gamma^\sigma[g_D]$ implies

$$\sum_k P_1^k \left[ y_k(x, \tau^*) = 1 | g_D \right] > \sum_{k,s} \int_x v(s_k = 1, b_k = 1) \mu_\sigma(s_k = 1|x).$$

Let $h_A^\dagger : \mathcal{X} \mapsto \Delta(W)$ such that $h_A^\dagger(s_k = 1|x) = \mathbf{1}\{y_k(x, \tau^*) = 1\}$ for all $x \in \mathcal{X}$. Then, $h_A^\dagger \in \mathcal{BR}^\sigma[g_D]$ when $\sigma$ is uniform (i.e., non-informative). Proposition 7 implies $Z(g_D, \sigma; v) \geq L(g_D, h_A^\dagger, \sigma)$, which leads to a contradiction. The inequality $L_{\text{Opt-LRT}}^{\alpha}(g_D) \geq L_{\text{Adp}}^{\alpha}(g_D)$ follows the Neyman-Pearson lemma. In addition, by Venkatesaramani et al. (2021; 2023), $L_{\text{Adp}}^{\alpha}(g_D) \geq L_{\text{Fixed}}^{\alpha}(g_D)$. Thus, we can conclude the proof of Theorem 2. $\square$

# N  PROOF OF LEMMA 1

First, we show that the test statistics $\mathcal{L}(\tilde{y}) = \sum_{j \in Q} \log \left( \rho_j(\tilde{y}_j | \widehat{H}_0) / \rho_j(\tilde{y}_j | \widehat{H}_1) \right)$ is normally distributed under $\widehat{H}_0$ and $\widehat{H}_1$, respectively, with $\mathcal{N}\left(\overline{\mathrm{M}}, \overline{\mathrm{V}}\right)$ and $\mathcal{N}\left(-\overline{\mathrm{M}}, \overline{\mathrm{V}}\right)$, where $\overline{\mathrm{M}} = \frac{1}{2} \sum_{j \in Q} \widehat{\mathrm{M}}_j^2$ and $\overline{\mathrm{V}} = \sum_{j \in Q} \widehat{\mathrm{M}}_j^2$. For each $y_j$, $\tilde{y}_j \sim \mathcal{N}(0, 1)$ under $\widehat{H}_0$, and $\tilde{y}_j \sim \mathcal{N}(\widehat{M}_j, 1)$ under $\widehat{H}_1$. Thus, the log-likelihood ratio for each $y_j$ is $\log \left( \frac{\rho_j(y_j | \widehat{H}_0)}{\rho_j(y_j | \widehat{H}_1)} \right)$. Since $\rho_j(\cdot | \widehat{H}_0)$ and $\rho_j(\cdot | \widehat{H}_1)$ are the density functions of normal distribution, the log-likelihood ratio becomes

$$\log \left( \frac{\frac{1}{\sqrt{2\pi}} e^{-\frac{y_j^2}{2}}}{\frac{1}{\sqrt{2\pi}} e^{-\frac{(y_j - \widehat{M}_j)^2}{2}}} \right) = \frac{(y_j - \widehat{M}_j)^2 - y_j^2}{2} = \frac{-2y_j\widehat{M}_j + \widehat{M}_j^2}{2} = -y_j\widehat{M}_j + \frac{\widehat{M}_j^2}{2}.$$

Under $\widehat{H}_0$, the mean is $\mathbb{E}[y_j | \widehat{H}_0] = 0$ and the variance is $\text{Var}[y_j | \widehat{H}_0] = 1$. Hence, the mean of $\mathcal{L}(y)$ under $\widehat{H}_0$ is

$$\mathbb{E}[\mathcal{L}(y)] = \mathbb{E}\left[ \sum_{j \in Q} \left( -y_j\widehat{M}_j + \frac{\widehat{M}_j^2}{2} \right) \right] = \sum_{j \in Q} \left( -\mathbb{E}[y_j]\widehat{M}_j + \frac{\widehat{M}_j^2}{2} \right) = \sum_{j \in Q} \frac{\widehat{M}_j^2}{2},$$

and the variance is

$$\text{Var}[\mathcal{L}(y)] = \text{Var}\left[ \sum_{j \in Q} \left( -y_j\widehat{M}_j + \frac{\widehat{M}_j^2}{2} \right) \right] = \sum_{j \in Q} \text{Var}[-y_j\widehat{M}_j] = \sum_{j \in Q} \widehat{M}_j^2.$$

Similarly, under $\widehat{H}_1$, the mean of $\mathcal{L}(y)$ is

$$\mathbb{E}[\mathcal{L}(y)] = \mathbb{E}\left[ \sum_{j \in Q} \left( -y_j\widehat{M}_j + \frac{\widehat{M}_j^2}{2} \right) \right] = \sum_{j \in Q} \left( -\mathbb{E}[y_j]\widehat{M}_j + \frac{\widehat{M}_j^2}{2} \right) = \sum_{j \in Q} \left( -\widehat{M}_j^2 + \frac{\widehat{M}_j^2}{2} \right) = \sum_{j \in Q} -\frac{\widehat{M}_j^2}{2}.$$

In addition, the variance of $\mathcal{L}(y)$ under $\widehat{H}_1$ is

$$\text{Var}[\mathcal{L}(y)] = \text{Var}\left[\sum_{j \in Q}\left(-y_j\widehat{\mathtt{M}}_j + \frac{\widehat{\mathtt{M}}_j^2}{2}\right)\right] = \sum_{j \in Q}\text{Var}[-y_j\widehat{\mathtt{M}}_j] = \sum_{j \in Q}\widehat{\mathtt{M}}_j^2.$$

Since the test statistics $\mathcal{L}(y)$ is normally distributed under $\widehat{H}_0$ and $\widehat{H}_1$, we have

$$Z_0 = \frac{\overline{y} - \overline{\mathtt{M}}}{\sqrt{\overline{\mathtt{V}}/\sqrt{m}}} \sim \mathcal{N}(0,1) \text{ and } Z_1 = \frac{\overline{y} + \overline{\mathtt{M}}}{\sqrt{\overline{\mathtt{V}}/\sqrt{m}}} \sim \mathcal{N}\left(\frac{-2\overline{\mathtt{M}}}{\sqrt{\overline{\mathtt{V}}/\sqrt{m}}}, 1\right),$$

where $\overline{y}$ is the sample mean. For a given significance level $\widehat{\alpha}$, the threshold for $Z_0$ is set so that $\Pr(Z_0 < z_{\widehat{\alpha}}) = \widehat{\alpha}$, corresponding to the value $\overline{\mathtt{M}} + z_{\widehat{\alpha}}\sqrt{\frac{\overline{\mathtt{V}}}{m}}$. For a given Type-II error rate $\widehat{\beta}$, the threshold for $Z_1$ is set so that $\Pr(Z_1 < z_{\widehat{\beta}}) = \widehat{\beta}$, where $z_{\widehat{\beta}}$ aligns with $-\overline{\mathtt{M}} - z_{\widehat{\beta}}\sqrt{\frac{\overline{\mathtt{V}}}{m}}$. To maintain the consistency of decision-making between $\widehat{H}_0$ and $\widehat{H}_1$, the threshold at which we switch decisions from failing to reject $\widehat{H}_0$ to rejecting $\widehat{H}_0$ under $\widehat{H}_0$ and $\widehat{H}_1$ are equated. Therefore, we have

$$\sqrt{m}\overline{\mathtt{M}} + z_{\widehat{\alpha}}\sqrt{\overline{\mathtt{V}}} = -\sqrt{m}\overline{\mathtt{M}} - z_{\widehat{\beta}}\sqrt{\overline{\mathtt{V}}}.$$

Thus, $\mathcal{F}\left(\widehat{\alpha}, \widehat{\beta}\right) = m$ holds.

Next, we show the monotone relationship between $\widehat{\beta}$ and $m$ given $\mathcal{F}\left(\widehat{\alpha}, \widehat{\beta}\right) = m$ while everything else is fixed. Since, $z_{\widehat{\beta}} = \Phi^{-1}(1 - \widehat{\beta})$, where $\Phi$ is the cumulative distribution function (CDF) of the standard normal distribution, $z_{\widehat{\beta}}$ decreases as $\widehat{\beta}$ increases as the quantile function $\Phi^{-1}$ decreases as the probability increases. As a result, $(z_{\widehat{\alpha}}, z_{\widehat{\beta}})$ decreases when $\widehat{\beta}$ increases. Therefore, $\mathcal{F}\left(\widehat{\alpha}, \widehat{\beta}\right) = m$ implies that $m$ decreases when $\widehat{\beta}$ increases. $\qquad\square$

## O    PROOF OF THEOREM 3

We first obtain the following lemma, which extends Theorem 2.7 of (Dong et al., 2021).

**Lemma 6.** *Fix $\alpha \in (0,1)$. Let $g_D$ be Gaussian defined above with each $g_D^j(\cdot|b) \in \Delta(\mathcal{Y}_j)$ as the density function of $\mathcal{N}(\mathtt{M}_b^j, \mathtt{V}^j)$ given any $b \in W$, where $\mathtt{V}^j = \left(2\text{sens}^j(f)/\widehat{\mathtt{M}}_j\right)^2$. Let $P_b(g_D^j)$ denote the probability distribution associated with $g_D^j(\cdot|b)$. Suppose $\max_{b,b'} |\mathtt{M}_b^j - \mathtt{M}_{b'}^j| \leq \text{sens}^j(f)$. Then, it holds*

$$T\left[P_b(g_D^j), P_{b'}(g_D^j)\right](\alpha) \geq T\left[\mathcal{N}(0,1), \mathcal{N}(\widehat{\mathtt{M}}_j, 1)\right].$$

*Proof.* For any two $b, b' \in W$, $y(b) = f_j(b,d) + \delta_j$ and $y(b') = f_j(b', d') + \delta_j'$ are normally distributed with means $f^j(b,d) + \mathtt{M}_b$ and $f^j(b', d') + \mathtt{M}_{b'}$, respectively, and a common variance $\mathtt{V}^j$. Then, we have

$$T\left[P_b(g_D^j), P_{b'}(g_D^j)\right](\alpha) = T\left[\mathcal{N}\left(f^j(b,d) + \mathtt{M}_b, \mathtt{V}\right), \mathcal{N}\left(f^j(b', d') + \mathtt{M}_{b'}, \mathtt{V}\right)\right](\alpha)$$

$$= \Phi\left(\Phi^{-1}(1 - \alpha) - \frac{|f^j(b,d) - f^j(b', d') + \mathtt{M}_b - \mathtt{M}_{b'}|}{\sqrt{\mathtt{V}^j}}\right),$$

where $\Phi$ is the cumulative distribution function (CDF) of the standard normal distribution. Since $\mathtt{V}^j = \left(2\text{sens}^j(f)/\widehat{\mathtt{M}}_j\right)^2$ and $\max_{b,b'} |\mathtt{M}_b^j - \mathtt{M}_{b'}^j| \leq \text{sens}^j(f)$, by definition of sensitivity, we obtain

$$T\left[\mathcal{N}\left(f^j(b,d) + \mathtt{M}_b, \mathtt{V}\right), \mathcal{N}\left(f^j(b', d') + \mathtt{M}_{b'}, \mathtt{V}\right)\right](\alpha) \geq \Phi\left(\Phi^{-1}(1-\alpha) - \widehat{\mathtt{M}}_j\right)$$

$$= T\left[\mathcal{N}(0,1), \mathcal{N}(\widehat{\mathtt{M}}_j, 1)\right](\alpha).$$

$$\square$$

Lemma 6 shows that distinguishing between $b$ and $b'$ is as hard as distinguishing between $\mathcal{N}(0,1)$ and $\mathcal{N}(\widehat{\mathtt{M}}_j, 1)$. Thus, if the $\alpha$-LRT attacker only observes $y_j$ for $j$th SNV, then the maximum power he can obtain is $1 - T\left[\mathcal{N}(0,1), \mathcal{N}(\widehat{\mathtt{M}}_j, 1)\right](\alpha)$, which leads to the WCPL for the vNM defender among all possible powers

when different membership vectors are realized. considered are independent, $1 - T\left[\mathcal{N}(0,1), \mathcal{N}(\widehat{\mathbb{M}}_j, 1)\right](\alpha)$ serves as the performance bound for every $j \in Q$.

Given any two $b, b' \in W$, define the hypothesis testing problem: $H_0$ : the membership vector is $b$ versus $H_1$ : the membership vector is $b'$. From the assumption of independent SNVs, we can obtain the log-likelihood statistics

$$\mathtt{lrs}(y; g_D, b, b') \equiv \sum\nolimits_{j \in Q} \log\left(\frac{\rho_D^j(y_j | H_0)}{\rho_D^j(y_j | H_1)}\right).$$

Let $P_i[\cdot | g_D]$ denote the probability distribution associated with $H_i$ for $i \in \{0, 1\}$.

**Lemma 7.** *Fix $\alpha \in (0,1)$. Let $g_D$ be Gaussian defined above with each $g_D^j(\cdot | b) \in \Delta(\mathcal{Y}_j)$ as the density function of $\mathcal{N}(\mathbb{M}_b^j, \mathbb{V}^j)$ given any $b \in W$, where $\mathbb{V}^j = \left(2\mathrm{sens}^j(f)/\widehat{\mathbb{M}}_j\right)^2$. Suppose $\max_{b,b'} |\mathbb{M}_b^j - \mathbb{M}_{b'}^j| \leq \mathrm{sens}^j(f)$. Then, it holds for all pair $b, b' \in W$,*

$$\max_\tau P_1\left[\mathtt{lrs}(\tilde{y}; g_D, b, b') \geq \tau | g_D\right] \leq 1 - T\left[\mathcal{N}(0,1), \mathcal{N}\left(\sqrt{\sum\nolimits_{j \in Q} \widehat{\mathbb{M}}_j^2}, 1\right)\right](\alpha), \tag{14}$$

*with $P_0\left[\mathtt{lrs}(\tilde{y}; g_D, b, b') < \tau | g_D\right] = \alpha$.*

*Proof.* Since the SNVs are independent, the joint probability density $P(y|H_i)$ over $\mathcal{Y}$ that is equal to the product $\prod_{j \in Q} \rho_D^j(y_j | H_i)$ for $i \in \{0, 1\}$. It is a $|Q|$-fold composition of $\{\rho_D^j\}_{j \in Q}$, where each $\rho_D^j$ accesses to the same dataset. In addition, $\max_\tau P_1\left[\mathtt{lrs}(\tilde{y}; g_D, b, b') \geq \tau | g_D\right]$ is the power of $\alpha$-LRT given $g_D$ for any $b, b' \in W$. Then, (14) follows Corollary 3.3 of Dong et al. (2021). $\qquad\square$

Let $\mathbf{I}_{|Q|}$ denote a $|Q| \times |Q|$ identity matrix. Let $\widehat{\mathbf{M}} \equiv (\widehat{\mathbb{M}}_1, \ldots, \widehat{\mathbb{M}}_{|Q|})$. Consider two multivariate normal distribution $\mathcal{N}(0, \mathbf{I}_{|Q|})$ and $\mathcal{N}(\widehat{\mathbf{M}}, \mathbf{I}_{|Q|})$. Here, $\mathcal{N}(0, \mathbf{I}_{|Q|})$ is rotation invariant, and $\mathcal{N}(\widehat{\mathbf{M}}, \mathbf{I}_{|Q|})$ can be rotated to $\mathcal{N}\left(\sqrt{\sum_{j \in Q} \widehat{\mathbb{M}}_j^2}, 1\right)$. In addition, the rotation here is an invertible transformation. Therefore, $T\left[\mathcal{N}(0,1), \mathcal{N}\left(\sqrt{\sum_{j \in Q} \widehat{\mathbb{M}}_j^2}, 1\right)\right](\alpha)$ is the same as the $T\left[\mathcal{N}(0, \mathbf{I}_{|Q|}), \mathcal{N}(\widehat{\mathbf{M}}, \mathbf{I}_{|Q|})\right](\alpha)$ for any $\alpha$ because the trade-off function is invariant under invertible transformations Dong et al. (2021). Let $\widehat{\beta} = T\left[\mathcal{N}(0, \mathbf{I}_{|Q|}), \mathcal{N}(\widehat{\mathbf{M}}, \mathbf{I}_{|Q|})\right](\alpha)$. Thus, the $\alpha$-LRT with the LR statistics formulated by $\mathcal{L}(y)$ has the power $1 - \widehat{\beta}$. Therefore, it holds that $\mathcal{F}(\alpha, \widehat{\beta}) = |Q|$.

Now, let us focus on when the attacker (either Bayesian or LRT) targets a specific individual $k$. Given any subjective prior $\sigma$ and $Q$, let $\mu_{1|0}^\sigma[|Q|] = \int_r \mu_\sigma(s_k = 1 | r) \rho_D(r | b_k = 0)$. By Proposition 7, a Bayesian attacker's strategy that mirrors the distribution of the posterior belief leads to the WCPL for the defender. Hence, $\mu_{1|0}^\sigma[|Q|]$ captures the highest Type-II errors of the Bayesian attacker. Then, $\mathcal{F}\left(\alpha, \mu_{0|1}^\sigma[|Q|]\right)$ captures the number of SNVs (i.e., $|Q|$) so that $\alpha$-LRT can attain the power $\mu_{0|1}^\sigma[|Q|]$ when the set $Q$ of SNVs of each individual are used in the dataset, leading to $L(g_D, \tau^*, \alpha) = L^\sigma(g_D)$. If $\mathcal{F}\left(\alpha, \mu_{0|1}^\sigma[|Q|]\right) \geq |Q|$, then more SNVs needs to be used to make $\alpha$-LRT have the power $\mu_{0|1}^\sigma[|Q|]$. This is equivalent to $\widehat{\beta} < \mu_{0|1}^\sigma[|Q|]$, which implies $L(g_D, \tau^*, \alpha) \leq L^\sigma(g_D)$. $\qquad\square$

# P  EXPERIMENT DETAILS

## P.1  DATASET

Our experiments use three datasets: *Adult dataset* (UCI Machine Learning Repository), *MNIST dataset*, and *genomic dataset*. The genomic dataset was provided by the 2016 iDASH Workshop on Privacy and Security Tang et al. (2016), derived from the 1000 Genomes Project 1000 Genomes Project Consortium et al. (2015). The genomic dataset used in our experiments was initially provided by the organizers of the 2016 iDash Privacy and Security Workshop Tang et al. (2016) as part of their challenge on Practical Protection of Genomic Data Sharing Through Beacon Services. In this research, we follow Venkatesaramani et al. (2021; 2023) and employ SNVs from chromosome 10 for a subset of 400 individuals to construct the Beacon, with another 400 individuals excluded from the Beacon. We use 800 individuals with different numbers of SNVs of each individual on Chromosome 10. In the experiments, we randomly select 400 individuals from the 800 to constitute a dataset according to the uniform distribution. The experiments were conducted using an NVIDIA A40 48G GPU. PyTorch was used as the deep learning framework.

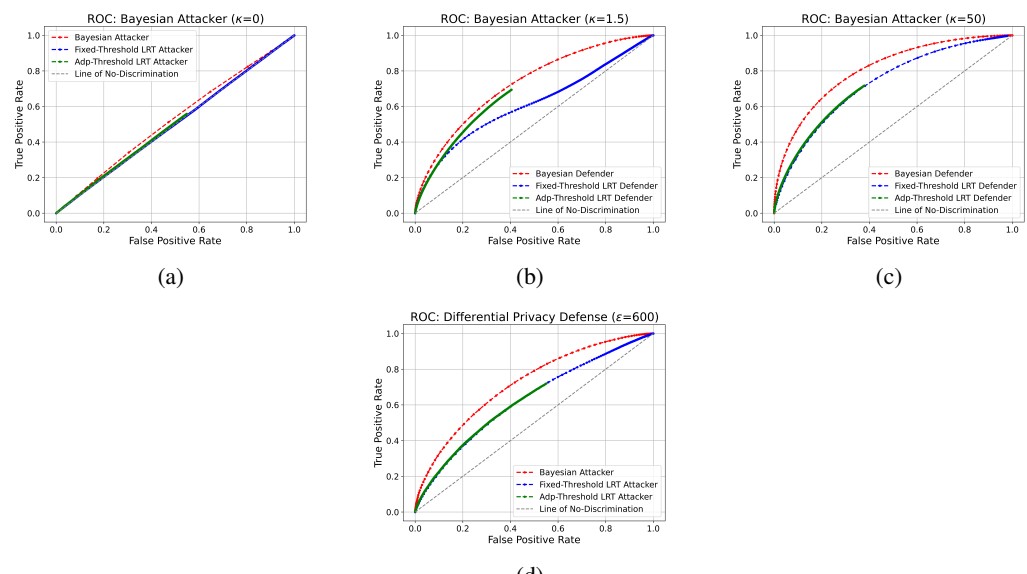

(a)          (b)          (c)

(d)

Figure 2: (a)-(c): Bayesian Defender with $\kappa = 0, 1.5, 50$, respectively. (d): Different attackers under non-strategic DP with $\epsilon = 600$.

## P.2    NOTES: EXPERIMENT DETAILS FOR LRTS

The output of the defender's neural network $G_{\lambda_D}$ is a noise term within the range $[-0.5, 0.5]$. We assess the strength of privacy protection using the attacker's ROC curve, converting $H_{\lambda_A}$'s output to binary values $s_k \in \{0, 1\}$ by varying thresholds. A lower AUC indicates stronger privacy protection by $G_{\lambda_D}$. In addition to the proxies from Section 3.1, we use the sigmoid function to approximate the threshold-based rejection rule of the LRT. Specifically, $\mathbf{1}\{\mathtt{lrs}(d_k, x) \leq \tau\}$ is approximated by $1/(1+\exp(-(\tau - \mathtt{lrs}(d_k, x))))$, where $\mathtt{lrs}(d_k, x)$ is the log-likelihood statistic. Similarly, the sigmoid function approximates $\mathbf{1}\{\mathtt{lrs}(d_k, r) \leq \tau^{(N)}(r)\}$. The fixed- and adaptive-threshold LRT defenders optimally select $g_D$ by solving (FixedLRT) and (AdaptLRT), as detailed in Appendix C.

## P.3    NOTES: BAYESIAN DEFENDER VS. DIFFERENTIAL PRIVACY

In this experiment (Figure 1c), we illustrate the advantages of the Bayesian defender (i.e., using the BNGP strategy) over standard DP in addressing defender-customized objectives for the privacy-utility trade-off, when the same utility loss is maintained in the trade-off of privacy and utility.

In this experiment, we consider a specific loss function for the defender:

$$\ell_D(\delta, b, s) \equiv v(s, b) + \sum\nolimits_{j \in Q} \kappa_j |\delta_j|,$$

where $\kappa_j \geq 0$ represents the defender's preference for balancing the privacy-utility trade-off for the summary statistics of the $j$-th attribute (e.g., SNV in genomic data). In genomic datasets, each SNP position corresponds to a specific allele at a particular genomic location, and the importance of these positions can vary significantly depending on their association with diseases or traits in medical studies. Consequently, different SNPs may require varying levels of data quality and utility, necessitating less noise for some positions. For SNPs where higher data utility is crucial, we assign larger $\kappa_j$ values to increase the weight of noise costs in the defender's decision-making process. This position-dependent weighting enables a more customized and refined privacy-utility trade-off.

In the experiment, we define $\vec{\kappa} = (\kappa_j)_{j \in Q}$ for SNV positions, where $\kappa_j = 0$ for 90% of the 5000 SNVs and $\kappa_j = 50$ for the remaining 10%. The BNGP strategy in this setting results in an average utility loss of 0.0001.

The sensitivity of the summary statistics function $f(\cdot)$ is given by sensitivity $= \frac{m}{K^\dagger}$ (see Appendix E), where $m = |Q|$ and $1 \leq K^\dagger \leq K$ is the number of individuals in $U$ included in the dataset. For the experiment, the dataset comprises 400 individuals, each with 5000 SNVs, resulting in a sensitivity of $\frac{m}{K^\dagger} = 12.5$.

The scale parameter of the Laplace distribution in the DP framework is:

$$\frac{\text{sensitivity}}{\epsilon}.$$

To match the utility loss of 0.0001 (measured as the expected absolute value of the noise), the scale parameter must equal 0.0001. This implies:

$$\frac{\text{sensitivity}}{\epsilon} = 0.0001,$$

which gives $\epsilon = 1.25 \times 10^5$.

In general, the number of SNVs ($m$) is often much larger than the number of individuals ($K$), i.e., $m \gg K$. Consequently, small $\epsilon$ values (e.g., between 1 and 10) result in very large scale parameters for the Laplace distribution. Therefore, relatively large $\epsilon$ values are chosen to preserve the utility of genomic datasets. For example, in (Venkatesaramani et al., 2023), the values of $\epsilon$ are selected from $\{10,000, 50,000, 100,000, 500,000, 1 \text{ million}, 5 \text{ million}, 10 \text{ million}\}$.

### P.4    NOTES: SCORE-BASED ATTACKER:

Figure 1g compares attackers under the defense trained against the score-based attacker. The Bayesian attacker significantly outperforms the others, achieving near-perfect classification, while LRT and adaptive LRT perform similarly but lag behind. As explained in (Dwork et al., 2015), the score-based attacker is assumed to have less external information and knowledge than the Bayesian, the fixed-threshold LRT, and the adaptive LRT attackers. Theoretically, the score-based attacker uses $\mathcal{O}(n^2 \log(n))$ SNVs, where $n$ is the number of individuals in the dataset. In the experiments for Figure 1g, to guarantee certain accuracy for the score-based attacker, we consider 20 individuals and each time a dataset of 5 individuals with 4000 SNVs being randomly sampled. In this setting, the Bayesian attacker performs very well (with AUC close to 1).

### P.5    NOTES: ADULT AND MNIST DATASET

**Adult Dataset**    In the experiments using the Adult dataset, the original mechanism $f$ (i.e., without privacy protection) releases the summary statistics of the Adult dataset. Specifically, we turn the attributes of the Adult dataset into binary values to simplify the representation of categorical and continuous attributes. For example, categorical attributes like "occupation" or "education level" are one-hot encoded, while continuous attributes like "age" are discretized into binary intervals. This binary transformation allows us to construct a dataset that represents the presence or absence of specific attribute values, making it compatible with our framework for privacy protection and utility optimization. The summary statistics released include the counts or proportions of individuals possessing specific binary attributes. These statistics form the basis for evaluating the membership inference risks and utility trade-offs in our experiments. By using this transformed representation, we ensure the methodology aligns with the assumptions of our privacy-utility trade-off framework.

**MNIST Dataset**    For the MNIST dataset, the original mechanism is a trained classifier that outputs predicted class probabilities for given input images. Specifically, this classifier is trained on the MNIST training set to perform digit recognition, mapping each image to a probability distribution over the 10-digit classes (0 through 9). In our experiments, we consider the privacy of the test data (or inference dataset) used to query the classifier. The attacker aims to infer whether a specific test image belongs to the inference dataset based on the output probabilities provided by the classifier.

### P.6    NOTES: BGBP RESPONSE

The BGBP response acts as a constraint for the *defender* since the defender's choice of $G$ induces $H$, which (1) represents the attacker's best response, and (2) satisfies the conditions defined by $\text{PH}[\vec{G}; \epsilon]$. The attacker, however, simply responds optimally to the defender's choice of $G$. To incorporate the conditions for $H$ set by $\text{PH}[\vec{G}; \epsilon]$, we apply the penalty method to the defender's loss function. In our experiments, we relax the strict pure differential privacy framework and focus on a class of neural networks $G$ that select $\epsilon$ for a Gaussian distribution $\mathcal{N}(0, \text{Var}(\epsilon))$, where $\text{Var}(\epsilon) = \text{C}/\epsilon^2$ and $\text{C}$ is a fixed constant. For the *composition* of five mechanisms, four are pre-designed with noise perturbation using $\mathcal{N}(0, \text{Var}(\epsilon))$. The defender's neural network $G_5$ selects $\epsilon$ for the fifth mechanism, constrained by the BGBP response with $\text{BDP}[\vec{G}; 5\epsilon]$. We evaluate whether the target $5\epsilon$ can be approximately achieved if the attacker's performance aligns closely with that of a single $5\epsilon$-DP mechanism (*One Mechanism*), where the single $5\epsilon$-DP mechanism is also perturbed by Gaussian noise $\mathcal{N}(0, \text{Var}(5\epsilon))$. Our experimental results demonstrate that the BNGP strategy, constrained by BGBP response, successfully implements parameterized privacy in a generative manner.

## P.7 ADDITIONAL EXPERIMENT:

As shown in Figure 2, the privacy strength of the defense decreases (resp. increases) as $\kappa$ increases (resp. decreases), as we would expect, since $\kappa$ captures the tradeoff between privacy and utility. Figure 2d demonstrates the performances of the Bayesian, fixed-threshold, and adaptive-threshold attackers under $\epsilon$-DP defense where $\epsilon = 600$. The choice of such a large value of $\epsilon$ is explained in Appendix E). Similar to the scenarios under the Bayesian defense, the Bayesian attacker outperforms the LRT attackers under the $\epsilon$-DP.

## P.8 NETWORK CONFIGURATIONS AND HYPERPARAMETERS

The **Defender** neural network is a generative model designed to process membership vectors and produce beacon modification decisions. The input layer feeds into two fully connected layers with batch normalization and activation functions applied after each layer. The first hidden layer uses ReLU activation, while the second hidden layer uses LeakyReLU activation. The output layer applies a scaled sigmoid activation function. The output of the Defender neural network is a real value between -0.5 and 0.5, which is guaranteed by the scaled sigmoid activation function. All Defender neural networks were trained using the Adam optimizer with a learning rate of 0.001, weight decay of 0.00001, and an ExponentialLR scheduler with a decay rate of 0.988.

The **Attacker** neural network is a generative model designed to process beacons and noise to produce membership vectors. The input layer feeds into two fully connected layers with batch normalization and activation functions. The first hidden layer uses ReLU activation. The output layer applies a sigmoid activation function. All Attacker models were trained using the Adam optimizer, a learning rate of 0.0001, weight decay of 0.00001, and an ExponentialLR scheduler with a decay rate of 0.988.

The specific configurations for each model are provided in the tables below. Table 1a shows the configurations of the neural network Defender under the Bayesian, the fixed-threshold, and the adaptive-threshold attackers when the trade-off parameter $\kappa$ is a vector (i.e., each $\kappa_j = \kappa$ for all $j \in Q$). Table 1b shows the configurations of Defender when the trade-off parameter is a vector; i.e., $\vec{\kappa} = (\kappa_j)_{j \in Q}$ where $\kappa_j = 0$ for the 90% of 5000 SNVs and $\kappa_j = 50$ for the remaining 10%. Table 2a lists the configurations of the neural network Attacker under the Bayesian, the fixed-threshold LRT, and the adaptive-threshold LRT defenders. Table 2b lists the configurations of Attacker under the standard $\epsilon$-DP which induces the same $\vec{\kappa}$-weighted expected utility loss for the defender.

### Table 1: Bayesian Defender Configurations

(a) Defender with scalar $\kappa$

| Layer | Input Units | Output Units |
|---|---|---|
| Input Layer | 830 | 1500 |
| Hidden Layer 1 | 1500 | 1100 |
| Hidden Layer 2 | 1100 | 500 |
| Output Layer | 500 | 5000 |

(b) Defender with vector $\vec{\kappa}$

| Layer | Input Units | Output Units |
|---|---|---|
| Input Layer | 830 | 1000 |
| Hidden Layer 1 | 1000 | 3000 |
| Hidden Layer 2 | 3000 | 4600 |
| Output Layer | 4600 | 5000 |

### Table 2: Attacker Configurations

(a) Attacker vs. Defender

| Layer | Input Units | Output Units |
|---|---|---|
| Input Layer | 5000 | 3400 |
| Hidden Layer 1 | 3400 | 2000 |
| Output Layer | 2000 | 800 |

(b) Bayesian Attacker vs. $\epsilon$-DP

| Layer | Input Units | Output Units |
|---|---|---|
| Input Layer | 5000 | 3000 |
| Hidden Layer 1 | 3000 | 1000 |
| Output Layer | 1000 | 800 |

## P.9 COMPLEXITY ANALYSIS OF THE NEURAL NETWORKS

Here, we use `snp_dim` to denote the number of single-nucleotide variants (SNVs) per data point and `ind_dim` to represent the total number of individuals.

We provide a complexity analysis of the Attacker ($D$) and Defender ($G$) neural networks used in our general-sum GAN. This analysis covers the trainable parameters and computational complexity for both networks.

The **Attacker** takes input of dimension `snp_dim` and produces a membership vector of dimension `ind_dim`. Its architecture consists of fully connected layers, batch normalization, and activation functions. Specifically, the first linear layer maps the concatenated input to a hidden layer of dimension `Hidden_Layer_1_dim`, followed by another linear layer reducing the dimensionality to `Hidden_Layer_2_dim`, a batch normalization layer, a ReLU activation function, and a final linear layer mapping to `ind_dim` with a Sigmoid activation function for output.

The total number of trainable parameters in the Attacker is derived as follows. The first linear layer has $(\texttt{snp\_dim}) \cdot \texttt{Hidden\_Layer\_1\_dim}$ weights and `Hidden_Layer_1_dim` biases. The second linear layer includes $\texttt{Hidden\_Layer\_1\_dim} \cdot \texttt{Hidden\_Layer\_2\_dim}$ weights and `Hidden_Layer_2_dim` biases, while the batch normalization layer adds $\texttt{Hidden\_Layer\_2\_dim} \cdot 2$ scale and shift parameters. The final linear layer contributes $\texttt{Hidden\_Layer\_2\_dim} \cdot \texttt{ind\_dim}$ weights and `ind_dim` biases. Therefore, the total number of trainable parameters in the Attacker is:

$$\texttt{snp\_dim} \cdot \texttt{Hidden\_Layer\_1\_dim} + \texttt{Hidden\_Layer\_1\_dim} \cdot \texttt{Hidden\_Layer\_2\_dim}$$
$$+ \texttt{Hidden\_Layer\_2\_dim} \cdot 2 + \texttt{Hidden\_Layer\_2\_dim} \cdot \texttt{ind\_dim} + \texttt{ind\_dim}.$$

For computational complexity during a forward pass, the dominant operations occur in the linear layers, leading to a total complexity of:

$$\mathcal{O}\big(B \cdot [\texttt{snp\_dim} \cdot \texttt{Hidden\_Layer\_1\_dim} + \texttt{Hidden\_Layer\_1\_dim} \cdot \texttt{Hidden\_Layer\_2\_dim}$$
$$+ \texttt{Hidden\_Layer\_2\_dim} \cdot \texttt{ind\_dim}]\big),$$

where $B$ is the batch size.

The **Defender** network takes input of dimension $\texttt{ind\_dim} + \texttt{noise\_dim}$ and produces an output of dimension `snp_dim`. Its architecture comprises multiple fully connected layers, batch normalization layers, and activation functions. The first linear layer maps the input to a hidden layer of dimension `Hidden_Layer_1_dim`, followed by a second linear layer reducing the dimensionality to `Hidden_Layer_2_dim`. Batch normalization and ReLU activation are applied at this stage. A third linear layer further reduces the dimensionality to `Hidden_Layer_3_dim`, followed by another batch normalization layer and a LeakyReLU activation. Finally, the output layer maps the representation to `snp_dim` with a ScaledSigmoid activation function for output.

The total number of trainable parameters in the Defender is as follows. The first linear layer contributes $(\texttt{ind\_dim} + \texttt{noise\_dim}) \cdot \texttt{Hidden\_Layer\_1\_dim}$ weights and `Hidden_Layer_1_dim` biases. The second linear layer includes $\texttt{Hidden\_Layer\_1\_dim} \cdot \texttt{Hidden\_Layer\_2\_dim}$ weights and `Hidden_Layer_2_dim` biases, and the batch normalization layer adds $\texttt{Hidden\_Layer\_2\_dim} \cdot 2$ scale and shift parameters. The third linear layer has $\texttt{Hidden\_Layer\_2\_dim} \cdot \texttt{Hidden\_Layer\_3\_dim}$ weights and `Hidden_Layer_3_dim` biases, while the second batch normalization layer adds $\texttt{Hidden\_Layer\_3\_dim} \cdot 2$ scale and shift parameters. The output layer includes $\texttt{Hidden\_Layer\_3\_dim} \cdot \texttt{snp\_dim}$ weights and `snp_dim` biases. Thus, the total number of trainable parameters is:

$$(\texttt{ind\_dim} + \texttt{noise\_dim}) \cdot \texttt{Hidden\_Layer\_1\_dim} + \texttt{Hidden\_Layer\_1\_dim} \cdot \texttt{Hidden\_Layer\_2\_dim}$$
$$+ \texttt{Hidden\_Layer\_2\_dim} \cdot 2 + \texttt{Hidden\_Layer\_2\_dim} \cdot \texttt{Hidden\_Layer\_3\_dim}$$
$$+ \texttt{Hidden\_Layer\_3\_dim} \cdot 2 + \texttt{Hidden\_Layer\_3\_dim} \cdot \texttt{snp\_dim}$$
$$+ \texttt{snp\_dim}.$$

The forward-pass computational complexity is dominated by the linear layers, resulting in:

$$\mathcal{O}\big(B \cdot [(\texttt{ind\_dim} + \texttt{noise\_dim}) \cdot \texttt{Hidden\_Layer\_1\_dim}$$
$$+ \texttt{Hidden\_Layer\_1\_dim} \cdot \texttt{Hidden\_Layer\_2\_dim} + \texttt{Hidden\_Layer\_2\_dim} \cdot \texttt{Hidden\_Layer\_3\_dim}$$
$$+ \texttt{Hidden\_Layer\_3\_dim} \cdot \texttt{snp\_dim}]\big).$$

In summary, the Attacker and Defender networks are both computationally efficient and scalable. Their forward-pass complexities scale linearly with the batch size and input dimensions, while the number of trainable parameters remains manageable for modern deep-learning hardware. This ensures that the networks are expressive enough for the task while being feasible for practical implementation.

### P.10 AUC Values of ROC Curves with Standard Deviations

Tables 3, 4, and 5 show the AUC values of the ROC curves shown in the plots of the experiments.

Table 3: AUC Values For Different Attackers Under Varying $\kappa$

| Attacker | Figure 2a ($\kappa = 0$) | Figure 1a and 2b ($\kappa = 1.5$) | Figure 2c ($\kappa = 50$) |
|---|---|---|---|
| Bayesian attacker | $0.5205 \pm 0.0055$ | $0.7253 \pm 0.0069$ | $0.8076 \pm 0.0040$ |
| Fixed-Threshold LRT attacker | $0.5026 \pm 0.0062$ | $0.6214 \pm 0.0322$ | $0.7284 \pm 0.0089$ |
| Adaptive-Threshold LRT attacker | $0.1552 \pm 0.0100$ | $0.1716 \pm 0.0144$ | $0.1719 \pm 0.0174$ |

Table 4: AUC Values of Attackers For Figures 1b to 1h

| Figure | Scenarios | AUC $\pm$ std | Condition |
|---|---|---|---|
| 1b | Under Bayesian Defender | $0.7237 \pm 0.0066$ | $\kappa = 1.5$ |
| | Under Fixed-threshold LRT Defender | $0.9124 \pm 0.0026$ | $\kappa = 1.5$ |
| | Under Adaptive-threshold LRT Defender | $0.7487 \pm 0.0027$ | $\kappa = 1.5$ |
| 1c | Under Bayesian Defender | $0.5318 \pm 0.0222$ | $\vec{\kappa}$ |
| | Under $\epsilon$-DP Defender | $0.9153 \pm 0.0025$ | $\vec{\kappa}$ |
| 1d | Bayesian Attacker | $0.5600 \pm 0.0040$ | $\kappa = 1.5$ |
| | Fix-LRT Attacker | $0.5287 \pm 0.0052$ | $\kappa = 1.5$ |
| | Adp-LRT Attacker | $0.1431 \pm 0.0120$ | $\kappa = 1.5$ |
| | Score-Based Attacker | $0.1267 \pm 0.0207$ | $\kappa = 1.5$ |
| 1e | Bayesian Attacker | $0.6317 \pm 0.0050$ | |
| | Fix-LRT Attacker | $0.5865 \pm 0.0060$ | |
| | Adp-LRT Attacker | $0.1722 \pm 0.0752$ | |
| | Score-Based Attacker | $0.1223 \pm 0.0170$ | |
| 1f | Bayesian Attacker | $0.5868 \pm 0.0035$ | |
| | Fix-LRT Attacker | $0.5615 \pm 0.0065$ | |
| | Adp-LRT Attacker | $0.2076 \pm 0.0160$ | |
| | Score-Based Attacker | $0.1229 \pm 0.0028$ | |
| 1g | Bayesian Attacker | $1 \pm 0$ | |
| | Fix-LRT Attacker | $0.8618 \pm 0.0019$ | |
| | Adp-LRT Attacker | $0.2221 \pm 0.0106$ | |
| | Score-Based Attacker | $0.1542 \pm 0.0227$ | |
| 1h | Bayesian Attacker | $0.7422 \pm 0.0085$ | $\kappa = 1.5$ |
| | Decision-Tree Attacker | $0.6609 \pm 0.0110$ | $\kappa = 1.5$ |
| | SVM Attacker | $0.5226 \pm 0.0108$ | $\kappa = 1.5$ |

Table 5: AUC Values of Figure 1i

| Scenarios | AUC $\pm$ std |
|---|---|
| Composition ($\epsilon = 0.05$) | $0.7387 \pm 0.0050$ |
| One Mechanism ($\epsilon = 0.05$) | $0.7427 \pm 0.0063$ |
| Composition ($\epsilon = 0.1$) | $0.8033 \pm 0.0057$ |
| One Mechanism ($\epsilon = 0.1$) | $0.8241 \pm 0.0035$ |
| Composition ($\epsilon = 0.3$) | $0.8921 \pm 0.0037$ |
| One Mechanism ($\epsilon = 0.3$) | $0.9018 \pm 0.0033$ |
| Composition ($\epsilon = 0.6$) | $0.9108 \pm 0.0032$ |
| One Mechanism ($\epsilon = 0.6$) | $0.9201 \pm 0.0032$ |
| Composition ($\epsilon = 1$) | $0.9318 \pm 0.0031$ |
| One Mechanism ($\epsilon = 1$) | $0.9373 \pm 0.0030$ |

