# OpenReview forum: "Bayes-Nash Generative Privacy Protection Against Membership Inference Attacks"
_ICLR.cc/2025/Conference — Submitted to ICLR 2025_

### Official Review · Reviewer_9RQW · 2024-11-03

**Soundness:** 3
**Presentation:** 3
**Contribution:** 3
**Rating:** 6
**Confidence:** 4

**Summary:**

This paper introduces a Bayesian game model for privacy-preserving data sharing, particularly focusing on defending against membership inference attacks (MIAs). The authors propose a GAN-style algorithm to approximate a Bayes-Nash equilibrium, balancing the defender's privacy and utility concerns against an attacker's attempts to maximize privacy leakage. The model incorporates Bayes-Nash generative privacy and Bayes generative privacy risk, accounting for the attacker's heterogeneous preferences towards true and false positives. The method is applied to genomic data summary statistics, demonstrating its effectiveness over state-of-the-art privacy-preserving approaches. The paper also establishes conditions for the equivalence between Bayes-Nash generative privacy and pure differential privacy and shows the composition and post-processing properties of BGP risk. Empirical results validate the theoretical analysis, illustrating the superiority of the Bayesian game-theoretic approach in protecting privacy while maintaining data utility.

**Strengths:**

1. The paper presents a novel approach to privacy protection by modeling the interaction between a defender and an attacker as a Bayesian game, which allows for a more nuanced understanding of privacy risks.
2. The paper introduces a GAN-style algorithm to approximate the Bayes-Nash equilibrium, providing an efficient way to train models in the context of privacy protection.
3.  The paper offers both theoretical analysis and empirical results, demonstrating the effectiveness of the proposed method in protecting privacy while maintaining data utility.
4.  The paper is generally good to follow.

**Weaknesses:**

1. Some notations used in the paper are confusing.
2. The game-theoretical framework of the interplay between attacker and defender needs more justification.
3. The experimental results are thin at the moment and lacks enough illustration.

**Questions:**

1. The derivation of Equation (2) needs more elaboration, what does \theta represent in the equation?
2. Inference cost is considered in the attacker's decision but not the defender's. More justifications are needed for such modeling. What will happen if defense cost is also considered?
3. From the experimental results, it seems that the Bayesian attack proposed in the paper is the best-performed attack. Does different settings of prior information affect the attacking results?
4. What is the difference of experimental settings of results shown in Figure 1a and 1b? According to Figure 1b, does it mean that Bayesian defender is inferior than fixed-threshold LRT defender? (Note that this is an examplar question, I find the results section quite thin and unconvincing).

---

> ### Author Response · Authors · 2024-11-21
> **Response to Reviewer Comments (Part 1)**
>
> We thank the reviewer for their valuable comments and suggestions!
>
>
> > **W1. Some notations are confusing.**
>
> **Response:**
>
> We have taken the following steps to improve clarity:
> 1. **Notation Tables:** In Appendix A, we have included detailed tables of notations for each section to help readers quickly reference and understand the symbols used throughout the paper.
> 2. **Simplification of Notations:** We carefully reviewed and modified the notations throughout the paper based on the reviewer’s suggestions.
>
>
>
> > **W2. The game-theoretical framework of the interplay between attacker and defender needs more justification.**
>
> **Response:**
>
> The game-theoretic framework is particularly well-suited for modeling the interaction between the defender and attacker due to their inherently interdependent strategies. The defender’s strategy determines the level of privacy protection, directly influencing how much sensitive information is preserved or leaked through the mechanism’s output. At the same time, the attacker’s strategy dictates the extent to which sensitive information can be inferred, resulting in privacy risk for the defender.
>
> From a decision-theoretic perspective, the outcomes for both the defender and the attacker are mutually dependent, creating a natural strategic interaction. Game theory provides a rigorous framework to model these interactions, analyze trade-offs, and derive equilibrium strategies that optimally balance privacy and utility. Its extensive application to security and privacy challenges highlights its relevance, particularly in adversarial decision-making and risk management (Do et al., 2017).
>
>
> In our work, the defender’s and attacker’s objective functions, defined in Equations (5) and (3) respectively, are naturally modeled as a Bayesian game. The resulting equilibrium captures the worst-case privacy scenario for the defender by explicitly accounting for the rational responses of the attacker.
>
>
> *Do et al. (2017). Game theory for cyber security and privacy. ACM Computing Surveys, 50(2). DOI: 10.1145/3057268.*
>
>
> > **W3. (also Q4) Experimental results are thin and unconvincing. What is the difference of experimental settings of results shown in Figure 1a and 1b? According to Figure 1b, does it mean that Bayesian defender is inferior than fixed-threshold LRT defender?**
>
> **Response:**
>
>
> We have thoroughly revised Section 5 (Experiments) to clearly explain the purpose and conclusions of each figure.
>
> In the experiments, the ROC curve measures the performance of the attacker’s MIAs. The closer the ROC curve is to the upper-left corner, the better the MIA performs, indicating a higher true positive rate (TPR) and a lower false positive rate (FPR). This corresponds to greater privacy loss and weaker privacy protection.
>
> Figure 1a compares the defender’s performance against three attackers: Bayesian, fixed-threshold LRT, and adaptive LRT attackers, using the genomic dataset. Please note that there are typos in the original legends: the labels should indicate “Attacker” instead of “Defender,” and the title should read “Defender.”
>
> Figure 1b shows the Bayesian attacker’s performance across three scenarios, where the mechanism is protected by different defense models. Specifically, the Bayesian defender employs the BNGP strategy, the fixed-threshold LRT defender adopts a strategy by best responding to the fixed-threshold LRT attacker, and the adaptive LRT defender uses a strategy by best responding to the adaptive LRT attacker.
>
> As shown in Figure 1b, the ROC of the attacker under the fixed-threshold LRT defender is closest to the upper-left corner, indicating that the Bayesian attacker performs best in this case. This means the fixed-threshold LRT defender offers the weakest privacy robustness and is outperformed by the Bayesian defender using the BNGP strategy. Contrary to the reviewer’s concern, the Bayesian defender is not inferior but rather demonstrates superior privacy protection.

---

> ### Author Response · Authors · 2024-11-21
> **Response to Reviewer Comments (Part 2)**
>
> **Questions**
>
>
> > **Q1: The derivation of Equation (2) needs more elaboration, what does \theta represent in the equation?**
>
> **Response:**
>
> Equation (2) introduces the Bayes-weighted membership advantage (BWMA) which is an extension of the standard per-individual membership advantage (MA) given by Equation (1).
> Here, $\theta$ is the prior distribution of the membership information. This extension captures the following additional components of the attacker's decision-making:
> 1. **Prior Knowledge:** BWMA leverages the prior distribution $\theta$. This setting allows the model to account for Bayesian decision-making.
> 2. **Heterogenerous Preferences:** BWMA introduces a parameter $0<\gamma\leq 1$ to weight the trade-off between the TPR and the FPR. Smaller values of $\gamma$ reflects a stronger preference for maximizing TPR, while larger values prioritize minimizing FPR. When $\lambda=0.5$, the attacker values TPR and FPR equally.
>
>
>
> > **Q2: Inference cost is considered in the attacker's decision but not the defender's. More justifications are needed for such modeling. What will happen if defense cost is also considered?**
> >
>
> **Response:**
>
> Thank you for this question. We indeed account for the defender’s corresponding cost through their total expected loss, which consists of two key components: privacy loss and utility loss. This formulation reflects the inevitable trade-off between privacy and utility in privacy-preserving strategies that the defender aims to optimize. Our proposed privacy protection framework aims to optimally balance the privacy-utility trade-off.
>
> Our approach and theoretical results are general and apply to a wide range of defender's objective functions. These objective functions can be customized to different applications and scenarios, allowing the defender to incorporate various priorities and costs. The only assumption imposed on the defender’s objective function is summarized in Assumption 1, which ensures the relationship between privacy risk and the defender’s loss is meaningful and realistic.
>
>
> > **Q3: Does different settings of prior information affect the attacking results?**
>
> **Response:**
>
> We thank the reviewer for this insightful question.
>
> The Bayesian attack, using BGP responses, is indeed the best-performing privacy attack under the given privacy strategy. Our focus on the worst-case scenario privacy risk aims to enhance the robustness of the privacy strategy, aligning with similar worst-case considerations in other privacy frameworks such as differential privacy or Pufferfish privacy.
>
> Yes, different prior information affects the performance of attackers and the resulting privacy risks. Specifically, the attacker’s prior knowledge shapes their inference capabilities, imposing varying levels of privacy risk for the defender. Our framework explicitly accounts for these differences by supporting scenarios where the attacker holds diverse subjective priors. This enables a comprehensive analysis of privacy risks under various assumptions about the attacker’s knowledge. Such flexibility is particularly relevant when the defender is aware of prior privacy risks and can adapt their privacy protection strategies accordingly.
>
> In information security and privacy, addressing key considerations such as the extent to which information must remain hidden, what can be compromised, and what has already been compromised is essential. These considerations align with the goals of achieving accurate and tight characterizations of privacy auditing, accounting, and composition, which ensure effective privacy budget allocation, enhance awareness of privacy risks, and enable adaptive strategies to balance utility and security.
>
> The subjective prior reflects information about what has already been compromised. Our formulation involving subjective priors captures two key scenarios: (1) when the prior privacy risk (i.e., the attacker’s knowledge) is known or evaluated by the defender, and (2) when the privacy risk is accounted for by the defender, such as through knowledge of the number or type of data accesses that have already occurred. This knowledge can be systematically incorporated into the analysis via subjective priors, akin to updated priors in sequential Bayesian updates.
>
> Our proposed game-theoretic BNGP strategy leverages this subjective prior to achieve an optimal balance in the privacy-utility trade-off by addressing core considerations around hiding sensitive information, managing necessary compromises, and mitigating privacy risks effectively.
>
>
>
>
>
> > **Q4: What is the difference of experimental settings of results shown in Figure 1a and 1b? According to Figure 1b, does it mean that Bayesian defender is inferior than fixed-threshold LRT defender? (Note that this is an examplar question, I find the results section quite thin and unconvincing).**
>
> **Response:**
>
> Since Q4 is closely related to comment W3, we have addressed this question in our response to W3 above.

---

> > ### Comment · Reviewer_9RQW · 2024-12-02
> >
> > I appreciate the responses and revisions from the authors. I have raised my rating to 6.

---

> > > ### Author Response · Authors · 2024-12-02
> > > **Thank you**
> > >
> > > We sincerely thank the reviewer for their thoughtful feedback, which has helped us improve our paper. We also greatly appreciate the updated rating.

---

### Official Review · Reviewer_FHZn · 2024-11-05

**Soundness:** 2
**Presentation:** 2
**Contribution:** 3
**Rating:** 5
**Confidence:** 3

**Summary:**

The paper proposes a game-theoretic framework for private generative models. Privacy risk is measured by vulnerability to membership-inference attacks. The defender is a randomized generative model that processes a dataset drawn by a population by a neural network. The defender. The attacker is modelled as a Bayesian agent with a membership prior that infers membership by applying a discriminator neural network to output generated by the defender. The paper describes attacker and defender losses that are minimized simultaneously using GAN techniques.

**Strengths:**

The paper proposes a novel framework for generative privacy and experimental results demonstrate that Bayesian membership-inference attacks in the framework can be uniformly stronger than a frequentist attack.

**Weaknesses:**

The paper is difficult to read due to cumbersome notation and a lack of motivating exposition.

In particular, the paper contains a high number of symbols such as $\hat{\mathcal{L}}^\sigma_A(G_{\lambda_D}, H_{\lambda_A})$ and $\textrm{BN}[G; \sigma]$ that are complex and somewhat difficult to memorize. I found this to distract from the important concepts presented in the paper. To give a couple of examples:
- You could consider fixing some values like the distributions $\sigma$ and $\theta$ at some conspicuous location in the text (since they do not appear to be treated as variables outside of e.g. Prop 1) and then replacing symbols like $\textrm{Adv}(h_A, \sigma, \theta, \gamma; g'_D)$ with something like $\textrm{Adv}^\gamma(h_A, g'_D)$.
- Some notation like $\mathcal{L}^\sigma_A(G_{\lambda_D}, H_{\lambda_A}; \gamma)$ seems to have more "moving parts" than necessary and could be replaced with something like $\hat{\mathcal{L}}^\gamma_A(G, H)$ (assuming you need to maintain $\gamma$ as a variable)
- There are many sets such as $\texttt{Br}[G; \sigma, \gamma]$ introduced, which require extra back-and-forth to recall their meaning. They could be replaced by a phrase "$H^*$ is a best response to $G$" or by a short formula like $H^* \in \underset{H}{\operatorname{argmin}}  \mathcal{L}^\sigma_A(G, H)$.

In addition, a number of main results such as Theorem 1 are difficult to interpret due a lack of clear explanation. It would be helpful to expand a bit on what "a BNGP mechanism using CEL ensures robust privacy protection" (l299) means. As another example, Definition 4 felt a bit unclearly motivated and a bit difficult to parse. The exposition given for Proposition 5 could also be clarified.

Overall, the paper would benefit from broad notational simplification as well as clear intuitive exposition for technical definitions and theorems.

**Questions:**

l247: "Therefore, we use $\ell_U(\Vert \delta \Vert_p)$ as the utility loss for the defender." Does this account for the effect of clipping?

l269: "This equilibrium is a reformulation of the $\sigma$-BNE using neural networks." Are we guaranteed that the equilibrium exists for typical classes of neural networks?

l295: Theorem 1 assumes that "that given any $G$, $\mathcal{L}_D$ increases as the TPR or the $\textrm{Adv}(H,\sigma,\theta,0.5; G)$ increases." Can the authors explain this assumption a bit further and speak to whether it is realistic?

l445: Figure 1(a) shows a non-concave tradeoff curve (blue) for a likelihood-ratio test attacker. This is surprising to me because Neyman-Pearson ensures the LRT attacker realizes the tradeoff curve of the mechanism and these curves must be concave/convex (see e.g. Proposition 2.2 in Dong, Roth, and Su 2019). How can this be explained?

The following are small typographical issues:
- l63: abbreviation CEL used before definition
- l74: LRT not defined yet
- l93.5: "quantify" -> "quantifies"
- l223.5: "differentable"
- l262-263: $\mathcal{L}_D$ defined for the second time; $\mathcal{L}_A^\sigma$ seems to conflict with existing notation $\mathcal{L}_A$ from l202
- l268: it is a bit confusing to me to use "risk" to refer to a discriminator network rather than a scalar quantity
- l445: Some of the legends in Figure 1 mention "defender" but I assume that "attacker" is what is meant.

---

> ### Author Response · Authors · 2024-11-21
> **Response to Reviewer Comments (Part 1)**
>
> We thank the reviewer for their valuable comments and suggestions!
>
> > **W1. Cumbersome notation and Lack of Motivating Exposition.**
> >
> **Response:**
>
> We thank the reviewer for the valuable feedback on the notations and their impact on readability.
>
> Regarding the notations, we have taken the following steps to improve clarity:
> 1. **Notation Tables:** In Appendix A, we have included detailed tables of notations for each section to help readers quickly reference and understand the symbols used throughout the paper.
> 2. **Simplification of Notations:** We carefully reviewed and modified the notations throughout the paper based on the reviewer’s suggestions. We sincerely appreciate your valuable feedback and suggestions!
>
>
> Regarding the motivating exposition, we have taken the following steps:
> 1. we have significantly improved the introduction to enhance its accessibility and clarity for readers unfamiliar with the concepts in this paper. The revised introduction now follows a structured approach: it clearly outlines the problem, highlights the motivation (Line 31-Line 53), introduces our proposed approach, and conveys the key conceptual contributions (Line 66-Line 81).
> 2. Specifically, the revised text clearly highlights how the Bayes-Nash Generative Privacy (BNGP) strategy addresses key limitations of standard differential privacy frameworks, including avoiding intractable sensitivity calculations, circumventing worst-case proofs for compositions, and enabling compositions of mechanisms with arbitrary correlations.
> 3. We have provide explanations for the main results including theorems and propositions.
>
>
> > **W2. Explain Main Results Theorem 1**
>
> **Response:**
>
> We appreciate the reviewer's feedback regarding the clarity of the main results and their explanations. We have thoroughly revised and enhanced the presentation of our results and related definitions.
>
> For example, we have slightly modify the Theorem 1 (without changing the original result) and add the following explanation (starting from Line 310 in the revised version):
>
> *"By definition, a BNGP strategy $G^{\star}$ responds to the BGP response $H^{\star}$, ensuring the optimal privacy-utility trade-off by considering the worst-case privacy loss when the attacker minimizes $\mathcal{L} _{\textup{CEL}}$.  It is important to note that $\mathtt{TPR} ,  \mathtt{Adv}^\gamma ,  \widetilde{\mathcal{L}}_D$  are independent of $\mathcal{L} _{\textup{CEL}}$. Theorem 1 establishes that $G^{\star}$ achieves the optimal privacy-utility trade-off given $\widetilde{\mathcal{L}} _{D}$ by leveraging the worst-case privacy risk under the chosen privacy strategy. Specifically, the first inequalities in (i) and (ii) show that, under $G ^{\star}$, an attacker using $H ^{\star}$ achieves the worst-case privacy risk for the defender, and no other attacker can induce a strictly higher privacy loss in terms of $\mathtt{TPR}$ or $\mathtt{Adv}^{0.5}$. The second inequalities in (i) and (ii) further demonstrate that $G ^{\star}$ minimizes the defender's perceived privacy risk, ensuring that no alternative privacy strategy $G'$ achieves a strictly lower privacy loss against the worst-case attacker."*
>
>
> For Definition 4, we have modified the original "$\epsilon$-Bayes Generative Differential Privacy Risk" to the new "$\epsilon$-Bayes Generative Bounded Privacy Response" ($\epsilon$-BGBP response) and provided the following explanation (starting from Line 386 in the revised version):
>
> *"
> An $\epsilon$-BGBP response satisfies both (i) the conditions of a BGP response and (ii) the linear constraints in $\mathtt{DPH}[\vec{G};\epsilon]$. However, the attacker optimizing $\mathcal{L}_{\textup{CEL}}$ does not consider $\mathtt{DPH}[\vec{G};\epsilon]$ as a constraint in their optimization. In other words, $\mathtt{DPH}[\vec{G};\epsilon]$ is not a restriction on the attacker's strategy. Instead, it is the defender’s choice of \(\vec{G}\) that must ensure the induced attacker's BGP response also satisfies the constraints in $\mathtt{DPH}[\vec{G};\epsilon]$. That is, $\mathtt{DPH}[\vec{G};\epsilon]$ constrains the defender's optimization problem.
> "*
>
> **(Continued on Part 2)**

---

> ### Author Response · Authors · 2024-11-21
> **Response to Reviewer Comments (Part 2)**
>
> >**(Continuation of W2)**
>
> **Response:**
>
> To further facilitate our mitivations and the conceptual messages, we have added a new Proposition 5. The original Proposition 5 now becomes Proposition 6 in the revised version. We have added explanation for Proposition 6 in the revised version starting from Line 394:
>
> *"
> Proposition 6 establishes the necessity and sufficiency of using the BGBP response to implement a pure ($\xi=0$) differentially private mechanism. Consequently, for a composition (or a single mechanism) $\mathcal{M}(\vec{G}^{\star})$ to satisfy $\epsilon$-DP, the defender selects $\vec{G}^{\star}$ based on a given $\epsilon$, ensuring:
> $$
> \vec{G} ^{\star} \in \arg\min _{\vec{G}} \widetilde{\mathcal{L}} _{D}(\vec{G}, \text{s.t. } H ^{\star}), \quad
> H ^{\star} \in \arg\min\nolimits _{H} \mathcal{L} _{\textup{CEL}}(\vec{G}, H) \cap \mathtt{DPH}[\vec{G};\epsilon].
> $$
> This choice of $\vec{G}^{\star}$ guarantees that $\mathcal{M}(\vec{G})$ is an $\epsilon$-DP mechanism that optimally balances the privacy-utility trade-off for a given privacy risk characterized by $\epsilon$.
> "*
>
>
> > **Question 1. Does the utility loss account for clipping?**
>
> **Response:**
>
> We appreciate the reviewer's question. To clarify:
>
> 1. The utility loss function $\ell_U(\|\delta\|_p)$ does not explicitly account for the effect of clipping. However, minimizing $\ell_U(\|\delta\|_p)$ implies minimizing the loss with clipping in our case study of sharing summary statistics of the genomic dataset. Importantly, our approach and the main theoretical results are agnostic to whether clipping is applied. They hold under both clipped and un-clipped scenarios without requiring any specific assumptions tied to either case.
> 2. Our results are designed to accommodate defender-customized objective functions that reflect the defender’s preferences over the privacy-utility trade-off. This flexibility ensures the applicability of our approach across diverse scenarios with varying trade-off requirements.
> 3. We have clearly stated the assumptions (Assumption 1) required for the defender’s objective function to ensure the validity of our approach. For further details, please see the revised paper, starting from Line 233, where these assumptions are explicitly outlined.
>
>
>
> > **Q2. Are we guaranteed that the equilibrium exists for typical classes of neural networks?**
>
> **Response:** Thank you for this queston. The existence of the equilibrium in our framework depends on whether the neural networks used in the general-sum GAN are ideal or parameterized. We address both cases below.
>
> 1. For the ideal, non-parameterized neural networks, the equilibrium exists because these idealized networks can perfrectly represent the true probability distributions. In this case, the BNE is guaranteed to exist because there are no representational constraints.
> 2. For parameterized neural networks with specific classes of parameters, the guarantee of equilibrium existence depends on the representational capacity of the network. Parameterized neural networks operate within a finite-dimensional parameter space, which constrains the set of functions they can represent. In contrast, the true underlying probability distributions often reside in a richer, infinite-dimensional functional space that cannot always fully captured by finite parameterizations.
>
> This introduces an approximation gap between the distributions representable by the parameterized network and the true underlying distributions. This gap is intrinsic to the parameterization itself and persists even in the theoretical limit of perfect optimization and infinite data. The parameterized networks can approximate the equilibrium in practice. However, the existence of a true equilibrium in this constrained setting depends on the alignment between the parameterized network's capacity and the structure of the distributions.

---

> ### Author Response · Authors · 2024-11-21
> **Response to Reviewer Comments (Part 3)**
>
> > **Q3: Can the authors explain this assumption of Theorem 1 a bit further and speak to whether it is realistic?**
> >
>
> **Response:**
>
> We have summarize this assumption by Assumption 1 in the revised version (Line 233):
>
> **Assumption 1.** *For a given $g _{D}$, the defender's expected loss $\mathcal{L} _{D}(g _{D}, h _{A})$ increases as either $\mathtt{TPR}(h _{A}, g _{D})$ or $\mathtt{Adv}^{0.5}(h _{A}, g _{D})$ increases.*
>
> Assumption 1 ensures a meaningful choice of the objective function for privacy-utility trade-off.
>
>
> This assumption ensures a meaningful formulation of the privacy-utility trade-off by establishing a relationship between the defender’s expected loss and the privacy risk, which depends on the attacker’s strategy, under a given privacy strategy $g _{D}$ (or equivalently G). Specifically, it reflects the intuition that as privacy risk increases (i.e., higher $\mathtt{TPR}(h _{A}, g _{D})$ or $\mathtt{Adv}^{0.5}(h _{A}, g _{D})$, depending on how the attacker responds) the defender, who uses a fixed $g _{D}$, incurs greater loss. Consequently, the defender’s goal is to minimize both the privacy risk and the utility loss.
>
>
> A common class of loss functions satisfying Assumption 1 includes those where the defender’s expected loss is expressed as an additive combination of privacy risk and utility loss. It is standard practice to quantify privacy loss using measures such as the attacker’s true positive rate ($\mathtt{TPR}$) or membership advantage ($\mathtt{Adv}$).
>
>
> Furthermore, it is realistic to assume that the utility loss is determined solely by the defender’s privacy protection strategy and remains independent of the attacker’s responses. This is because, for a given privacy strategy $g _{D}$, the utility component of the trade-off is fixed and does not depend on how the attacker responds.
>
>
> > **Q4:Curves in Figure 1(a)**
>
> **Response:**
>
> We sincerely thank the reviewer for pointing out the issue and the typo in Figure 1a. We have corrected the legend, which should indicate "Attacker" instead of "Defender".
>
> Regarding the trade-off curve, you are absolutely correct that under the Neyman-Pearson lemman, a Uniformly Most Powerful (UMP) LRT attacker would produce a ROC curve that is continuous, non-decreasing, and concave.
>
> However, in our experiments, the baseline LRT attackers are fixed-threshold LRT and adaptive-threshold LRT attackers (e.g., Sankararaman et al., 2009; Shringarpure & Bustamante, 2015; Venkatesaramani et al., 2021; 2023). These LRT models are in general not UMP tests for any given privacy protection strategies. As such, their resulting trade-off curves are not guaranteed to satisfy the concavity properties associated with UMP tests.
>
>
>
> **Typographical Issues**
>
> Thank you for pointing our these issues! We have corrected the typoes. Regarding the use of the term "risk" (Line 298, Definition 2), we have changed "Bayes Generative Privacy (BGP) risk" to "**BGP response**".

---

> ### Comment · Reviewer_FHZn · 2024-11-27
>
> Thank you for addressing my points about writing/notational clarity in such detail. The paper is easier to read now. Thank you also for the clarification regarding Figure 1a.
>
> I perhaps did not emphasize this strongly enough in my original review but I remain concerned about the privacy guarantees provided by your framework. Real models do of course have finite representational capacity and thus we don't seem to have a general assurance of convergence to some equilibrium. There does not appear to be a theoretical analysis of the privacy risk due to finite representational capacity of the attacker nor due to incomplete training of the GAN. After all it is not possible to train for an infinite anmount of time with an infinite amount of data.
>
> I also remain unconvinced in the empirical evidence for the strength of your privacy model. Figure 1a does not show the optimal alpha-LRT attacker. The adaptive threshold attacker is also not shown completely, so it doesn't seem possible to fully compare to the Bayesian attacker.
>
> I am also having some trouble understanding the theoretical guarantee of Theorem 2. It is not clear to me how one would set about finding an aligned prior. This seems critical for ensuring a meaningful privacy guarantee.
>
> There is also a connection to existing privacy guarantees of the DP framework through Proposition 6 but this seems to be mostly a connection in definition-only. This result would be strengthened by an algorithm or explanation how to train the network so as to ensure all BGP responses are in fact eps-BGBP. In any case, it is still not clear what improvement is offered over the existing DP framework.

---

> > ### Author Response · Authors · 2024-11-29
> > **Response to Official Comment by Reviewer FHZn (Part 1)**
> >
> > We are glad to hear that the revisions have improved the paper’s readability and that the clarifications regarding Figure 1a were helpful. Below, we provide detailed responses to the remaining comments.
> >
> > > Comment 1: I perhaps did not emphasize this strongly enough in my original review but I remain concerned about the privacy guarantees provided by your framework. Real models do of course have finite representational capacity and thus we don't seem to have a general assurance of convergence to some equilibrium. There does not appear to be a theoretical analysis of the privacy risk due to finite representational capacity of the attacker nor due to incomplete training of the GAN. After all it is not possible to train for an infinite anmount of time with an infinite amount of data.
> >
> > **Response:**
> >
> > We appreciate the reviewer’s thoughtful concern about the finite representational capacity of real-world models and the practical limitations of training GANs. Indeed, in any optimization-based framework, including ours, perfect convergence to a theoretical equilibrium is generally not achievable due to finite data, limited model capacity, and computational constraints.
> >
> >
> > This challenge is not unique to our framework but is inherent to all optimization-based methods, including those leveraging differential privacy. For example, when differential privacy is used as a privacy constraint, finding the optimal mechanism to balance the privacy-utility trade-off is similarly intractable in general and relies on approximations and finite training resources.
> >
> >
> > Balancing privacy and utility in optimization-based frameworks is a complex and challenging problem. In addition to the inherent intractability of solving the underlying optimization problems, providing robust privacy guarantees further complicates the process. Differential privacy (DP) and its variants often rely on adversarial worst-case proofs, which require tightly or analytically characterizing the worst case. This is generally non-trivial, for example:
> >
> > * **Sensitivity calcualtion:** Calculating sensitivity, which determines the noise distribution required for DP, is generally NP-hard (Xiao & Tao, 2008). Even for widely used methods like differentially private neural network training, determining a tight sensitivity bound for stochastic gradient descent remains an open problem.
> > * **Composition:** Calculating the tightest privacy bounds for the composition of independent mechanisms is known to be in general a #P-complete problem (Murtagh & Vadhan, 2015). Furthermore, tightly characterizing the composition of correlated mechanisms remains an open challenge.
> >
> > These challanges make the optimization of privacy-utility trade-offs in DP frameworks highly complex and computationally prohibitive.
> >
> > In contrast, our work provides a relatively simpler approach to optimally balance the privacy-utility trade-off with theoretical guarantees. By avoiding certain challenges of DP (i.e., characterizations of sensitivity and composition), our framework offers robust privacy guarantees while being relatively easier to implement compared to a DP-based privacy strategy.

---

> > ### Author Response · Authors · 2024-11-29
> > **Response to Official Comment by Reviewer FHZn (Part 2)**
> >
> > > Comment 2: I also remain unconvinced in the empirical evidence for the strength of your privacy model. Figure 1a does not show the optimal alpha-LRT attacker. The adaptive threshold attacker is also not shown completely, so it doesn't seem possible to fully compare to the Bayesian attacker.
> >
> > **Response:**
> >
> > **1: Optimal LRT attack model**
> >
> > Thank you for your comment. We have conducted additional experiments to demonstrate the results of the optimal LRT attack model. Please find the updated plot at this [anonymized link](https://zenodo.org/records/14238639).
> >
> > In our experiments, we consider the true prior $\theta$ to be uniform, with $\sigma=\theta$. In this scenario, the optimal LRT attack theoretically performs identically to the Bayesian attack (since the likelihood ratio coincides with the posterior ratio) in terms of the ROC curve (and hence AUC), still consistent with the ordering established in Theorem 2. Thus, in such cases, the defender can use the optimal LRT attack as a constraint to achieve the same optimal privacy-utility trade-off as when considering the Bayesian attacker.
> >
> > However, frequentist approaches, such as LRT, lack a straightforward mechanism to incorporate prior knowledge, unlike Bayesian methods that explicitly use priors. Instead, frequentist approaches typically rely on indirect or ad hoc techniques, such as the careful design of experiments, proper model assumptions, or estimation methods approximating prior-like effects (e.g., empirical Bayes methods).
> >
> > These limitations of frequentist approaches make using our BGP response as the privacy constraint a more straightforward, easier-to-implement, and theoretically robust alternative to the optimal LRT attack, particularly when prior privacy risk analysis (via the prior) is involved.
> >
> > To the best of our knowledge, there is no existing attack model that employs the Neyman-Pearson optimal (i.e., UMP test) LRT in summary statistic privacy that is both (i) optimally adapts to the defender's strategy and (ii) achieves Neyman-Pearson optimality. Existing fixed (threshold) LRT model (Sankararaman et al., 2009; Shringarpure & Bustamante, 2015; Venkatesaramani et al., 2021; 2023) exploit intrinsic properties of binary summary statistics, such as SNV data, but do not adapt to the defender's strategy. Meanwhile, the Adaptive (threshold) LRT attacker (Venkatesaramani et al., 2021; 2023) adopts a heuristic threshold selection scheme.
> >
> > **2: ROC of Adaptive LRT**
> >
> > The **ROC curves of the Adaptive LRT attacker** are indeed partially shown due to the nature of its threshold selection scheme, which inherently limits the range of the thresholds explored within the LRT framework. As a result, the ROC curve representation is restricted. As shown in the plots, the Adaptive LRT reduces false alarms (lower FPR) but at the cost of missing true positive (lower TPR). These trade-offs arise directly from its adaptive thrsholding mechanism under a given defense privacy strategy *and are not omissions in our experiments*.
> >
> > We will ensure the final manuscript clarifies these issues.

---

> > ### Author Response · Authors · 2024-11-29
> > **Response to Official Comment by Reviewer FHZn (Part 3)**
> >
> > > Comment 3: I am also having some trouble understanding the theoretical guarantee of Theorem 2. It is not clear to me how one would set about finding an aligned prior. This seems critical for ensuring a meaningful privacy guarantee.
> >
> > **Response:**
> >
> >
> > **Our privacy guarantee:**
> >
> > Our privacy guarantee for the BNGP framework, which ensures an optimal privacy-utility trade-off, is theoretically established by **Theorem 1**. Importantly, **finding an aligned prior is not a pre-requisite for our privacy guarantee.** Specifically, the first two inequalities in (i) and (ii) of Theorem 1 ensure that **no other attacker**, based on any conditional probability (**including those derived from frequentist approaches** given their significance level and power) defined by the corresponding ideal neural network $\widehat{H}$, can induce a strictly higher privacy loss as perceived by the defender than the BGP response. Here, the defender is modeled as a von Neumann-Morgenstern decision-maker who perceives the privacy risk as the expected privacy loss.
> >
> >
> > **Theorem 2:**
> >
> >
> > Theorem 2 establishes the theoretical ordering of privacy loss induced by the Bayesian attacker and three frequentist attackers as perceived by the defender when the Bayesian attacker's subjective prior is aligned, which (i.e., the comparison between Bayesian and optimal LRT attackers) reinforces the guarantees established by Theorem 1. Importantly, this does not imply that our privacy guarantee becomes non-robust when the Bayesian attacker's subjective prior is not aligned.
> >
> >
> > In the Bayesian game, the equilibrium assumes that the defender is aware of the attacker's subjective prior $\sigma$, including the case where $\sigma=\theta$. The (subjective and true) prior reflects the "prior privacy exposure/risk", and leveraging the attacker's prior knowledge is well-established concept in security and privacy frameworks.
> >
> >
> > As also shown in our response to [Q3](https://openreview.net/forum?id=o4X6UM18rI&noteId=BgGbytODeq) of reviewer 9RQW, the subjective prior reflects information about *what has already been compromised or at risk*. Our formulation involving subjective priors captures two key scenarios: (1) when the prior privacy risk (i.e., the attacker’s knowledge) is evaluated or estimated by the defender, and (2) when the overall privacy risk is accounted for by the defender, such as through knowledge of the number or type of data accesses that have already occurred (e.g., privacy accounting) before the current design of the private mechanism. This knowledge can be systematically incorporated into the analysis via subjective priors, akin to updated priors in sequential Bayesian updates.
> >
> >
> >
> >
> > Now, suppose that the evaluation of prior privacy risk suggests that the subjective prior that can be held by any attacker is **not aligned**. If the defender is aimed at addressing a specific attacker with a non-aligned subjective prior (excluding other attackers, such as frequentist ones), the corresponding BNGP strategy is optimal for that scenario. On the other hand, if the subjective prior reflects the overall prior privacy risk, the BNGP strategy designed by considering $\sigma = \theta$ remains robust. As demonstrated in Theorem 2, no frequentist approach can result in a strictly worse privacy loss for the defender when $\sigma = \theta$.

---

> > ### Author Response · Authors · 2024-11-29
> > **Response to Official Comment by Reviewer FHZn (Part 4)**
> >
> > > Comment 4: There is also a connection to existing privacy guarantees of the DP framework through Proposition 6 but this seems to be mostly a connection in definition-only. This result would be strengthened by an algorithm or explanation how to train the network so as to ensure all BGP responses are in fact eps-BGBP. In any case, it is still not clear what improvement is offered over the existing DP framework.
> >
> > **Response:**
> >
> > **Regarding the improvement over DP framework:**
> >
> > We appreciate the reviewer's thoughtful feedback. Our primary goal in this paper is **not** to improve privacy guarantee for specific mechanisms where the standard differential privacy (DP) framework can be readily implemented. Instead, as mentioned in our response to Comment 1, our BNGP strategy (Definition 3, without involving $\mathtt{DPH}$) offers an alternative approach to achieving an optimal privacy-utility trade-off by avoiding some of the challenges inherent in the standard DP framework.
> >
> > To clarify, **Theorem 1** demonstrates the theoretical privacy robustness of our framework. **Proposition 5** shows that mechanisms under our BNGP strategy are inherently DP, although determining the precise corresponding DP parameters is in general intractable.
> >
> >
> > Notably, our framework does not require knowledge of DP parameters like $\epsilon$ for its functionality. Furthermore, the advantages of our framework over the DP framework are independent of whether these parameters are known. Importantly, our framework measures privacy risk and utility loss in terms of expected loss, rather than relying on parameters such as those used in DP. This approach aligns with practical applications like financial risk management, where expected loss is a common metric for evaluating risk.
> >
> >
> > **Regarding Proposition 6:**
> >
> > Proposition 6 provides a way to ensure the mechanism using the BNGP strategy satisfies $\epsilon$-DP **with a known $\epsilon$**, if desired. It is important to note that the constraints captured by $\mathtt{DPH}$ differ from the probabilistic indistinguishability definition of DP. Specifically, $\mathtt{DPH}$ imposes constraints directly on the attacker's responses, rather than on the conditional probabilities between the data and mechanism outputs in the DP definition. This parameterization is also independent of sensitivity calculation.
> >
> >
> > One motivation for knowing the DP parameters is to enable compatibility between mechanisms in our BNGP framework and other DP mechanisms accessing the same dataset, particularly in global privacy accounting that uses DP parameterization schemes.
> >
> >
> > **On Algorithm Design and Explanation for Training:**
> >
> > Thank you for suggesting the development of algorithms for training the network constrained by $\mathtt{DPH}$. We agree that this is an important directin for future research. In this work, we adopt the well-established penalty-based method to incorporate the constraint into the defender's objective function. We will include a detailed description of this method in the appendix of the final version.

---

> ### Comment · Reviewer_FHZn · 2024-12-02
>
> I thank the authors for their efforts in addressing my comments, especially for the new optimal LRT plot.
>
> Nonetheless, I remain highly concerned about the privacy guarantees of your proposed framework. You are of course correct that privacy accounting for standard DP has outstanding computational and exactness issues. You are also correct that optimal privacy-utility tradeoffs are not fully understood in the standard DP framework. These issues essentially lead to pessemistic upper bounds on the privacy loss and thus unnecessary utility loss. As far as I can tell, however, your framework does not provide a guarantee (upper bound) on the privacy leakage. Theorem 1, for instance, asserts that the privacy loss under the proxy CEL attacker loss leads to no worse privacy leakage than under the true Nash equilibrium, yet there is no characterization of the Nash equilibrium privacy loss. Theorem 2 also does not provide a bound on the Bayesian privacy risk. Worse still, incomplete training of the GAN may also lead to additional privacy loss that is not accounted for in your framework. Under the standard DP setting, incomplete training does not result in an incorrect privacy loss guarantee.
>
> Overall, I think extending the standard DP framework to the Bayesian setting is a very nice direction and I greatly appreciate the authors efforts to improve the clarity of their work. However, due to a lack of a hard guarantee on the privacy leakage in both the idealized case (GAN fully trained) as well as the realistic case (GAN unfully trained), I will leave my score as it is.

---

> > ### Author Response · Authors · 2024-12-02
> > **Thank you**
> >
> > We sincerrely thank the reviewer for their thoughtful feedback and appreciation of our efforts to clarify and improve the manuscript. We understand the reviewer's concerns regarding the privacy guarantees in our framework and would like to provide the following clarifications.
> >
> > > **Regarding the lack of a hard guarantee on the privacy leakage in the idealized case (GAN fully trained)**
> >
> > **Response:**
> >
> > We indeed have the characterizations of theoretical privacy guarantee in the idealized case when the GAN are fully trained. In particular, **Theorem 1** shows that the Nash equilibrium achieved by the BNGP strategy (when the attacker uses the BGP response) represents the theoretical upper bound on the worst privacy loss that the defender can achieve for a given objective function. In our framework, $\mathcal{L} _{\textup{CEL}}$ quantifies the worst privacy loss for a given privacy strategy and satisfies the properties of post-processing and composition.
> >
> > The Bayesian privacy risk in **Theorem 2** is indeed the theoretical upper bound for the privacy loss as perceived by the defender under any given privacy strategy (not necessarily the defender's equilibrium strategy).
> >
> >
> > > **Regarding Privacy Risk in the Realistic Case::**
> >
> > **Response:**
> >
> >
> > We acknowledge the reviewer's concen regarding the lack of theoretical privacy loss characterization in realistic cases where the GAN is not fully trained.
> >
> >
> > In this work, our primary focus is on optimizing the privacy-utility trade-off for a defender-customized objective function, with particular emphasis on the utility part. This emphasis is especially relevant in certain critical applications, such as medical and genetic data analysis, where both privacy and utility are important, but utility often taks precedence. In these scenarios, achieving high utility is essential for accurate and meaningful outcomes, such as diagnostics, treatment planning, or research insights, which make it crucial to prioritize utility in the optimization process.
> >
> >
> > We adopt a **principled yet pragmatic approach** to achieve a tighter privacy-utility trade-off. The proposed approach enables **end-to-end optimization** without intermediate steps, such as sensitivity calculation, or artificial operations like gradient clipping that are commonly required in standard differential privacy.

---

### Official Review · Reviewer_vciw · 2024-11-06

**Soundness:** 3
**Presentation:** 2
**Contribution:** 3
**Rating:** 6
**Confidence:** 2

**Summary:**

The paper focus on the privacy protection against MIA under data sharing scenarios. They propose a Bayesian game model for the data sharing with the defender minimizes a combination of expected utility and privacy loss, while the Bayes-rational attacker maximizes privacy loss. To approximate a Bayes-Nash equilibrium, the author then propose a GAN-style algorithm. They also introduce Bayes-Nash generative privacy and Bayes generative privacy risk, and prove the composition and post-processing properties for BGP risk. Experiments are conducted to demonstrate the performance of the proposed approach in genomic summary statistics sharing scenario.

**Strengths:**

* A Bayesian game model is proposed for the privacy-preserving data sharing process
* New privacy measure is proposed with composition and post-processing propoerties
* Both theoretical analysis and empirical results are provided for the proposed Bayesian game-theoretic method
* Notations are clearly defined

**Weaknesses:**

* Since v(p, b) and the loss function be analyzed are proxies, the analysis of the approximation error should be provided.
* It would be better to provide the complexity analysis of the proposed method, along with the runtime of the defense mechanisms used in the experiments.

**Questions:**

In Eqn.2, why the coefficient of two terms are different (1-\gamma, \gamma)?

---

> ### Author Response · Authors · 2024-11-21
> **Response to Reviewer Comments**
>
> We thank the reviewer for their valuable comments and suggestions!
>
> > **W1: Since v(p, b) and the loss function be analyzed are proxies, the analysis of the approximation error should be provided.**
>
> **Response:**
>
> As demonstrated in Section 3.1, the proxies we have chosen do not induce approximation errors. Instead, these proxy-based formulations allow the defender to achieve the most robust privacy protection strategies for any objective function that captures the privacy-utility trade-off while satisfying Assumption 1. This ensures that the defender’s strategy directly aligns with the true optimization objective, eliminating the need for additional approximations or error-prone surrogates.
>
> > **W2. Complexity Analysis and Runtime of Experiments**
>
> **Response:**
>
> In Appendix P.9, we have included a comprehensive complexity analysis of both the Attacker (D) and Defender (G) neural network architectures. This analysis describes the number of trainable parameters and the computational complexity for both networks, accounting for their input dimensions, hidden layer configurations, and output dimensions. Specifically, we analyze the dominant operations, such as the forward passes through linear layers, and provide clear expressions for the overall computational complexity.
>
> Regarding runtime, the experiments were conducted using an NVIDIA A40 48G GPU with PyTorch as the deep learning framework. While precise runtimes for individual forward or training passes were not recorded, most experiments completed within 2 hours, with the most computationally intensive configurations taking up to 6 hours. These durations reflect the computational feasibility of our framework, given the complexity of the tasks and the scale of the models.
>
>
> > **Q1: In Eqn.2, why the coefficient of two terms are different (1-$\gamma$, $\gamma$)?**
>
> **Response:**
>
> We thank the reviewer for their question. Equation (2) introduces the Bayes-weighted membership advantage (BWMA), which extends the standard per-individual membership advantage (MA) in Equation (1). This extension incorporates the following considerations to model the attacker's decision-making process more comprehensively:
>
> 1. **Prior Knowledge:** BWMA leverages the prior distribution $\theta$ of the membership information. This reflects the attacker's Bayesian decision-making, accounting for their prior knowledge about the dataset.
>
> 2. **Heterogeneous Preferences:** BWMA introduces the parameter $0 < \gamma \leq 1$ to capture the trade-off between true positive rate (TPR) and false positive rate (FPR). The coefficients $(1-\gamma)$ and $\gamma$ represent the attacker's weighted preferences for maximizing TPR and minimizing FPR, respectively:
>    - A smaller $\gamma$ emphasizes maximizing TPR, reflecting scenarios where the attacker prioritizes correctly identifying members of the dataset.
>    - A larger $\gamma$ prioritizes minimizing FPR, corresponding to scenarios where the attacker seeks to reduce false positives.
>    - When $\gamma = 0.5$, the attacker values TPR and FPR equally, leading to a balanced trade-off.
>
> Thus, the coefficients $(1-\gamma)$ and $\gamma$ are designed to introduce flexibility in modeling diverse attacker strategies based on their preferences. This extension ensures that the BWMA captures a broader spectrum of attacker behaviors, enhancing the applicability of the framework.

---

### Official Review · Reviewer_1kHk · 2024-11-08

**Soundness:** 3
**Presentation:** 2
**Contribution:** 2
**Rating:** 6
**Confidence:** 2

**Summary:**

This paper proposes a Bayesian game model for formalizing the risk of disclosure with respect to data sharing. In this framework, the defender aims to minimize his privacy loss under a specified level of utility while the attacker aims at maximizing its membership advantage. A Generative Adversarial Network (GAN) approach is proposed for training the perturbation mechanism and model the interactions with the attacker.

**Strengths:**

-The related work section clearly summarizes the previous work on the quantification of privacy leakage as well as how to address formally the privacy-utility trade-off.

-The considered membership inference attack setting is clearly explained and formalized. The proposed approach aims to provide a firm foundation for developing optimal privacy mechanisms. An illustrative example based on the sharing of summary statistics is used to illustrate the proposed framework.

-To valide the proposed framework, experiments have been conducted on three datasets and the success of the resulting MIA has been compared against other state-of-the-art approaches. The results clearly demonstrate the potential of the approach with respect to other existing ones.

**Weaknesses:**

-The writing of the introduction is a bit confusing for someone who is not already familiar with the concepts used in this paper. It would help to rephrase it but also to provide an outline of the structure of the paper.

-Overall, the writing of the paper is highly technical and while the detailed proofs of the different lemma and theorems are given in Appendices, it would be great to provide some intuition or a sketch in the main paper.

-Some information are currently missing in the description of the experiments such as the value of the parameter epsilon used as well as the experimental details for the Adult and MNIST dataset.

-Actually, the framework considered that the data of each participant is just a binary value while in the majority of the setting (such as learning a machine learning model), the profile of the user is a feature vector.

-The training of a GAN is known to be difficult as some challenges have been addressed such as avoiding the overfitting of the discriminator to the generator. Ideally, it is good to also assess the privacy protection provided by training some external models, which is currently lacking in the current experiments.

**Questions:**

Please the main points raised in the weaknesses section.

---

> ### Author Response · Authors · 2024-11-21
> **Response to Reviewer Comments (Part 1)**
>
> We thank the reviewer for their valuable comments and suggestions!
>
>
> > **W1. The writing of the introduction is a bit confusing for someone who is not already familiar with the concepts used in this paper. It would help to rephrase it but also to provide an outline of the structure of the paper.**
> >
> **Response:**
>
> In the revised version, we have significantly improved the introduction to enhance its accessibility and clarity for readers unfamiliar with the concepts in this paper. The revised introduction now follows a structured approach: it clearly outlines the problem, highlights the motivation, introduces our proposed approach, and conveys the key conceptual contributions.
>
> We have simplified the language to make the motivation and novelty of our approach more explicit. Specifically, the revised text clearly highlights how the Bayes-Nash Generative Privacy (BNGP) strategy addresses key limitations of standard differential privacy frameworks, including avoiding intractable sensitivity calculations, circumventing worst-case proofs for compositions, and enabling compositions of mechanisms with arbitrary correlations.
>
> Additionally, we have added an outline of the paper structure at the end of the introduction to guide readers through the organization of the paper. This structure provides a clear roadmap for the subsequent sections.
>
> > **W2. Overall, the writing of the paper is highly technical and while the detailed proofs of the different lemma and theorems are given in Appendices, it would be great to provide some intuition or a sketch in the main paper.**
>
> **Response:**
>
> In the appendixes, we have made the following revisions to improve accessibility for readers.
>
> * In **Appendix A**, we have included tables of notations categorized by sections to enhance clarity and make it easier for readers to follow the technical details.
> * In **Appendix B**, we provide descriptions and intuitive explanations for the proof of each theoretical result.
>
> > **W3. Some information are currently missing in the description of the experiments such as the value of the parameter epsilon used as well as the experimental details for the Adult and MNIST dataset.**
>
> **Response:**
>
> We thank the reviewer for pointing out the missing details in the description of the experiments.
>
> The section of experiments (Section 5) has been extensively revised to enhance clarity.
>
> The value of $\epsilon$ in Figure 1c is $1.25 \times 10^{5}$ (Appendix P.3 provides detailed explanation). The experiment for Figure 1c shows an example that illustrate the advantages of the Bayesian defender (i.e., using the BNGP strategy) over the standard DP in addressing defender-customized objectives for the privacy-utility trade-off when the same utility loss is maintained in the trade-off. Figure 1c shows that even with equal utility loss, the $\epsilon$-DP defense can incur significantly greater privacy loss under the Bayesian attack.
>
>
> Experimental details for Adult and MNIST dataset are provided in Appendix P.5.
>
>
> > **W4. The framework considered that the data of each participant is just a binary value while in the majority of the setting (such as learning a machine learning model), the profile of the user is a feature vector.**
>
> **Response:**
>
> We appreciate the reviewer's observation regarding the binary representation of the participant data. To clarify:
>
> 1. Our framework and results are established under general settings where each data point is represented as a feature vector. The theoretical guarantees and methodology explicitly account for such general cases, rather than being specific to binary attributes.
> 2. The membership vector is binary because it represents the two possible states for each individual $k$: inclusion in the dataset (i.e., $b_{k}=1$) and and exclusion from the dataset (i.e., $b_{k}=0$). This binary distinction pertains to the dataset membership status and does not limit the attributes of the data points themselves.
> 3. The case study in Section 4 specifically focuses on genomic datasets, where the attributes of each data point are binary due to the nature of Single Nucleotide Variants (SNVs). This example illustrates the applicability of our framework in such contexts.
> 4. Our experiments also include results on non-binary attribute datasets, such as MNIST.

---

> > ### Author Response · Authors · 2024-11-21
> > **Response to Reviewer Comments (Part 2)**
> >
> > > **W5. External Models**
> >
> > **Response:**
> >
> > We appreciate the reviewer’s suggestion to assess privacy protection using external models. In fact, our experiments already include comparisons involving multiple models beyond GAN-based strategies. For example, in Figure 1b, we compare the performance of our Bayesian defender (using the BNGP strategy) with Fixed-threshold LRT and Adaptive-threshold LRT defenders under a Bayesian attacker (using the BGP response). Figures 1e and 1f present results for four attackers (Bayesian, Fixed-threshold LRT, Adaptive LRT, and Score-based) when the mechanisms are protected by two non-GAN defenders: a new pure DP defender (Steinke & Ullman, 2016) and a DP defender with a peeling model (Dwork et al., 2018), both applied to protect the mean estimator (Cai et al., 2021). Additionally, in Figure 1g, the defender’s strategy is trained by best responding to a Score-based attacker (Dwork et al., 2015), which also does not involve GAN.
> >
> > We have extensively revised Section 5 to enhance the clarity of the experiments.
> >
> > We hope these examples address the reviewer’s concern, but we acknowledge that further exploration with additional external models could enrich the experimental analysis.

---

> > > ### Comment · Reviewer_1kHk · 2024-12-03
> > >
> > > I thank the authors for their answers and updates to the paper. I have raised slightly my score but with respect to the value of epsilon, I think that the value used is far too high to provide any meaningful guarantees. Furthermore, the use of additional external models is also important to assess the privacy protection of the approach.

---

> > > > ### Author Response · Authors · 2024-12-03
> > > > **Thank you**
> > > >
> > > > We sincerely thank the reviewer for their comments and greatly appreciate their updated score.
> > > >
> > > > > Comment 1: with respect to the value of epsilon, I think that the value used is far too high to provide any meaningful guarantees
> > > >
> > > > **Response:**
> > > >
> > > >
> > > > Figure 1c considers scenarios where the defender has heterogeneous preferences over privacy-utility trade-offs by assigning different weights $\vec{\kappa} = (\kappa _{j}) _{j \in Q}$ to various attributes (e.g., SNP positions in the genomic data) of the summary statistics (please see Appendix P.3 for more detail).
> > > >
> > > > Figure 1c illustrates an example where the $\epsilon$ of the standard differential privacy (DP) framework is chosen such that the framework induces the same expected utility loss for the defender as the BNGP strategy. However, in this scenario, the corresponding $\epsilon$ could become too high to maintain a meaningful privacy guarantee. In many practical scenarios, such as sharing the summary statistics of genomic data, DP with a small $\epsilon$ is often too conservative to be practical. Conversely, empirically optimal $\epsilon$ values (e.g., for a given trade-off preference) tend to be so high that they fail to provide meaningful privacy guarantees, while small $\epsilon$ values are insufficient to ensure any useful utility in practice.
> > > >
> > > >
> > > > We use this example to highlight that the standard DP framework may lack the flexibility needed to effectively address heterogeneous preferences over the privacy-utility trade-off.
> > > >
> > > >
> > > >
> > > >
> > > > > Comment 2: Furthermore, the use of additional external models is also important to assess the privacy protection of the approach.
> > > >
> > > > **Response:**
> > > >
> > > > We acknowledge the reviewer’s emphasis on the importance of assessing the privacy protection of our GAN-like approach using additional external models.
> > > >
> > > > Indeed, we have accessed the privacy protection of our approach using some external models. In Figures 1d and 1h, we evaluate the privacy performance of the Bayesian defender (i.e., using the GAN-based BNGP strategy) against various attackers, where only the Bayesian attacker employs the discriminator-based strategy (i.e., the BGP response). The other attackers, using LRT, adaptive LRT, score-based, decision tree, and SVM approaches, are examples of external models. Our experimental results demonstrate that the GAN-based BNGP strategy is robust against these external models, effectively maintaining privacy protection under these diverse attack scenarios.

---

### Author Response · Authors · 2024-11-24
**Discussion period ending soon**

Dear reviewers, we tried to address all of the concerns raised in our rebuttal and paper revision.  We would be grateful if you could take a look at our responses and let us know if they indeed address your main concerns, or whether further clarifications would be helpful.

Best,

Authors

---

### Meta-Review · Area_Chair_bFdh · 2024-12-21

**Metareview:**

The submission proposes a game-theoretic framework that models the privacy protection as a Bayesian game between a defender and an attacker, along with the Bayes-Nash Generative Privacy (BGNP) strategy, which seeks to achieve the optimal privacy-utility trade-off depending on the defender’s preference.

The main downside of this submission is that the proposed framework does not provide an upper bound on the privacy loss, in particular when the GAN is incompletely trained. As a result, in its present form, the work does not meet the bar for publication at ICLR.

**Additional Comments On Reviewer Discussion:**

The reviewers raised different concerns about the paper, including:

1) Poor writing and confusion notation
2) The lack of upper bound on the privacy loss in the proposed framework

While the author(s) have satisfactorily addressed 1) in the rebuttal and discussion, the concern 2) remains.

---

### Decision · Program_Chairs · 2025-01-22

Reject